# Integrating fragment-based screening with targeted protein degradation and genetic rescue to explore eIF4E function

Swee Y. Sharp[1], Marianna Martella[1], Sabrina D'Agostino [1], Christopher I. Milton[1], George Ward[2], Andrew J. Woodhead [2] ✉, Caroline J. Richardson [2] ✉, Maria G. Carr[2], Elisabetta Chiarparin[2], Benjamin D. Cons [2], Joseph Coyle[2], Charlotte E. East[2], Steven D. Hiscock[2], Carlos Martinez-Fleites[2], Paul N. Mortenson [2], Nick Palmer[2], Puja Pathuri[2], Marissa V. Powers [1], Susanne M. Saalau [2], Jeffrey D. St. Denis[2], Kate Swabey[1], Mladen Vinković[2], Hugh Walton[2], Glyn Williams[2] & Paul A. Clarke [1] ✉

Eukaryotic initiation factor 4E (eIF4E) serves as a regulatory hub for oncogene-driven protein synthesis and is considered a promising anticancer target. Here we screen a fragment library against eIF4E and identify a ligand-binding site with previously unknown function. Follow-up structure-based design yields a low nM tool compound (**4**, $K_d = 0.09 \mu M$; LE 0.38), which disrupts the eIF4E:eIF4G interaction, inhibits translation in cell lysates, and demonstrates target engagement with eIF4E in intact cells ($EC_{50} = 2 \mu M$). By coupling targeted protein degradation with genetic rescue using eIF4E mutants, we show that disruption of both the canonical eIF4G and non-canonical binding sites is likely required to drive a strong cellular effect. This work highlights the power of fragment-based drug discovery to identify pockets in difficult-to-drug proteins and how this approach can be combined with genetic characterization and degrader technology to probe protein function in complex biological systems.

Protein synthesis accounts for around a quarter of a normal cell's energetic consumption and is increased in cancer cells where there is a continual demand for protein synthesis to support proliferation and survival[1,2]. In addition to the increased demand on protein synthesis following transformation, cancer cells often rapidly adapt to different physiological conditions and accomplish this by regulating protein translation[2–4]. This adaptability is frequently achieved by cancer cells hijacking the translational machinery through altering the activity or expression of regulatory factors[3,5].

All steps of the protein synthesis cycle are subject to regulation; however, initiation is the major rate-limiting point that controls translation[6,7]. Initiation involves the recruitment of ribosomes to the 5' end of the mRNA and occurs following the assembly of the eukaryotic initiation factor 4F (eIF4F) complex at the mRNA m7G cap. The eIF4F complex includes a core of eIF4E, eIF4G and eIF4A[5,6]. eIF4E recognises and binds the mRNA 5'cap, and recruits eIF4G to provide a scaffold function for the eIF4F complex, while eIF4A is an RNA helicase that is stimulated by binding to eIF4E and eIF4G[8].

The eIF4F subunits are linked to cancer in several ways including through gene amplification, being targets of oncogenes or having oncogenic activity in transformation or tumorigenesis experiments[2,9–17]. The eIF4E dependency of cancer cells has been demonstrated in a haplo-insufficient mouse model[18] where loss of one eIF4E gene allele did not limit global protein synthesis or embryonic

[1]RNA Biology and Molecular Therapeutics Team, Centre for Cancer Drug Discovery, Institute of Cancer Research, London SM2 5NG, UK. [2]Astex Pharmaceuticals, Cambridge Science Park, Cambridge CB4 0QA, UK. ✉e-mail: Andrew.Woodhead@astx.com; Caroline.Richardson@astx.com; Paul.Clarke@icr.ac.uk

development but was sufficient to induce resistance to cellular transformation and tumorigenesis. In addition to the wealth of data demonstrating that eIF4E contributes to oncogenic transformation, it is also clear that eIF4E is an essential regulatory hub in cancer signalling networks. Several oncogenic pathways converge on eIF4E: particularly the mTOR pathway that phosphorylates and regulates the activity of the eukaryotic translation initiation factor 4E binding proteins (4E-BP) suppressors of eIF4E activity[19,20] and the oncogenic RAS/RAF/MAPK pathway that promotes eIF4E activity through phosphorylation of S209 by the MAPK-interacting serine/threonine kinases[21].

Targeting eIF4E has long been considered a promising anticancer strategy but it has remained undruggable using conventional screening approaches. Tool compounds have been discovered that block capped mRNA from binding (such as synthetic m7-GTP analogues) or disrupt the eIF4E:eIF4G interaction[22–30] (such as 4EGI-1). However, these published tool compounds lack features of high-quality chemical probes that in addition to physiochemical properties such as solubility can include confirmed selectivity, cellular potency, and the availability of structurally related inactive controls[31]. To date there are no reports of these initial tool compounds progressing to chemical series with more drug-like physiochemical and pharmaceutical characteristics[31,32].

In contrast to high throughput screening, fragment-based screening uses smaller, less complex libraries to probe the whole protein, often using biophysical approaches[33–35]. Fragment screening has been used to discover binding sites in multiple systems, including KRAS, HCV protease, multiple kinases and TNFα.[33,34,36–40]. Here we set out to address the challenges of targeting eIF4E by screening a fragment library using a combination of ligand-observed NMR and X-ray crystallography to identify potentially druggable sites on eIF4E. Our unbiased approach differed from previously published screening approaches that specifically targeted either the mRNA cap-binding site[25,28–30] or the interaction between eIF4E and eIF4G peptide sequences[22,24,27]. To complement our fragment-based strategy, we coupled the screening approach with an eIF4E targeted protein

degradation (dTAG) and genetic rescue model to determine the contribution of fragment binding sites to eIF4E function[41].

## Results

### Protein engineering to enable X-ray fragment screening

In cells, eIF4E forms protein-protein interactions (PPI) with either eIF4G as part of the eIF4F translation initiation complex or with inhibitory 4E-BPs, but is unstable as a monomer[42]. Both eIF4G and the 4E-BPs bind to eIF4E at a common, canonical PPI site characterised by a large hydrophobic surface (Fig. 1a, b)[42,43]. Exposure of this hydrophobic surface in monomeric preparations of eIF4E affected the expression and purification of recombinant eIF4E, resulting in low yields and large amounts of aggregated material. The stability of recombinant eIF4E can be improved by the presence of m7-GTP, its analogues or m7-GTP capped mRNA, however, this would lead to occlusion of the cap-binding site which was undesirable for our screening purposes. To address this and produce sufficient recombinant eIF4E to enable our fragment screen, multiple eIF4E derived clones were evaluated to identify a clone which; 1) expressed high amounts of soluble protein, 2) was readily crystallized in the apo form, 3) diffracted to high resolution and 4) was suitable for soaking ligands. In total, 26 protein sequence modifications of eIF4E were explored including N-terminal truncations, GST-tagging, fusion proteins derived from 4E-BP1 (including different linker sequences), point mutations as well as two variants of eIF4E (D127 and N127[44]; Supplementary Data 1 and 2).

This strategy successfully identified a fusion protein where N-terminal residues 1–35 of eIF4E (N127 variant) were removed and replaced with the canonical binding sequence of 4E-BP1 attached via a flexible glycine-based linker to L36 of eIF4E (Fig. 1a, Supplementary data 1 and 2, Supplementary Fig. 1 and 2a). This resulted in an engineered form of eIF4E where the strongly hydrophobic region in the canonical binding site (Fig. 1b) is buried (Fig. 1c) making the protein more stable and less prone to aggregation than our starting expression

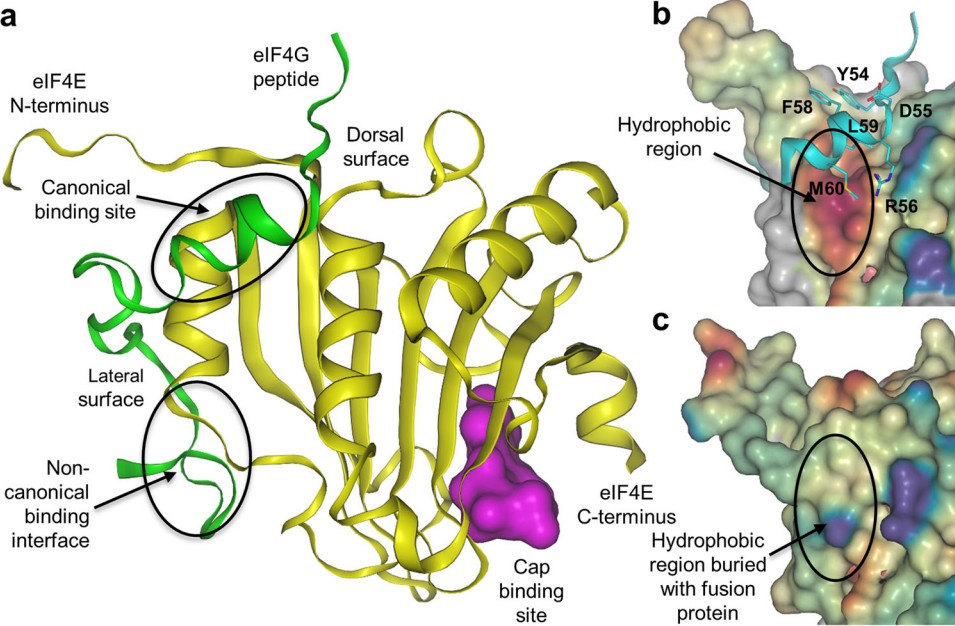

**Fig. 1 | Summary of eIF4E protein architecture and protein engineering to identify a suitable system for fragment screening. a** X-ray crystal structure of eIF4E (PDB: 5T46)[51] overlaid with a protein binding partner (eIF4G peptide). Protein secondary structure is represented as coloured ribbons, with eIF4E (yellow), and eIF4G peptide (green). The Connolly surface of m7-GDP is displayed in magenta. **b** Protein surface of eIF4E showing the canonical binding region as a Connolly surface coloured by hydrophobicity, with red indicating a strongly hydrophobic

(lipophilic) region. A fragment of 4E-BP1 is overlaid in ribbon representation (cyan). Residues highlighted (single letter codes) form part of the consensus sequence (YXXXXLφ) for the canonical binding partners, in this case 4E-BP1 (PDB: 3U7X)[53]. Where Y = Y54, X = any residue, L = L59, φ = lipophilic residue corresponding to M60 for 4E-BP1. **c** Protein surface of 4E-BP1-eIF4E fusion protein as a Connolly surface coloured by hydrophobicity.

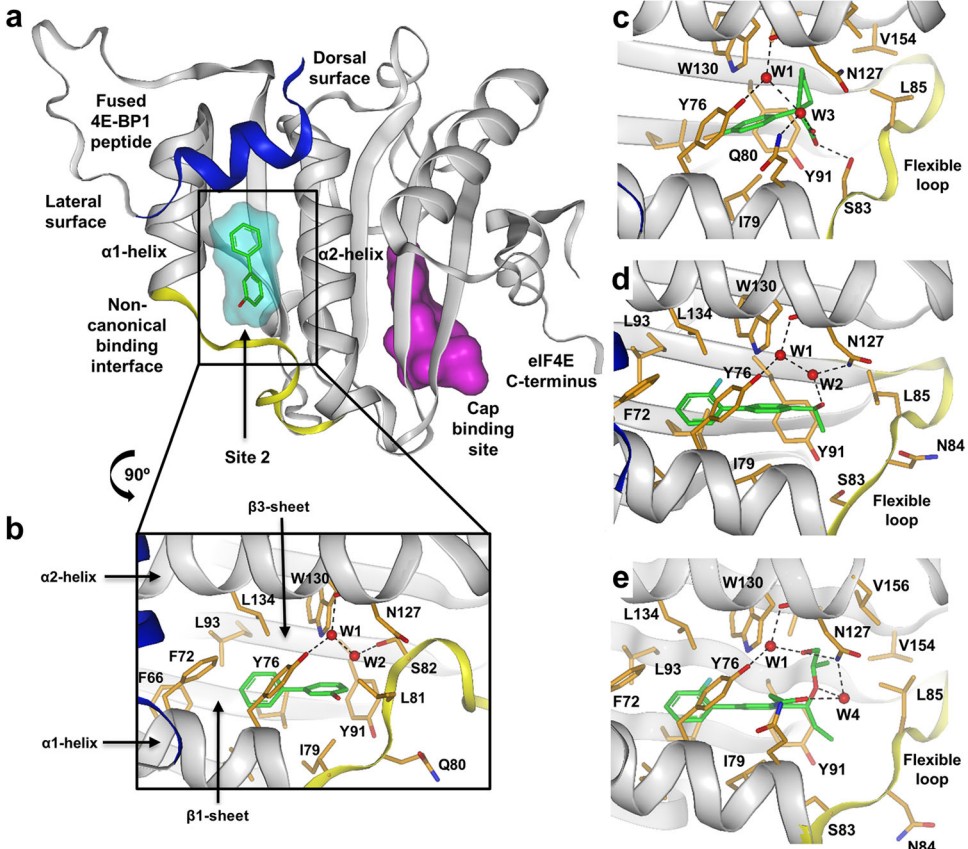

**Fig. 2 | Protein-ligand co-crystal structures of compounds 1–4 generated during fragment screening and fragment to lead optimisation.** The secondary structure of eIF4E is shown in ribbon representation. The majority of the protein (grey), canonical peptide derived from 4E-BP1 attached to the N-terminus of eIF4E (blue), loop region between the C-terminus of the α1-helix and N-terminus of the β3-sheet (yellow). The ligand is depicted as green lines. Sidechains from protein residues within 4 Å of the bound ligand are highlighted in orange with single letter codes. Hydrogen bonds between ligand, protein and water are denoted with black dashed lines. Key waters are shown as red spheres and labelled W1 – W4.

**a** Compound **1** bound in site 2 (resolution 1.85 Å). Key features are highlighted including the location of site 2 in comparison to the Cap binding site (the Cap-site ligand (m7-GTP) is shown for illustration purposes only and was not included during protein purification or the screening process). The surface representations of **1** and m7-GTP are shown in cyan and magenta respectively. **b** Magnified view of binding site 2 with compound **1** bound which has been rotated anti-clockwise by 90°. **c** Compound **2** structure (resolution 1.89 Å). **d** Compound **3** structure (resolution 1.93 Å). **e** Compound **4** structure (resolution 1.97 Å).

construct. Occlusion of the canonical eIF4G/4E-BP binding site could disadvantage screening using this engineered system, as potential hits would be missed. However, this is balanced by the flat, strongly hydrophobic nature of the occluded region rendering fragment binding at this site unlikely. Importantly, a global alignment of the engineered eIF4E structure with the X-ray crystal structure of wild-type unmodified eIF4E (from PDB structure 5T46), reveals similar structural features verified by the root-mean-square deviation (RMSD) of 0.9 Å (Supplementary Fig. 2a), making it suitable for our fragment screening strategy.

### Fragment screening identifies two binding sites on eIF4E

A library of 1371 fragments was screened against the apo, engineered protein using Astex's Pyramid™ platform with a combination of ligand observed NMR and X-ray crystallography[35,45–47]. Fifty fragment hits were identified to bind to eIF4E (3.6% hit rate), with a small number occupying the mRNA cap-binding site (site 1) (Supplementary Fig. 2b) and the majority binding at a second site of unknown functional relevance (site 2), which at the time was unreported, but was subsequently identified by Fischer and colleagues for a biphenyl-derivative of 4EGI-1 (i4EG-BiP)[48] (Fig. 2a, b and Supplementary Fig. 2a). The biaryl compound **1** binds site 2 in an enclosed hydrophobic cavity bounded by residues from the α1 and α2 helices and the β1, 2 and 3 sheets, plus a loop region linking the C-terminus of the α1 helix with the N-terminus

of the β3 sheet (Fig. 2a, b and Supplementary Fig. 3a). The binding of compound **1** appears to be driven primarily by hydrophobic van der Waals contacts and does not make any direct polar interactions with the protein. Isothermal titration calorimetry (ITC) indicated that the binding affinity ($K_d$) of **1** for eIF4E (Table 1) was 400 μM with reasonable ligand efficiency (LE 0.36)[49].

Compound **2** bound in a similar region to **1** but caused elongation of the α1-helix and movement of the loop region (consisting of residues 80–86 between the α1-helix and β3-sheet) resulting in a larger more solvent exposed binding site (Fig. 2c and Supplementary Fig. 3b). The loop is mobile with high B-factors and incomplete density for several residues. The cyclopentyl-carboxylic acid occupies the region of L81 and S82 in the compound **1** structure, displacing a water molecule (W2) observed in the crystal structure of **1**, making interactions with Q80 S83, N127 and Y76. The affinity of this fragment was much weaker, $K_d$ = 7400 μM (Table 1) with low ligand efficiency (LE 0.19), possibly related to the energetic penalty paid for inducing the change in protein conformation and/or inadequately filling the hydrophobic binding site. The change in α1-helix conformation on compound **2** binding is similar to that observed when 4EGI-1 binds to an adjacent site on the solvent exposed lateral surface of eIF4E. Compound **2** binds in close proximity to the non-canonical binding region of eIF4G / 4E-BP described by both Sekiyama[50] and Gruner[51] (Supplementary Fig. 2c).

**Table 1 | Biophysical and biochemical data for site 2 binders**

| Compound | Structure | eIF4E $K_d$ (µM)[a] | LE[b] | Inhibition of PPI in HeLa cells (µM)[e] |
|---|---|---|---|---|
| 1 | | 400 | 0.36 | - |
| 2 | | 7400 | 0.19 | - |
| 3 | | 20 | 0.40 | - |
| 4 | | 0.017 (0.09)[c,d] | 0.42 (0.38) | 1.4 µM (> 100 µM) |
| 5 | | - (3.0)[d] | - (0.3) | > 100 µM (> 100 µM) |

[a]Values were determined by isothermal titration calorimetry. $K_d$ values are $n = 1$ unless otherwise stated. [b]Ligand efficiency (LE) with units of (kcal/mol)/heavy atom[49]. [c]$K_d$ value for compound **4** is $n = 2$. [d]Values in parentheses are for the major variant of eIF4E (D127) all other data is for the minor variant D127N. [e]Compound effects on the interaction between eIF4E and eIF4G in HeLa cell lysates ($n = 4$), the effect on the eIF4E–4E-BP1 interaction is shown in parentheses. For all assay details see Experimental Section.

Analysis of wild type eIF4E crystal structures (PDB: 4UED, [https://doi.org/10.2210/pdb4UED/pdb], 3U7X, 5T46)[51–53] confirmed the existence of site 2 in published eIF4E structures. We also performed a cross species protein sequence analysis to assess the possible functional relevance of site 2[54]. We found higher sequence conservation was observed for site 2 compared to the global eIF4E sequence, with a similar conservation profile to the mRNA cap-binding site and slightly higher conservation than the canonical PPI site, suggesting that site 2 may have a functional role (Supplementary Figs. 4 and 5, Supplementary data 3 and 4).

**Fragment optimisation to a potent lead**

With the knowledge that site 2 had high cross species sequence conservation, we set out to assess whether it might prove more druggable than the cap-binding site as the polar charged nature of this site has provided a significant challenge for the development of cell active compounds[25,26,28–30]. We used structure guided optimisation to progress the low affinity fragments into tight binding chemical leads to investigate the functional relevance of this site.

Here we highlight the key steps leading to the discovery of a tight binding lead compound (Table 1). Initial optimisation of compound **1** focused on more efficiently filling the hydrophobic cavity and growing from the 4-position of the biphenyl to understand the structural requirements for inducing the loop movement. This led to compound **3** (Fig. 2d). Introduction of a fluorine at the 2'-position helped to reinforce the slightly twisted biaryl bound conformation and the fluorine atom fills a small lipophilic indentation formed by the sidechains of L45, L93, W130. Attempts to form polar contacts with the

NH of W130 or introduce polarity elsewhere in this region failed due to the enclosed hydrophobic nature of the binding site. We next installed a branched benzylic alcohol at the 4-position, which adds only 3 heavy atoms; the phenolic OH was removed as it did not appear to contribute to binding. This resulted in the same loop movement we observed for **2**. The hydroxyl group forms water mediated hydrogen bonds (W1, W2) with the sidechains of N127 and Y76 and the backbone carbonyl of N127. These two modifications resulted in a 20-fold improvement in affinity and a more efficient binder (Compound **3**, $K_d = 20$ µM; LE 0.4).

Compound **3** was further optimised to compound **4** by growth in three directions (Fig. 2e). (1) The hydroxyl of the benzylic alcohol was capped with a chiral ethylene glycol chain which resulted in the displacement of a weakly bound water molecule (W2 was also displaced by the binding of compound **2**) and the formation of a direct hydrogen bond to the sidechain of N127, whilst maintaining interactions with the sidechain of Y76 and the backbone carbonyl of N127 through a second water molecule (W1). The sidechain methyl group binds in a small lipophilic indentation formed by residues L85, L126, V154 and V156. (2) Growth from the 3-position of the biaryl with an ethanolamine side-chain resulted in an additional through water (W4) polar interaction to the sidechain of N127, whilst stabilising the bound conformation through a pseudo-6-membered ring between the NH and lone pair of the benzylic ether. (3) Extending the methyl to an ethyl fills the pocket formed by I79, S83 and Y91. Combining these modifications gave a compound with multiple polar interactions and excellent shape complementarity to the loop out conformation of site 2 (Figs. 2e, 3a, b and Supplementary Figs. 6 and 7a) leading to a significant improvement in binding affinity ($K_d = 0.017$ µM; LE 0.42) equating to over four orders of magnitude improvement from the initial fragment hit (Table 1). When compound **4** was profiled against the major variant of eIF4E (D127), the protein ligand crystal structure showed a similar binding mode for both variants (Supplementary Fig. 7b). A small change in flexible loop conformation (around L85) was observed for the major D127 variant compared to the minor N127 which may in part explain the 5-fold reduction in binding affinity ($K_d = 0.09$ µM; LE 0.38) (Table 1 and Supplementary Fig. 8). Thermodynamically, binding of compound **4** to both eIF4E variants was driven by a large favourable enthalpic contribution with a small entropic penalty (Supplementary Fig. 8). In all cases the stoichiometry estimates indicated a 1:1 interaction between compound **4** and eIF4E.

During the development of tool compounds or drugs, a recommended practice is to use a negative control compound to give confidence that the observed cellular phenotype may be driven by inhibition of the targeted protein. Comparison of the properties of chemical probes and their matched negative controls suggests that chemical similarity is a critical factor that dictates the value of a negative control[55]. The diastereoisomer of **4**, compound **5**, with inverted stereochemistry at the benzylic position fulfilled the criteria for a negative control as it had the same chemical composition as **4** but was unable to form the same network of hydrogen bonding interactions as **4**, leading to significantly reduced potency (Supplementary Fig. 7c, d; Table 1). To provide additional confidence in the selectivity/on-target activity of **4** we used **5** as a negative control in subsequent biological validation work[55].

**Compound 4 inhibits eIF4G binding in cell lysates**

Compound **4** binds tightly to eIF4E, inducing an extension to the α1 helix and a conformational change to the I79 – L85 loop. Comparison of the compound **4**-eIF4E structure with that of an eIF4G peptide[51] bound to eIF4E suggested that this conformational change would potentially impact on binding of eIF4G due to steric clashes between residues H78-N84 from eIF4E with residues D638-L641 from the eIF4G peptide (Fig. 3c, d). We used compound **4** to evaluate whether binding at site 2 could influence the formation of the eIF4F complex and impact on cap-dependent translation. Compound **4** treatment

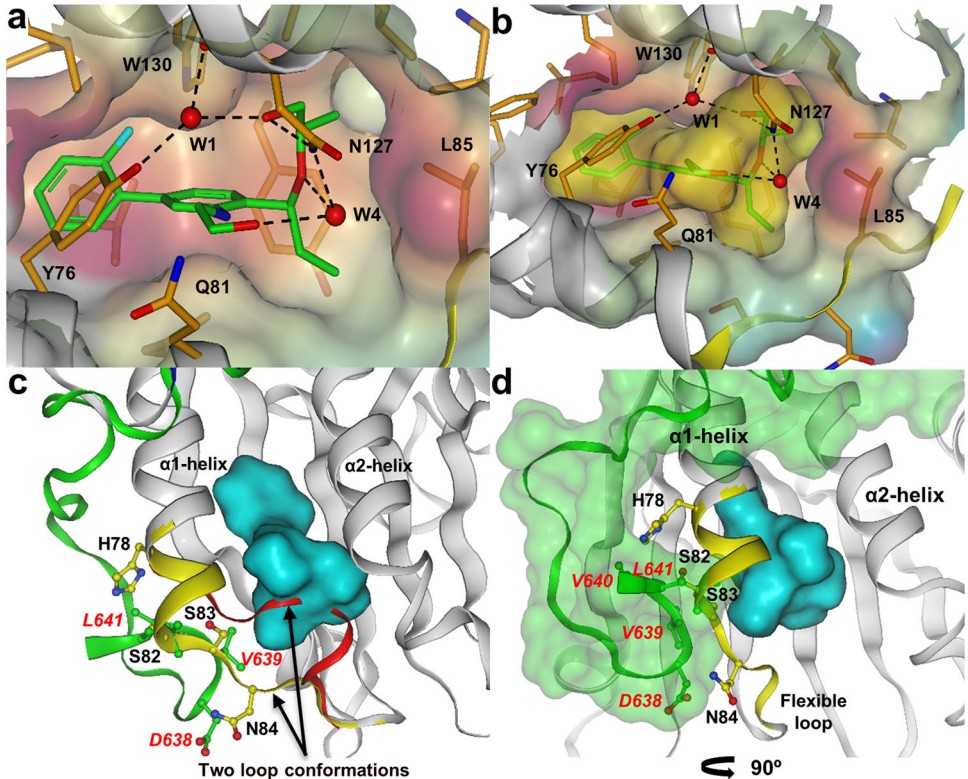

**Fig. 3 | Surface representations of site 2 and the non-canonical binding interface.** Key residues from eIF4E (black) or eIF4G peptide (red italics) are shown as single letter codes. **a** Compound **4** structure showing the protein Connolly surface coloured by hydrophobicity, red indicating strongly hydrophobic areas. As the site is highly enclosed, the surface associated with W76 and Q81 has been removed to improve visualisation. **b** Compound **4** structure showing the protein Connolly surface coloured by hydrophobicity and the ligand Connolly surface (yellow). **c** Overlay of eIF4G peptide (green ribbon) with the protein conformation of compound **4** bound eIF4E (grey ribbon), the reorganised α1-helix and flexible loop region is displayed as a yellow ribbon, the Connolly surface of compound **4** is displayed in cyan. The original loop conformation from the 5T46 structure of eIF4E bound to an eIF4G peptide is displayed as a red ribbon. **d** Overlay of eIF4G peptide (green ribbon and green Connolly surface) with the protein conformation of compound **4** bound eIF4E. The orientation has been rotated by ~90° in the horizontal plane compared to that shown in Figure 3c.

disrupted eIF4G binding to eIF4E determined by co-immunoprecipitation with an eIF4E antibody from SW620 human colorectal cancer cell lysates (Fig. 4a). 4E-BP1 suppresses translation initiation by competing with eIF4G for a common canonical binding site on eIF4E[19]. Consistent with this mode of competitive binding, incubation of cell lysates with a peptide (RIIY; RIIYDRKFLMECRNSPV)[56] derived from 4E-BP1 inhibited binding of both eIF4G and 4E-BP1 to eIF4E (Fig. 4a). This contrasted with compound **4** treatment, which did not disrupt 4E-BP1 binding to eIF4E even when eIF4G binding was lost. In the same assay, the weaker affinity compound **5**, had a limited effect on eIF4G or 4E-BP1 binding at the top concentration tested (Fig. 4a).

We also developed a quantitative electro-chemiluminescent co-immunoprecipitation binding assay to assess eIF4E-eIF4G and eIF4E-4E-BP1 interactions in cell lysates (Supplementary Fig. 9a). The assay uses microwell plates coated with an anti-eIF4E or an anti-Flag-epitope tagged antibody to capture protein complexes containing endogenous eIF4E or exogenous Flag-tagged eIF4E from cell lysates. The captured eIF4E or binding partners (eIF4G or 4EBP1) are detected by specific secondary antibodies and quantified by an electro-chemiluminescent reaction. The positive control RIIY peptide, but not a negative control RIIG peptide, disrupted both eIF4G and 4E-BP1 binding to eIF4E in SW620 cell lysates (Fig. 4b, c; Supplementary Fig. 9b). As before, compound **4** treatment of SW620 cells only affected eIF4G (EC$_{50}$ = 2.6 μM) and not 4E-BP1 binding to eIF4E (Fig. 4b–d). Similarly, in HeLa cervical carcinoma cell lysates, treatment with compound **4** disrupted the interaction with eIF4G (EC$_{50}$ = 1.4 μM) but not 4E-BP1 binding (Fig. 4b, c, e). Treatment with compound **5** had a limited effect on eIF4G binding compared to **4** (Fig. 4e). Similar to the

active RIIY peptide, compound **4** also significantly inhibited eIF4G:eIF4E binding between 1 and 10 μM in H1299 Non-small cell lung cancer (NSCLC) cell lysates (EC$_{50}$ ≈ 1.5 μM; Fig. 4f, g). The less active control compound **5** had a weaker effect than compound **4** in the H1299 cell lysate binding assay (~10% inhibition at 100 μM; Fig. 4g).

In addition to disrupting the interaction with eIF4G in three different human cancer cell lysates, an in vitro translation assay from HeLa cells revealed that compound **4** significantly inhibited cap-dependent translation (Fig. 4h between 1 and 10 μM (EC$_{50}$ ≈ 4 μM). The effect on IRES-driven translation was negligible (>100 μM) and the weaker isomer **5** had no significant effect on cap-dependent or -independent IRES-driven translation (Fig. 4h).

### Compound 4 binds eIF4E in cells

Having demonstrated disruption of the eIF4E:eIF4G interaction in lysates from three different cell lines we next determined the activity of compound **4** in intact cells. We first assessed any potential causes for the drop-off in activity moving from cell lysates to intact cells, including compound stability, permeability and cellular target engagement. Characterisation of compound **4** in human liver microsomes suggested that **4** had moderate clearance in vitro (Supplementary information), however, the enzymes present in microsomes are unlikely to be present in cell line models at appreciable levels. Compound **4** also exhibited good permeability as measured in a Caco2 assay (Papp (A-B) 9.0 × 10−6 cm/s, efflux ratio 1.5; Supplementary information) indicating that poor cellular uptake or efflux was unlikely to influence compound **4** activity in intact cells (Supplementary information).

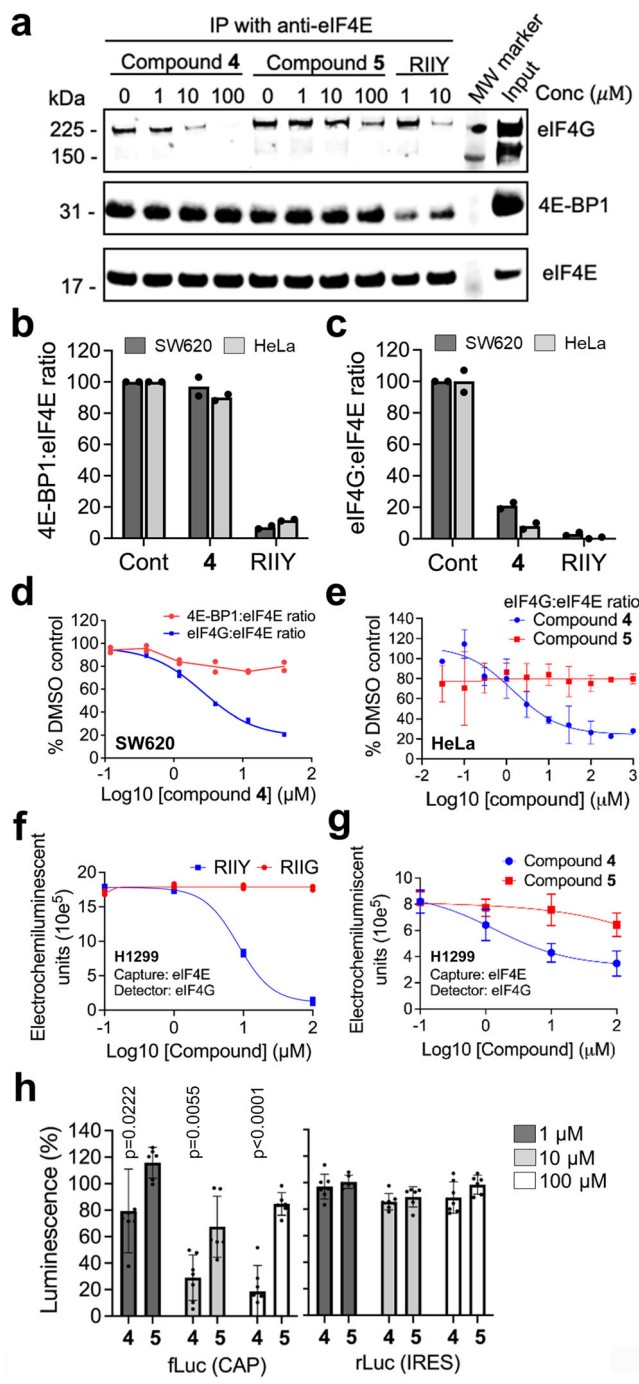

**Fig. 4 | Compound 4 inhibits eIF4G:eIF4E binding and cap-dependent translation in cell lysate assays. a** Lysates from SW620 cells were incubated with 1–100 μM compound **4** or **5** or positive control peptide (RIIY) for 30 min. Endogenous eIF4E was immunoprecipitated and immunoblotted for eIF4G, 4E-BP1 and eIF4E. Quantitation of 4E-BP1 (**b**) or eIF4G (**c**) with endogenous eIF4E in SW620 and HeLa cell lysates, determined by the electro-chemiluminescent binding assay following incubation for 30 min with DMSO vehicle (Cont), 100 μM compound **4** or 100 μM RIIY peptide. Complexes were immobilised by an eIF4E antibody and captured eIF4E, eIF4G and 4E-BP1 detected by their respective secondary antibodies. Values represent ratios of 4E-BP1:eIF4E or eIF4G:eIF4E electro-chemiluminescence relative to DMSO control (n = 2 biological replicates). **d** Electro-chemiluminescent assay for binding of eIF4G or 4E-BP1 with eIF4E in SW620 (n = 2 biological replicates), or (**e**) in HeLa lysates (n = 3 biological replicates, mean ± SD) following incubation for 30 min with compound **4** or **5**. Results are expressed as luminescence signals relative to DMSO control. **f** Quantification of eIF4E:eIF4G interaction in H1299 cells by electro-chemiluminescent assay. Cell lysates treated with RIIY 4E-BP1 derived peptide or RIIG negative control peptide at 0.1–100 μM for 30 min (n = 2 biological replicates). **g** Quantification of the endogenous eIF4E:eIF4G interaction in H1299 cell lysates at 0.1–100 μM (for 6 h) of compound **4** or **5**, as measured by electro-chemiluminescent assay (mean ± SD from n = 3 biological replicates). **h** HeLa cell lysates for in vitro translation were incubated for 30 min with 1, 10, 100 μM of compound **4** or **5**. Results are expressed as firefly or renilla luminescence signal normalized to DMSO control and expressed as % (mean ± SD from n = 3 biological replicates). Significance was determined using two-sided unpaired t-test comparing compound **4** to compound **5** at each concentration. Statistically significant p-values (p < 0.05) are shown on the plot and source data is located in the Source Data file.

less active control compound **5** also showed some limited evidence of activity (Fig. 5c).

Disappointingly, compound **4** lacked inhibitory activity (ns, p > 0.05) in dual reporter assays of cap-dependent and -independent protein synthesis established in H1299 or HEK293 human embryonic kidney cells (Fig. 5d; Supplementary Fig. 10a). Cap-regulated translation of eIF4E-dependent proteins in SW620 cells is inhibited by dephosphorylated 4E-BP1 following exposure to the mTORC2 kinase inhibitor AZD8055[58]. In an SW620 cell reporter assay for c-Myc-activity, AZD8055 clearly inhibited activity whereas compound **4** showed no activity at concentrations predicted from cell lysates to be active (Supplementary Fig. 10b). Not surprisingly, given the lack of activity in protein synthesis assays, compound **4** lacked on-target antiproliferative activity in H1299 and SW620 cells, with no difference in the activity of compounds **4** and **5** in H1299 cells following exposure to the highest concentration for 96 hr (50 μM; Fig. 5e; Supplementary Fig. 10c).

Overall, in contrast to our findings in biophysical and cell lysate assays, binding of compound **4** to eIF4E in intact cells translated poorly to the expected inhibition of eIF4G binding or cap-dependent protein synthesis and proliferation. We therefore decided to explore the role of site 2 using complementary genetic approaches.

## Mutation of eIF4E site 2 disrupts eIF4G binding in cells

We selected the human H1299 NSCLC cell line for these studies as literature[59], CRISPR DEPMAP public data (https://depmap.org/portal/gene/EIF4E?tab=overview) or in-house shRNA targeting eIF4F components, showed that eIF4A1, eIF4E1 and eIF4G1 were essential for H1299 cell survival (Supplementary Fig. 11).

We used our X-ray crystal structure analysis of compound **4** binding to design and construct a panel of mutations predicted to impact the compound **4** binding site 2 (L45, Y76, W130, L134, I79 and L85; Fig. 6a; Supplementary Fig. 12a, b; Supplementary Table 2). We also constructed control eIF4E mutants (Fig. 6a; Supplementary Fig. 12a, b; Supplementary Table 2) reported to disrupt known biological functions of eIF4E. These included: a defective m7G cap-binding mutant (W56A) that will block translation initiation and nuclear–cytoplasmic mRNA export, a mutant reported to disrupt

A cellular thermal shift assay (CETSA) performed in intact cells demonstrated target engagement, with compound **4** binding and stabilising eIF4E[57]. We initially showed that treatment of H1299 cells with 50 μM of **4**, but not **5**, protected eIF4E from thermal denaturation (Fig. 5a). We subsequently showed clear dose-dependent thermal protection of eIF4E (EC$_{50}$ = 2 μM; Fig. 5b) in intact cells which was similar to the EC$_{50}$s for disrupting eIF4G:eIF4E binding in lysates (Fig. 4). The electro-chemiluminescent binding assay also showed significantly greater inhibition of eIF4G:eIF4E binding by compound **4** compared to **5**. However, there was a drop-off in the activity of **4** in intact H1299 cells compared to eIF4E target engagement assay and lysate assays, as a high concentration (100 μM) and 16 hr exposure were required for compound **4** to significantly disrupt eIF4E:eIF4G binding (Figs. 4 and 5c). In addition, at these high concentrations the

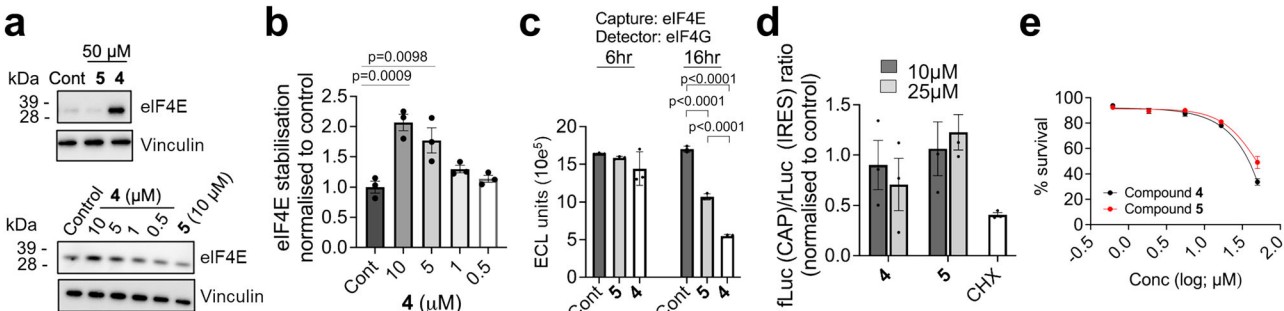

**Fig. 5 | Compound 4 binds eIF4E in intact H1299 cells. a** Top: Representative immunoblot (*n* = 2 biological repeats) showing eIF4E protein stabilisation in H1299 cells at 57.6 °C following compound treatment. Cells were treated with compound **4** or compound **5** (50 μM) for 6 hrs and then incubated at 57.6 °C. Bottom: Representative immunoblot (*n* = 3 biological repeats) following treatment with compound **4** (10, 5, 1, 0.5 μM) or compound **5** (10 μM) for 6 hrs followed by incubation at 57.6 °C. Vinculin was used for loading control. **b** eIF4E stabilisation in H1299 cells at 57.6 °C following compound **4** treatment was quantified from immunoblots using Image J and normalised to vinculin control. Statistically significant p values (*p* < 0.05) are shown on the plot (mean ± SEM from *n* = 3 biological replicates). **c** Quantification of the endogenous eIF4E:eIF4G interaction in intact H1299 cells treated at 100 μM for 6 or 16 hr (mean ± SD from *n* = 3 biological

replicates). Statistically significant *p*-values (*p* < 0.05) are shown on the plot. **d** Cells were transfected with a protein synthesis reporter expressing a bicistonic mRNA with a cap-dependent luciferase reporter (fLuc) and a cap-independent luciferase (rLuc) driven by a viral IRES and were treated with compound **4, 5** (10 or 25 μM) or cycloheximide (CHX; 100 μM) for 24 hr. Luminescence was measured using the Dual-glo luciferase assay. All p values for compounds **4** and **5** were not statistically significant (*p* > 0.05; mean ± SEM from n = 3 biological replicates). **e** Cell viability measured by cell titre blue assay in cells treated with compound **4** or **5** for 4 days (mean ± SEM from *n* = 3 biological replicates). For (**b**–**d**) significance (*p* < 0.05) was determined using an ordinary one-way ANOVA with Tukey multiple comparisons test, comparing compound **4** treatment with the control or negative control compound **5** treatment conditions. Source data is located in the Source Data file.

eIF4G/4E-BP binding (W73F) that will affect eIF4F complex formation, and a mutant of the conserved MNK1/2 substrate site that cannot be phosphorylated by MNK1/2 (S209A)[21,60,61].

Previous reports have demonstrated poor expression of some eIF4E mutants due to increased proteasomal degradation of the mutant resulting from disruption of the eIF4G/4E-BP interactions that protect eIF4E from turnover[60,62]. Therefore, we first determined the impact of eIF4E mutations on protein expression using a modified electro-chemiluminescent assay to capture exogenously expressed FLAG-tagged WT or mutant eIF4E (Supplementary Fig. 9a). This revealed that 7 of the 9 site 2 mutants exhibited very low expression (< 5% of the wild-type level) following transfection (Supplementary Fig. 12c). Only mutations that showed an expression level of > 25% of the wild-type control were used in subsequent experiments (W56A, W73F, L85R, L134R and S209A; Supplementary Fig. 12c).

We confirmed that eIF4E FLAG-tagged at the C- or N-terminus equally pulled down the endogenous eIF4G and that this interaction was disrupted by addition (post-lysis) of the 4E-BP competitor peptide (RIIY), but not by a closely related negative control peptide (RIIG)[56] (Supplementary Fig. 9c–f). We then used the same assay to measure endogenous eIF4G binding to exogenously expressed FLAG-tagged WT or mutant eIF4E. The W56A and S209A mutations had no significant impact on eIF4G binding whereas the W73F mutation disrupted the interaction between eIF4E and eIF4G as predicted from the literature[60,61,63] (Fig. 6b–d). The two site 2 mutations; L85R located in a flexible loop that undergoes a conformational change upon **4** binding and L134R located in the core of site 2 both significantly disrupted binding of eIF4G irrespective of their expression levels (Fig. 6b–d), indicating that modifications to site 2 can affect eIF4G binding in in cells.

## An eIF4E degradation model shows cell dependency on eIF4E

To further explore the role of site 2 in eIF4E and the potential impact of **4** binding to this site we explored the impact of site 2 mutations at a cellular and a molecular level. A CRISPR knock-in approach to interrogate the function of eIF4E mutants would be challenging, as altered or loss of eIF4E function for the extended period of time required to establish a knockout or knock-in isogenic line is unlikely to be tolerated by H1299 cells (Supplementary Fig. 11). Therefore, we used a genetic rescue approach where wild-type or mutant eIF4E are initially

expressed in the presence of endogenous cellular eIF4E. Retaining expression of endogenous eIF4E protects the cells from potential lethal effects of expressing a non-functional mutant thereby allowing establishment of the cell models. The impact of the exogenous control wild-type or mutant expression are subsequently revealed by removal of endogenous eIF4E protein. In our case we selected a targeted protein degradation (dTAG)[41] approach that allows the rapid removal of functional eIF4E protein.

H1299 cells expressing eIF4E that was tagged with FKBP12[F36V] were established (Fig. 7a, b). We confirmed that both the C- and N-terminally tagged eIF4E-dTAG protein retained the ability to bind to eIF4G and were able to rescue MCL1 expression, a commonly used cellular biomarker of eIF4E activity[60] following siRNA knockdown of endogenous eIF4E (Supplementary Fig. 13a, b). Next, we selectively removed the endogenous eIF4E by CRISPR knockout and from single cell clones established clonal lines that expressed only the dTAG-regulated eIF4E (Fig. 7a, b). Four eIF4E-dTAG CRISPR clones were selected for further experiments, two with C-terminal tags and two with N-terminal tags: (Fig. 7b).

Treatment with the heterobifunctional dTAG[V]-1[64], that recruits VHL E3 ligase, for 6 h at ≈ 2–5x the IC$_{50}$ resulted in near complete loss of eIF4E-dTAG and a corresponding decrease in MCL1 protein for at least 72 h following dTAG[V]-1 treatment (Fig. 7b; Supplementary Fig. 14a, b). We also measured 4E-BP1 expression and found no evidence for altered 4E-BP1 expression (Supplementary Fig. 15) following loss of eIF4E, so did not include 4E-BP1 in subsequent experiments. Loss of MCL1 protein was not observed with the negative control diastereomer of dTAG[V]-1 (neg-dTAG[V]-1) that cannot recruit VHL[64] (Supplementary Fig. 14b). Treatment of the parent H1299 cells with dTAG[V]-1 had no effect on MCL1 or proliferation (Fig. 7b; Supplementary Fig. 14c). In contrast, treatment for 6 days with dTAG[V]-1, but not the neg-dTAG[V]-1 control compound inhibited growth of all 4 clones (Fig. 7c; Supplementary Fig. 14c). In addition, the eIF4E-dTAG protein in the C3-1 clone retained a similar target engagement profile as endogenous eIF4E in cells treated with compound **4** (Supplementary Fig. 13c; Fig. 5a, b).

The four eIF4E-dTAG CRISPR clones were also characterised by global proteome profiling comparing eIF4E-dTAG degradation with eIF4E siRNA knockdown in the parent H1299 cells. We confirmed that expression of the eIF4E-dTAG was similar to the expression of

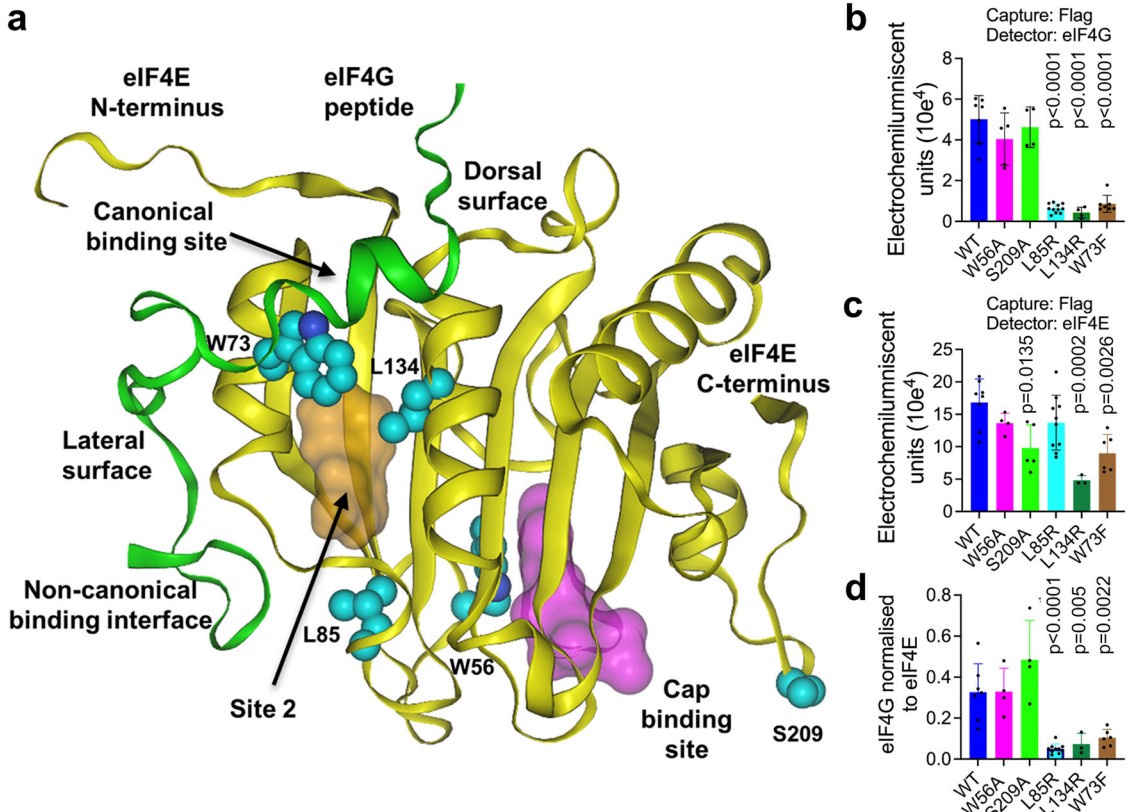

**Fig. 6 | Mutants of site 2 disrupt binding of eIF4G in H1299 cells. a** Crystal structure of eIF4E bound to a 35-residue fragment of eIF4G (PDB: 5T46) showing key features previously described in Figs. 1a and 2a. Highlighted in cyan are the residues selected for mutational analysis which gave suitable expression levels (W56, W73, L85, L134 and S209). **b** Quantification of eIF4E:eIF4G interaction determined by the electro-chemiluminescent assay in cells transfected with wild type (WT) eIF4E or a series of eIF4E mutants. **c** Quantification of N-terminal FLAG-tagged eIF4E expression in cells transfected with wild type (WT) eIF4E or a series of eIF4E mutants. **d** Quantification of eIF4G normalised to eIF4E (data from **b** and **c**) in cells transfected with wild type (WT) eIF4E or a series of eIF4E mutants. All plots (**b**–**d**) show mean ± SD from $n = 3$ biological replicates with significance for each mutant relative to the WT control using an ordinary one-way ANOVA with Tukey multiple comparisons test. Significant $p$-values ($p < 0.05$) are shown on the plots and the source data is located in the Source Data file.

endogenous eIF4E in the parent line (eIF4E-dTAG/parent eIF4E protein expression ratio of 0.8 to 1.7 – fold; Fig. 7d). There were also no major significant differences between the global proteome profiles of the control parent and untreated control dTAG models (Fig. 7e; Supplementary Fig. 16a). Compared to their controls, knockdown of eIF4E by siRNA treatment resulted in reduced eIF4E expression but not to the same extent as the eIF4E-dTAG models (Fig. 7d). dTAG$^V$-1 treatment resulted in a greater number of proteins with significantly altered expression ($p_{adj} < 0.05$) compared to the eIF4E siRNA knockdown (Supplementary Fig. 16b, c). Significant depletion of MCL1 ($p_{adj} < 0.05$) was only detected in the dTAG models which may reflect the more robust depletion of eIF4E protein compared to the siRNA conditions (Fig. 7d).

Overall, eIF4G binding, rescue from siRNA knockdown or CRISPR knockout and proteome profiling data confirmed that the eIF4E-dTAG could functionally replace the endogenous eIF4E and clearly showed the robustness of the dTAG-system to efficiently and rapidly remove the target protein.

**Targeting site 2 can interfere with eIF4E function**
We selected clone C3-1 for rescue experiments and expressed wild-type or each of the five mutants that we had evaluated for eIF4G binding described earlier (Fig. 6a; Supplementary Fig. 17a). We confirmed that both the wild type and mutant eIF4E proteins were expressed and that dTAG$^V$-1 treatment depleted eIF4E-dTAG but left expression of the rescue-eIF4E wildtype or mutants intact (Supplementary Fig. 17a, b). As expected, the expression of an un-degradable

wild-type eIF4E in clone C3-1 completely rescued the loss of MCL1 at 6 and 16 hr, and proliferation resulting from dTAG$^V$-1 treatment induced depletion of the eIF4E-dTAG (Fig. 8a–c; Supplementary Fig. 17). In contrast, expression of the W56A defective cap-binding mutant failed to recover MCL1 protein loss or inhibition of proliferation resulting from eIF4E-dTAG loss; this is consistent with a functional dependence on mRNA cap-binding (Fig. 8a–c; Supplementary Fig. 17). Expression of the MNK1/2 kinase substrate site mutant S209A was as effective as wild-type eIF4E in maintaining both proliferation and MCL1 protein expression, demonstrating that the phosphorylation of S209 is dispensable for eIF4E function at least in this cell line (Fig. 8a–c; Supplementary Fig. 17).

We then compared the eIF4E mutations that had affected the interaction with eIF4G in the cellular binding assay (W73F, L85R, L134R) (Fig. 6). Unexpectedly, we found the reference W73F mutation was able to maintain MCL1 expression and proliferation as effectively as the wild-type eIF4E (Fig. 8a–c; Supplementary Fig. 17b–d). We also found W73F was the only variant to exhibit increased expression following dTAG$^V$-1 treatment (Supplementary Fig. 17). The L85R mutant, which is predicted to perturb the site 2 flexible loop and potentially disrupts eIF4G binding, had a more subtle phenotype: as it rescued proliferation, but only partially rescued MCL1 protein expression after 6 hr dTAG$^V$-1 treatment and failed to rescue MCL1 at 16 h (Fig. 8a–c; Supplementary Fig. 17b–d). These contrasted with the L134R mutant located at the core of site 2, that is in the proximity of both the W73 key canonical PPI interaction and the flexible loop L81, that was unable to rescue proliferation and MCL1 protein expression consistent with

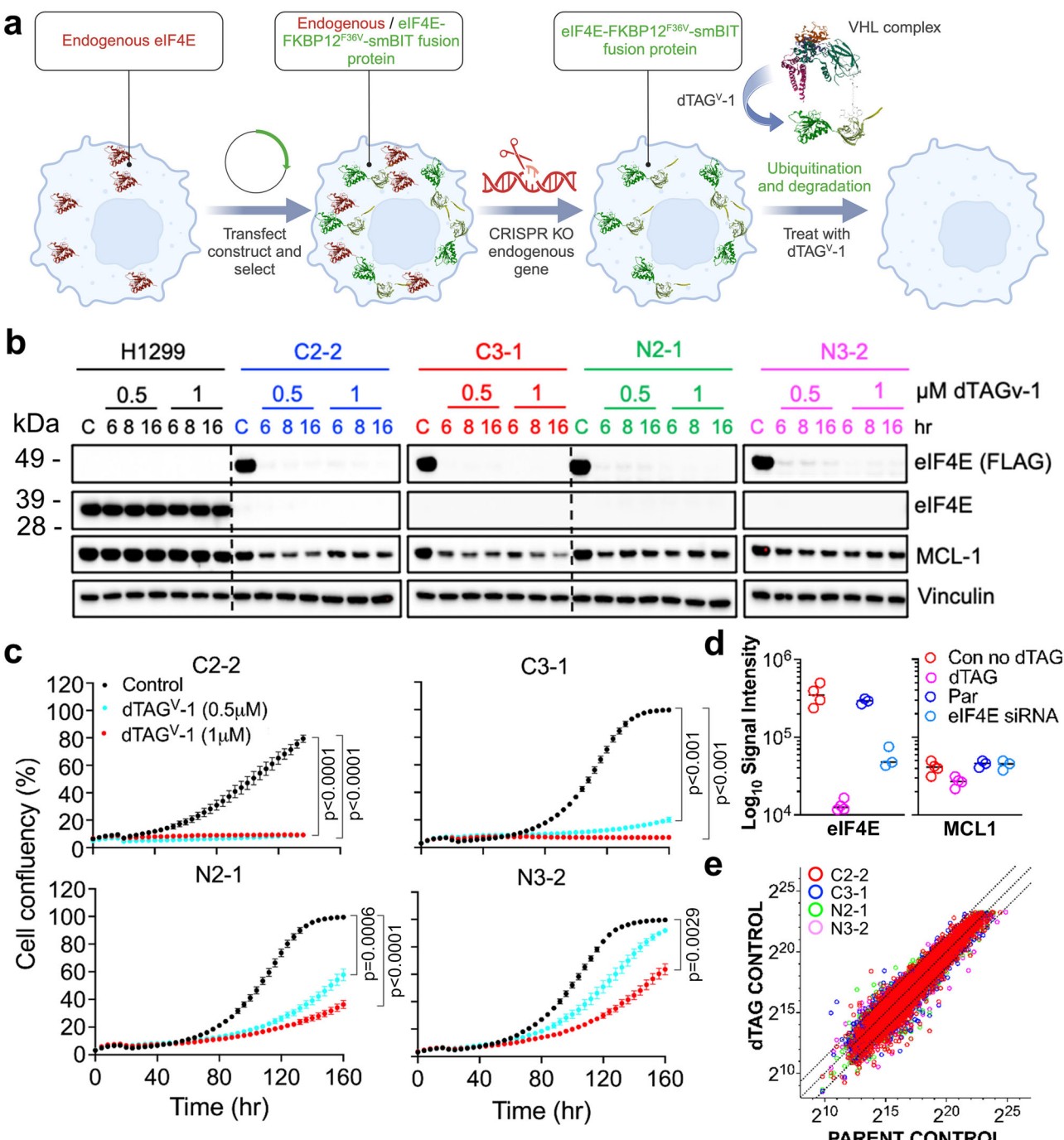

**Fig. 7 | Characterization of an eIF4E dTAG-degradation model. a** Schematic of the generation of stable dTAG eIF4E single clones in H1299 cells for eIF4E degradation (created in BioRender. Powers, M. (2024) https://BioRender.com/t36s392). A stable cell pool expressing eIF4E-dTAG was established, endogenous eIF4E was removed using CRISPR/Cas9, and single cell clones were isolated. Treatment of isolated clones with the heterobifunctional dTAG$^V$−1 molecule recruits E3 ligase VHL to degrade eIF4E-dTAG. Four eIF4E-dTAG CRISPR clones were selected: two C-terminal FLAG-tagged (C2-2, C3−1) and two N-terminal FLAG-tagged (N2−1, N3-2). **b** Immunoblot analysis of eIF4E, FLAG-tagged eIF4E, and MCL1 expression. Parental H1299 cells and the selected eIF4E-dTAG clones were treated with 0.5 and 1 μM dTAG$^V$−1 for 6, 8, or 16 h. Vinculin was used as a loading control ($n = 3$ biological replicates). **c** Real-time cell growth measurements of the selected eIF4E-dTAG clones treated with 0.5 or 1 μM dTAG$^V$−1. Cell confluency (%) was monitored every 4 h over 6 days using Incucyte Zoom. Significance was determined using a one-way ANOVA with Tukey multiple comparisons test comparing dTAG$^V$−1

treatment with the control in each cell line. Significant p values ($p < 0.05$) are shown (mean ± SEM from $n = 3$ biological replicates), the source data is located in the Source Data file. **d** Parent H1299 cells were treated with eIF4E siRNA (1 μg; eIF4E siRNA) or control siTOOLs (PAR) for 72 h ($n = 3$ biological replicates). The four eIF4E-dTAG clones were treated with vehicle (Con no dTAG) or 500 nM dTAG$^V$−1 (dTAG) for 72 h ($n = 1$ repeat from each individual clone) and global proteomes profiled. Significant differences between control and treated samples were determined using MSstats. Plots show signal intensity data for eIF4E (CON no dTAG v dTAG and eIF4E siRNA v PAR both $p_{adj} < 0.05$) and MCL1 ($p_{adj} > 0.05$; not significant). **e** Log$_2$[signal intensity] for proteome profiles of the individual eIF4E-dTAG clones with the control parent H1299 cells. Each point represents the ratio of individual protein expression between the control parent and different control dTAG clones (C2-2, C3-1, N2−1 and N3-3). Diagonal lines represent a 1.5-fold change in protein expression (Supplementary Data 5).

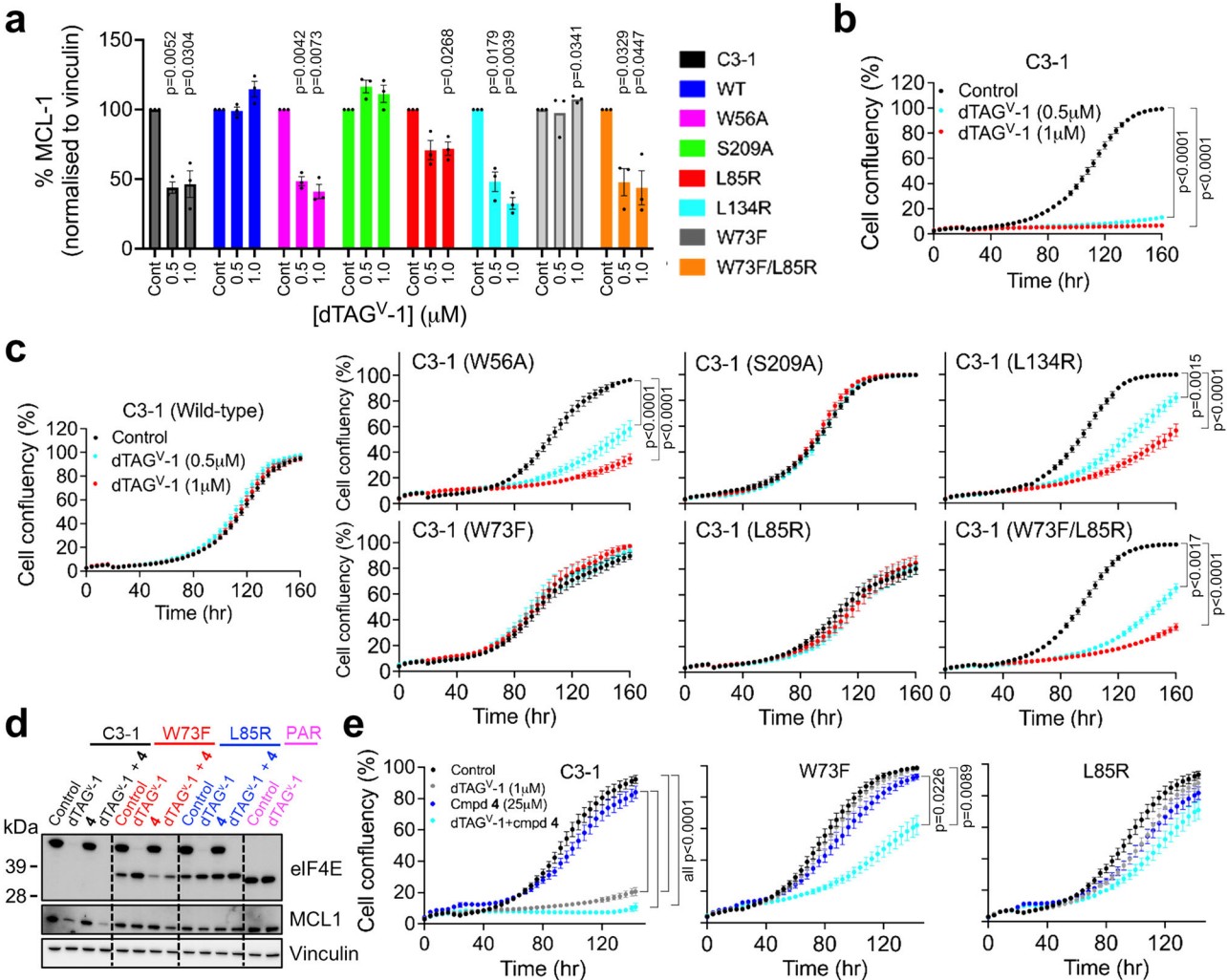

**Fig. 8 | Genetic rescue of eIF4E dTAG degradation through expression of wild type eIF4E or mutant of the different functional domains of eIF4E.** eIF4E dTAG C3-1 clone or C3−1 were transfected with wild-type eIF4E (WT) or a series of eIF4E mutants and treated with 0.5 or 1 μM dTAG$^V$-1. **a** Quantification of MCL1 immunoblot signal using Image J. Results are expressed as % MCL1 normalized to vinculin (mean ± SEM from $n$ = 3 biological replicates) following 6 hr dTAG$^V$−1 treatment (Supplementary Fig. 17b). One sample t and Wilcoxon tests (two-sided), using 100% as hypothetical control value, were performed and compared with the control in each cell line. **b, c** Real-time cell growth measurements monitored every 4 h over a 6-day period. Cell confluency (%) was calculated using Incucyte Zoom software and (mean ± SEM $n$ = 3 biological replicates). Ordinary one-way ANOVA with Tukey multiple comparisons test compared treatment with the control in each cell line.

Significant $p$-values (< 0.05) are shown on the plots. **d** Representative immunoblot of 3 biological replicates of eIF4E and MCL1 protein expression following 16 hr treatment of dTAG$^V$-1 (1 μM) or compound 4 (25 μM) or in combination in H1299 parental cells, the eIF4E dTAG C3-1 clone or C3-1 transfected with W73F or L85R mutant. Vinculin was used as loading control. H1299 (PAR) cell lines were treated with 1 μM dTAG$^V$-1 only as qualitative control. Vinculin was used as loading control. Quantification from MCL1 immunoblots normalised to control is shown in Supplementary Fig. 19c. **e** Real-time cell growth of the eIF4E dTAG C3-1 clone and C3-1 transfected with W73F or L85R mutant following treatment with 25 μM compound 4 ± 1 μM dTAG$^V$-1 were monitored and analysed as described for (**c**). Significant $p$-values (< 0.05) are shown on the plots and the source data is located in the Source Data file.

targeting of this site disrupting eIF4E function. Importantly, in these experiments the observed levels of variation in rescue construct expression were unlikely to account for differences in their ability or inability to rescue loss of the eIF4E-dTAG as for example, the L85R and L134R mutants were equally expressed in the dTAG-eIF4E model (Supplementary Fig. 17e, f), but showed opposing effects in the rescue experiments (Fig. 8a−c; Supplementary Fig. 17b−d).

The full retention of activity of the W73F mutant in the proliferation and MCL1 biomarker rescue assays and the full rescue of proliferation, but partial rescue of the MCL1 biomarker by the L85R mutant suggested that in intact cells the impact of these mutants was not sufficient to fully disrupt the eIF4E function. However, a double W73F/L85R mutant, which was expressed at the same level as the two single mutants, had a combinatorial effect on eIF4E function in cells (Fig. 8a−c; Supplementary Fig. 17b−d). In the rescue format, the double

mutant exhibited a loss of function similar to the W56A defective capbinding mutant with a more profound loss of MCL1 protein expression and cell proliferation than the W73F and L85R single mutations alone (Fig. 8a−c; Supplementary Fig. 17).

**Cooperativity between W73F mutant and site 2 ligand binding**
Finally, having demonstrated that compound 4 treatment could disrupt binding to eIF4G and cap-dependent protein synthesis in lysates (Fig. 4) and that mutations to site 2 could disrupt eIF4G binding in cells and eIF4E function in genetic rescue experiments (Figs. 6 and 8a−c) we tested the impact of compound 4 treatment on eIF4E activity in the rescue cell models.

Compound 4 induced a loop out conformation of the site 2 flexible loop, and the L85R mutation was also predicted to induce a similar conformation (Fig. 2). Since loss of eIF4E activity in the rescue

experiments was more pronounced when the L85R mutation was combined with the W73F canonical eIF4G/4E-BP1 binding site mutation, we tested whether treatment with compound **4** could also act in concert with the W73F mutation in a similar manner to L85R in the L85R/W73F double mutant (Fig. 8c).

Binding studies with the mutant eIF4E expressing cells showed that the active W73F and L85R mutants were denatured at a lower temperature than the WT, an effect that was even more pronounced in the inactive L134R and W73F/L85R mutants (Supplementary Fig. 18a, b). The W73F appeared to be destabilised, rather than stabilised, by compound **4** that may in part be due to the effects on its expression following treatment (Fig. 8d). Indeed, the destabilisation effect was specific to the active compound, but not the inactive compound **5** (Supplementary Fig. 18c). The L85R mutant showed evidence for compound **4** binding (Supplementary Fig. 18c). This was anticipated as switching the hydrophobic L85 to a larger, polar arginine side chain was predicted to cause steric and electrostatic clashes with F47, C89, Tyr91 and V156 and favour the loop out conformation required for compound **4** binding (Fig. 2; Supplementary Fig. 7a). Neither the inactive L134R that is predicted to disrupt site 2 nor the inactive W73F/L85R mutant that is predicted to disrupt both the canonical and non-canonical site showed evidence for compound **4** binding (Supplementary Fig. 18c).

We treated the C3-1 clone, W73F and L85R dTAG-rescue models with 25 μM compound **4** in combination with dTAG$^V$-1. Treatment with compound **4** in the absence of dTAG$^V$-1 treatment had no effect on the MCL1 biomarker or proliferation in any of the models tested, consistent with previously detected poor effects on protein synthesis in intact cells (Figs. 5e and 8d; Supplementary Fig. 10). Compound **4** and dTAG$^V$-1 co-treatment of C3-1 or the L85R rescue line gave no additional loss of proliferation or MCL1 expression (Fig. 8d, e; Supplementary Fig. 19a). The increased expression of W73F protein expression we previously detected following dTAG$^V$-1 treatment was reduced by co-treatment with **4** (Fig. 8d; Supplementary Figs. 17b, e and 19c). In addition, the combined treatment of compound **4** and dTAG$^V$-1 in the W73F mutant model resulted in a significant combinatorial effect on proliferation that was similar to that seen with the L85R/W73F double mutant (Figs. 7c and 8e). MCL1 protein expression was decreased on the immunoblot images, however, this decrease was not significant upon quantification (Fig. 8d; Supplementary Fig. 19c), which we concluded to be a reflection of the partial ($\approx$50%) sensitisation following the combined dTAG$^V$-1 and compound **4** co-treatment. Importantly no significant effects were seen with the neg-dTAG$^V$-1 and compound **4** (Supplementary Fig. 19a) or the less active compound **5** and dTAG$^V$-1 (Supplementary Fig. 19b).

## Discussion

Targeting translation initiation has long been considered a promising anticancer strategy and theoretically can be blocked at several different points[6,65]. Whether the different strategies currently being explored will exert similar effects on cap-dependent protein synthesis in cancer cells or maintain a therapeutic window for cancer cells over normal tissue, remains unclear, therefore it is still important that these different therapeutic approaches are explored. Targeting eIF4E is particularly attractive as it is the rate limiting factor for cap-dependent protein synthesis and is also a key signalling convergence point for two major oncogenic pathways[14–17,19–21,66,67]. Biochemical screening of compound libraries has identified tool molecules that have demonstrated proof-of-concept[22–24]. However, development of these tools into potent inhibitors with drug-like characteristics has so far proved unsuccessful.

Fragment-based screening is a powerful technique for probing the surface of proteins to identify potentially druggable binding sites[33,34]. However, the identification of binding sites brings with it the challenge of determining the biological relevance of the site. A fragment-based screening approach has not previously been reported for eIF4E. This in part may be attributed to the hydrophobic nature of eIF4E[42,43] which hampers production of large quantities of recombinant protein. We overcame this challenge by engineering eIF4E so that a problematic hydrophobic patch was masked by a short, tethered peptide based on the canonical binding sequence of 4E-BP1[53]. This system was a critical enabler for identifying fragment hits and the iterative structure-guided design cycles that generated a tight-binding tool compound together with a closely related, less active control[31]. This strategy was allied with cell-based mutational analysis, targeted protein knockdown and rescue experiments of eIF4E activity to explore the functional relevance of fragment binding pockets identified by the screen[41].

Our fragment screen, using a combination of X-ray crystallography and ligand observed NMR, identified several low affinity fragment hits. We detected the mRNA cap-binding site as one hotspot for fragment binding and also a second site with unknown functional relevance for eIF4E. We focused on the second site as multiple attempts to develop potent cell-active cap-binding antagonists has been hampered by the highly polar nature of this site[28–30]. Binding of our lead compound induced a conformational change to the α1-helix and flexible loop proximal to the non-canonical eIF4G/4E-BP binding site that could potentially impact on eIF4E interactors. Consistent with this idea we determined that the lead compound **4** could disrupt the eIF4E:eIF4G interaction and inhibited cap-dependent translation in lysates from multiple cell lines. We were able to show target engagement in intact cells with a CETSA EC$_{50}$ for eIF4E of a similar order of magnitude to the EC$_{50}$s determined for disrupting eIF4G:eIF4E binding in cell lysates. However, target engagement in cells did not lead to similarly potent effects on the eIF4G:eIF4E interaction or cap-dependent protein synthesis.

Recently Fischer and colleagues[48] reported a biphenyl analogue of 4EGI-1[22], i4EG-BIP binding in the same cavity as compound **4**, rather than the original site reported for 4EGI-1[22,48,68]. Although their study lacked a matched negative control compound and reported lower potencies (eIF4G peptide:eIF4E F$_P$ assay IC$_{50}$ = 67 μM[48]) than compound **4** (full-length eIF4G:eIF4E IC$_{50}$ ≈ 1–2 μM in lysates and intact cell engagement assays), its discovery nonetheless provides independent support for the functional relevance of site 2.

Displacement of eIF4G and recruitment of 4E-BP1 to eIF4E has been reported following incubation of lysates with 4EGI-1 and i4EG-BIP small molecules[22,48]. In contrast, despite validating the two binding assays with a 4E-BP-derived peptide, we consistently found no evidence for displacement or recruitment of 4E-BP1 following addition of compound **4** to cell lysates. This was unexpected as structural data suggests the conformational change induced by compound **4** binding to eIF4E should disrupt the non-canonical binding interface of both eIF4G and 4E-BPs. We speculate we are measuring different pools of eIF4G:eIF4E and 4E-BP1:eIF4E and the differential displacement of eIF4G, but not 4E-BP1, by **4** in cell lysates may result from disruption of the pool of weaker eIF4G:eIF4E interactions alone[51,69–71]. These observations suggest further in-depth studies to understand the complexities of 4E-BP1 or eIF4G binding to eIF4E and the impact of small molecules in lysates or intact cells.

We developed a dTAG eIF4E protein degradation model that will be used elsewhere to further explore eIF4E biology and mechanism in detail. Here were used this model to further explore the site 2 pocket. The positive control W73F mutation disrupted binding to eIF4G as expected, but unexpectedly retained activity similar to the wild type eIF4E in the rescue experiments. In addition to disrupting canonical binding of eIF4G and 4E-BP, W73 mutants are susceptible to proteasomal degradation thought to result from the loss of protective interactions with binding partners[60,62]. CETSA analysis showed that W73F was more susceptible to thermal denaturation but retained a site 2 capable of binding compound **4**. We speculate that in the co-immunoprecipitation binding experiments run in the presence of the

endogenous eIF4E, the weaker binding exogenous W73F mutant cannot compete with the endogenous eIF4E for eIF4G-binding, resulting in the assay reporting an absence of binding for W73F. The rescue experimental format is run in the absence of any competitive eIF4E following dTAG$^\text{v}$-1 treatment and leaves the weaker binding W73F mutant free to bind eIF4G and rescue function. This would also explain the increased levels of W73F protein following loss of eIF4E-dTAG that would result from W73F being protected from degradation through eIF4G binding[60,62].

We were able to use two mutants, L85R and L134R, to explore site 2 function. The L85R mutation was chosen for its position on the flexible loop and its proximity to several hydrophobic residues (F47, C89, Y91, V156). From our data we postulated that mutating L85 to a polar or bulkier residue would result in steric or electrostatic clashes and mimic the conformational change observed following compound **4** binding. The L85R mutant disrupted binding to eIF4G, consistent with compound **4** disrupting the eIF4E:eIF4G interaction. However, the L85R mutant had limited impact on eIF4E function, retaining the ability to rescue proliferation, but only partially rescuing the molecular readout. The activity of the L85R mutant had some resemblance to the compound **4** profile of poor activity in eIF4E:eIF4G assays and no activity in both cell proliferation and protein synthesis reporter assays. The second mutant, L134R, was selected due to its location at the core of site 2, and its proximity (7–8 Å) to W73 (key canonical PPI interaction) and L81 (flexible loop). We predicted that mutating L134 to a larger or more polar residue might mimic ligand binding. Interestingly the L134R mutant disrupted binding to eIF4G and lacked activity in functional rescue assays, possibly through long range effects on both the canonical and non-canonical interfaces. Together the impact that the mutant and lead compound have on eIF4G binding and eIF4E functional readouts suggests that site 2 may be required for eIF4E activity and that binding of a small molecule or disruption of site 2 by mutation can impact eIF4E function. Unlike the single mutants the W73F/L85R double mutant was unable to rescue loss of the eIF4E-dTAG protein. Similarly, co-treatment experiments demonstrated a combinatorial effect between compound **4** treatment of the rescue model and expression of the W73F mutant. Collectively, these data suggest that site 2 has a role in eIF4E function, but also that for site 2 binding by a small molecule to effectively inhibit eIF4E activity in cells the compound would need to have sufficient potency to disrupt both the canonical and non-canonical interaction with eIF4G.

Overall, our approach demonstrates the power of coupling fragment-based screening with target-degradation and genetic rescue approaches to find and explore functional pockets on difficult to drug proteins such as eIF4E. For one site we were able to use structure guided design to elaborate low-affinity fragments into a potent lead compound. Taken together our observations from the mutant and compound treatment data contribute to understanding some of the challenges associated with disrupting an extended PPI interface and suggest it may be necessary to disrupt both the canonical and non-canonical regions of eIF4E to drive a strong functional effect in cells. However, the discovery of compound **4** and the associated structural understanding, and the knowledge that the L134R site 2 mutation inactivates eIF4E provides hope that a significantly more potent small molecule inhibitor could drive a more profound cellular effect. Alternatively compound **4** could be used as a platform for the development of eIF4E degraders which may lead to improved cellular activity and a deeper understanding of eIF4E biology.

## Methods
### Cloning
The eIF4E X-ray crystallographic construct used for fragment screening (cloneID=2983, Supplementary Table 1a) starts with His6 tag, followed by 15 residues derived from 4E-BP1 (GARII YDRAF LMACR), then Gly4 linker and TEV tag before residues 36–217 (C-term) of eIF4E 127 N variant (Supplementary Table 1b). TEV tag in this protein was later found to be resistant to TEV protease. Human eIF4E was subcloned into a pET-30a vector and mutated to achieve desired protein sequence. The plasmid was transformed into BL21-CodonPlus (DE3)-RIPL *E. coli* cells for expression.

### Protein expression and purification of eIF4E
Cells were grown at 37 °C in Terrific Broth supplemented with 50 μg/ml kanamycin for 4 h and then the temperature was dropped to 18 °C. After 2 hr, the cells were induced with 1 mM IPTG and grown overnight. The harvested cells were suspended in lysis buffer at 4 °C, lysed by sonication and clarified by centrifugation. The protein was captured by immobilized metal ion affinity chromatography (HisTrap Fast Flow Crude; Cytiva) and purified by a gel-filtration column (Superdex 200, GE Healthcare). Fractions were analysed by SDS-PAGE (Novex), pooled and concentrated. Expected mass for the protein is 24972.4 Da and analysis by mass spectrophotometry showed single species with a purified protein mass of 24972.0 Da.

### Crystallisation and structure determination
The protein (cloneID=2983, Supplementary Table 1a) was concentrated to 8.0 mg/ml in protein buffer (25 mM HEPES/NaOH pH 7.5 [HEPES acid, NaOH used for adjusting pH], 150 mM NaCl, 5 mM DTT). Native crystals grew from sitting drops containing 0.5 μl protein and 0.5 μl seed solution in Greiner CrystalQick protein crystallisation plates. Protein crystal seeds were kept frozen at −80 °C and diluted before use around 10,000 times with reservoir solution (40% v/v PEG 400, 0.1 M Tris/HCl pH = 8.0, 2 mM DTT). Crystals appeared within a few hrs and reached their final size in four days. Although protein spontaneously crystallise in HEPES/NaOH pH=7.5 buffer, it does not nucleate in Tris/HCl pH = 8 without seeds. Seed dilution was optimised to give on average 1–2 crystals per drop. A soaking protocol was utilized to generate ligand-bound structures. Compound dissolved in DMSO solution was mixed with soaking stock solution to achieve a final composition of soaking solution: 25 mM compound, 2.5% v/v DMSO, 36 % v/v PEG400, 0.17 M HEPES 7.5, 2 mM DTT. 7 μl of soaking solution was added directly to the well containing crystal(s) and incubated overnight. Soaked crystals were flash-frozen the following day in liquid nitrogen for data collection. Diffraction data were collected at beamline ID29 at the ESRF (compounds **1** and **2**) or using *in house* Pilatus300/FRX system (compounds **3, 4** and **5**) and processed with autoProc. The structure was solved by molecular replacement in Phaser[72] using the protein component of 1IPB as a search model. Iterative rounds of model building using Coot[73] and AstexViewer and refinement using Buster[74] and Refmac[75] were employed to complete the model. Coordinates and geometric restraints for the ligands were created using CSDOPT, an in-house developed program. The crystals grew in the P2$_1$ space group with two molecules in the asymmetric unit. Ligand poses were identified using automated computational protocol comprised of: 1) limited 6D molecular replacement using CSEARCH, an in-house developed program, with reference apo model of protein, 2) structure refinement using Buster, 3) water placement, 4) another round of structure refinement using Buster, 5) calculation of electron density maps, 6) ligand placement using AutoSolve[76] Binding site was excluded from continuous solvent model during refinements. Ligands were observed in the binding sites of both chains. Several rounds of model rebuilding and refinement were used to complete the model.

### NMR fragment screen
NMR fragment screening was carried out on a Bruker 500 MHz Avance spectrometer equipped with a 5 mm TXI cryoprobe with z-axis gradients. All experiments were carried out in 2.5 mm capillaries at a

temperature of 4 °C. Cocktails of four fragments, each at a concentration of 500 μM were screened against 21 μM protein at pH 7.2 in 20 mM Tris, 20 mM phosphate, 100 mM KCl, 5 mM DTT. The buffer contained 15% $D_2O$ as lock solvent, 2% DMSO-d6 and 30 μM TSP for referencing. Cocktails were screened using a CPMG experiment with a 2 ms spin echo time repeated 300 times for a total relaxation period of 600 ms in the transverse plane and a 1 s recycle delay. Competition at the Cap-site (site 1) was probed by addition of 60 μM of m7-GTP and re-recording of the CPMG experiments. Data were acquired and processed in Topspin 2.1 and analysed using a combination of Topspin 2.1 and in house software. Signal intensities for each fragment were scaled to the reference intensity in the absence of eIF4E protein before and after competitor addition.

### Evaluating the evolutionary conservation of binding sites
eIF4E orthologs were identified by BLASTP searches (E value < 0.01) against SwissProt / TrEMBL protein sequences from the mammalian, vertebrate, and rodent databases[77]. A multiple-sequence alignment (MSA) of the top BLASTP hits from each species was carried out using MAFFT[78]. The global sequence identity of each ortholog was plotted against the sequence identity of each identified binding site (Supplementary Fig. 5) (cap site (site 1), site 2 and the canonical PPI site). To define the site, we selected the 20 protein residues closest to a representative ligand, for site 1 – m7-GTP, for site 2 – compound **4** and for the canonical PPI site – a 4E-BP1 derived peptide (from PDB structure 3U7X). Overall sequence and specific site identity for representative examples is included in tabulated form for comparison (Supplementary Fig. 4 and Supplementary data 3 and 4).

### Structural alignment
The structural alignment was restricted to the backbone atoms. RMSD values were calculated using the Protein Structure Alignment tool in Maestro, Schrödinger 2023-1 Suite [Schrödinger Release 2023-1: Maestro; Schrödinger, LLC, New York, NY, 2023].

### Isothermal titration calorimetry
eIF4E protein was purified in house for Isothermal titration calorimetry (ITC). ITC experiments were performed on either a MicroCal VP-ITC or a MicroCal Auto-ITC200 at 25 °C in a buffer comprising 50 mM HEPES, 100 mM NaCl, 1 mM TCEP and 5% DMSO at pH 7.5. Typical titrations on the MicroCal VP-ITC were performed using $17 \times 16.4$ μL injections with a 250 s spacing between each injection and with a stirring speed of 307 rpm. Typical titrations on the MicroCal Auto-ITC200 were performed using $12 \times 3$ μL injections with a 150 s spacing between each injection and with a stirring speed of 750 rpm. ITC data analyses were performed with Origin 7.0 software using the "One Set of Sites" model.

### Compound supply
Compound **1** was purchased from Sigma Aldrich, compound **2** was purchased from Acros Organics. Compounds **3**–**5** were synthesised in house as described in the Supplementary information.

### Cell culture and treatments
The human non-small cell lung cancer cell line, H1299, was obtained and authenticated by STR profiling from ATCC (Manassas, VA, USA, cat#CRL-5803, RRD:CVCL_0060). H1299 cells were maintained in RPMI (Fisher Scientific, Loughborough, UK, cat#21875-034) supplemented with 10% foetal bovine serum (PAN Biotech, Wimborne, UK, cat#P30-3702), nonessential amino acids (Fisher Scientific, cat#11140-035). The human colorectal adenocarcinoma cell line SW620 (Cat#87051203; RRID:CVCL_0547), and the human cervical epithelioid carcinoma cell line HeLa (Cat#93021013) were obtained from ECCAC (Salisbury, UK) and maintained in DMEM + Glutmax (Fisher Scientific, cat#61965-026) supplemented with 10% foetal bovine serum. All cell lines were cultured at 37 °C in 5% $CO_2$.

### Co-immunoprecipitation assay
Cells were harvested by trypsinisation and washed with PBS before lysis in MSD lysis buffer (150 mM NaCl, 20 mM Tris pH 7.5, 1 mM EDTA, 1 mM EGTA, 1% Triton X-100) containing protease inhibitors (Roche, complete mini tablet, cat# 11836153001) and phosphatase inhibitors (50 mM NaF and 100 mM $Na_3VO_4$). Lysate was placed on ice for 30 min and then cleared in a microfuge at 17,000 x g for 10 min at 4 °C. The protein concentration of the cleared lysate was determined by BCA assay (Pierce) and adjusted to 4 mg/ml with lysis buffer. For immunoprecipitation, 100 μl of lysate was diluted with 890 μl PBS and either 10 μl of compound or peptide (RIIY 4E-BP1 peptide (RIIYDRKFL-MECRNSPV, Peptide Protein Research, Hampshire, UK). Monoclonal anti-eIF4E antibody (R&D Systems, Cat# MAB3228, RRID:AB_2097694) was biotinylated using the Lightning-Link kit (R&D Systems Inc. Cat# 371-0010) according to the manufacturer instructions and 20 μg used to coat 1 ml streptavidin magnetic beads (Pierce Cat# 88817) for 1 hr at room temperature (RT) with rotatory shaking. The magnetic beads were then washed 3x with PBS using a magnet and resuspended in 1 ml PBS. 100 μl of lysate and 100 μl of beads were then incubated with shaking overnight at 4 °C. Samples were then washed 3x with 1 ml PBS, resuspended in 100 μl PBS and boiled for 10 min with 1x SDS-PAGE sample buffer containing 10 μl 1 mM DTT.

### Electro-chemiluminescent binding assay
Endogenous eIF4E protein-protein interactions with eIF4G or 4E-BP1 were measured by the electro-chemiluminescent assay (MesoScale Discovery). This assay uses plates with carbon electrodes and electrochemiluminescent labels (SULFO-TAGs) that are conjugated to detection antibodies. When an electric current is applied to the plate electrodes the SULFO-TAG labels will emit light that is measured to quantify analytes in the sample. Streptavidin-coated 96-well High Bind Streptavidin SECTOR plates with carbon electrodes (MSD, Rockville, US, cat#L15SB-2) were coated with 25 μl of 5 μg/ml biotinylated eIF4E antibody (R&D Systems Inc, Minneapolis, US, cat# MAB3328, RRID:AB_2097694) in PBS for 1 hr at RT with shaking. For measuring exogenously expressed FLAG-tagged-eIF4E, 96-well Multiarray SECTOR plates with carbon electrodes (MSD, cat#L15XB) were coated with 25 ul of 4 μg/ml anti-FLAG M2 (Sigma-aldrich, cat#F1804, RRID:AB_262044) in PBS for 1 hr at RT with shaking. The plates were washed with 3 ×150 μl/well MSD wash buffer (cat#R61AA-1; 0.2% Tween-20 in PBS), then blocked with 150 μl/well block buffer (MSD, cat# R93AA) for 1 h. The prepared plates were washed 4x in 150 μl in MSD wash buffer and incubated with 50 μg lysates from HeLa or SW620 cells or 7.5 μg lysates from cells transfected with FLAG-tagged eIF4E. Lysates were added to each well in a 25 μl final volume and incubated for 1 hr (for endogenous eIF4E) or 3 hr (for exogenous tagged-eIF4E) respectively at RT with shaking. The incubated plates were washed 4x in 150 μl in MSD wash buffer and were re-blocked by the addition of 25 μl of blocking buffer containing a 1:500 dilution of the secondary antibody and incubated for 1 hr at RT with rotational shaking. Secondary antibodies used were specific for eIF4E (rabbit anti-eIF4E, Cell Signalling, Cat#9742, RRID:AB_823488), eIF4G1 (rabbit anti-eIF4G, CST, Cat# 2469, RRID:AB_2096028) or 4E-BP1 (rabbit anti-4E-BP1, CST, Cat#9452, RRID:AB_331692). Plates were then washed again 4x in 150 μl in MSD wash buffer and 25 μl of a 1:50 dilution of Goat anti-rabbit Sulfo-Tag (MSD, Cat# R32AB-1) in MSD blocking buffer was added and the plates incubated for 1 h at RT with rotational shaking. After a final wash step, 150 μl Read Buffer T4X (MSD, Cat# R92-TC2) was added and the plate read using the MSD QuickPlex SQ 120 plate reader.

### In vitro translation assay
HeLa In vitro translation (IVT) assay kits were purchased from Thermo Fisher(cat#88882). Bicistronic reporter mRNA was prepared from using a CMV/T7-renilla luciferase-IRES(FDMV)-firefly luciferase

reporter vector as template DNA (supplied by Oxford Genetics Ltd.) using Message Machine T7 Ultra transcription Kit (Thermo Fisher, cat# AM1345) to generate in vitro transcribed mRNA. DNA template was removed with Turbo DNase from the transcription kit and the mRNA purified using RNeasy columns (Qiagen, cat#74204). RNA concentration was adjusted to 1 μg/μl using a Nanodrop. In vitro translation assays were then set up using a HeLa IVT kit according to the manufacturer's protocol. The reaction was analysed using Dual Glo luciferase reporter assays (Promega, cat#E2920). Compounds were diluted from 100x DMSO stocks 1:10 in PBS and 2.5 μl transferred to 96 well PCR plates (Starlab, cat#I1402-9700). 22.5 μl IVT reaction mix (1.5 mL HeLa lysate, 60 μl RNAsin (Promega, cat#N2115), 300 μl accessory proteins, 600 μl reaction buffer, 200 μl nuclease free water and 30 μl 1ug/ml mRNA) was added and incubated for 30 min at 30 °C (PCR alphacycler 4, PCRmax, Stone, Staffordshire, UK). The samples were placed on ice for 10 min, then 10 μl was removed and transferred to a white 96 well assay plate (Corning, cat#3917). 50 μl Dual Glo luciferase stop and glo reagent 1 (Promega, cat# E2920) was then added for 5 min at RT and luminescence (Firefly luciferase, Cap-independent translation) read using a topcount NXT microplate reader (Perkin Elmer). 50 μl of reagent 2 was then added and incubated for 5 min at RT before reading luminescence from Renilla luciferase (cap-dependent translation).

H1299 cells were transfected with a biscistronic vector using Lipofectamine 3000 transfection reagent (Fisher Scientific, cat#L3000008) for 24 hr and then treated with compounds for further 24 h. Luminescence was measured after 24 h of compound treatment using the Dual Glo luciferase reporter assays as above.

## Viral library protocol
H1299 cells were seeded in 40 ml RPMI (Fisher Scientific, cat#21875091) at $5 \times 10^6$ cells/ml in 17x T225 Flasks. After 24 hrs culture the media was replaced with 24 ml of RPMI supplemented with 10 mg/ml Protamine and one of 16 pools of a lentiviral shRNA whole Genome Library at 1.6e5 TU/ml were added to each flask (Dharmacon SMARTvector). After a further 24 h culture the virus-containing media was replaced with 40 ml RPMI supplemented with 4 μg/ml puromycin in all flasks including an un-transduced control flask. Following 4 days culture the virus-transduced cells were harvest from all flasks and pooled. Cells were re-seeded in 120 ml of RPMI at $5 \times 10^6$ cells/ml in a T-600 multilayer flask, the remaining cells were pelleted and stored at −80 °C as the reference control. Cultures were maintained for 14 cell-doublings (15 days) and were passaged every 5 days with the remaining cells collected pelleted and stored at −80 °C.

At the end of the experiment genomic DNA was extracted from the −80 °C stored pellets (QIAgen Blood and Cell Culture DNA Maxi Kit#13362) and DNA quantity and purity assessed using a Nanodrop1000 spectrophotometer. The shRNA constructs were amplified from the genomic DNA using Phusion Hot Start II DNA polymerase (Fisher Scientific, cat#F-549L) with SMARTvector forward and reverse indexing PCR primers (Dharmacon, cat#PRM7668). The PCR product was gel purified (Qiagen QIAquick Gel Extraction Kit no. 28704) and the concentration of the purified PCR amplicon was measured using the Qubit dsDNA HS Assay (Fisher Scientific, cat#Q57851) before analysis by next generation sequencing (Open BioSource)

## Generation of stable dTAG and eIF4E CRISPR clones
To generate eIF4E CRISPR clones, 2 sgRNA were used as a single guide for electroporation: sgRNA2 (GACTACAGTGATGATGTATG), sgRNA 3 (CTAGGAGGGTATACAAGGAA). Stable H1299 cells expressing dTAG C- or N-terminal FLAG were pelleted and washed 3 times with PBS. To prepare the cas9 and sgRNAs, in PCR RNA-free tube, cas9 10 μg/μl was diluted 1:10 with nuclease free water. 1 ul of diluted cas9 and 1 ul of

each sgRNA (1 μg/μl) were mixed and incubated for 30 min. For the electroporation Neon Transfection system (MPK10096), cells were resuspended with 10 μl of Buffer R. 10 μl of resuspended cells were aliquoted into each PCR tube containing pre-mixed sgRNA-Cas9 RNP. The electroporation is performed at 1200 V for 30 ms, 2 pulse. After electroporation cells were incubated at 37 °C. CRISPR knockout endogenous eIF4E clones from the C- or N-terminal FLAG transfected with sgRNA2 or sgRNA3 were selected.

## Gene constructs and gateway cloning
For cellular binding or rescue experiments, wild type and mutants eIF4E tagged at their N- or C-terminal with a FLAG-peptide (DYKDDDDK) sequence were synthesised and cloned into the entry clone pDONR221 containing a gene of interest flanked by attL sites (Fisher Scientific, cat#12536017) and then transferred to expression plasmids F560 or F3037 (Kindly provided by Dr Steve Hobbs, ICR, London). This vector is a bicistronic expression vector encoding the insert as the upstream open reading frame and puromycin (F560) or hygromycin (F3037) resistance open reading frame downstream of an EMCV IRES. This was performed by an LR gateway recombination reaction (100 ng entry clone, 100 ng/μl destination vector, 2 μl of LR Clonase II enzyme mix (Fisher Scientific, cat#11791020), and made up to 8 μl of TE buffer, pH 8.0). The reaction mix was incubated at 25 °C for 1 hr, and then ended by addition of 0.2 μg proteinase K and incubation at 37 °C for 10 min. The expression vector was transformed into DH5α library efficiency competent cells, plated on LB agar and clones selected for sequence verification (Fisher Scientific, cat#18263012). Plasmid DNA was extracted using the QIAGEN Plasmid Maxi Kit (Manchester, UK, cat#12123) following manufacturer's instructions. The resulting product was redissolved in 100 μl nuclease-free water.

The human eIF4E gene was synthesised with human FKBP12$^{V36F}$ (aa1-108), smBIT peptide for HiBit detection (VSGWRLFKKIS) and tagged at the N-terminal or the C-terminal and cloned into the Gateway™ entry plasmid pDONR221 vector (GeneArt, Thermo Fisher, USA). Inserts were then transferred using gateway cloning to expression plasmid F560.

Plasmids (2.5 μg) were transfected into cells using Lipofectamine 3000 transfection reagent (Fisher Scientific, cat#L3000008) in Opti-Mem (Fisher Scientific, cat#11058021) and left for 6 hr before adding fresh media for 48 hr. Depending on the selection encoded by the plasmid construct transfected cells were selected using 1 μg/ml puromycin (Fisher Scientific, cat#A1113802) or 400 μg/ml hygromycin (Fisher Scientific, cat#10687010).

## IncuCyte ZOOM live cell imaging system
Real time monitoring of cell growth was performed with the IncuCyte ZOOM live cell imaging system (Essen BioSciences, USA). Cells were seeded in 96 well plates (Corning, cat#353072) and treated on the next day. The resulting phase-contrast images were analysed using IncuCyte ZOOM 2018A software and results were expressed as percentage confluency of cells.

## Cell viability
Cell viability was measured by CellTiter-Blue® Cell Viability Assay. Cells were seeded in 96 well plates (Corning, cat#353072) and left overnight. Compounds were added at a range of concentrations and plates incubated at 37 °C for 96 hr. Cell titre blue reagent (Promega, Hampshire, UK, cat# G8081) was added (5 μl/well) and incubated for a further 2 h at 37 °C. Fluorescence ($579_{Ex}/584_{Em}$) was measured using a PHERAstar plate reader.

## Immunoblotting
After seeding overnight in 6 well plates (Corning, cat#353046), cells were treated with compounds for the indicated times and concentration, harvested and the cell pellet lysed on ice in cell lysis buffer (Cell

Signaling, cat#9803), supplemented with 1x Halt™ protease inhibitor cocktail (Fisher Scientific, cat#78429) and cOmplete™ protease inhibitor cocktail (Sigma-Aldrich, cat#11836153001) for 1 hr. Protein was quantified using BCA protein kit (Fisher Scientific, cat#23225). Western blot was performed using NuPAGE™ 4–12% Bis-Tris gels (Fisher Scientific, cat#NP0336BOX) and transferred using the Invitrogen iBLOT™ gel transfer device system (Fisher Scientific, Cat#IB21001). Blots were probed at 4 °C overnight for eIF4E (Cell Signaling, Cat#2067, RRID:AB_2097675), MCL1 (Cell Signaling, cat#39224, RRID:AB_2799149), monoclonal ANTI-FLAG® M2 (Sigma-Aldrich, cat#F1804, RRID:AB_262044), and vinculin (Cell Signaling, cat#13901, RRID:AB_2728768). All primary antibodies were diluted (1:1000) in 5% BSA (Sigma-Aldrich, cat#A7906) in 1x TBS and 0.1% Tween 20 (Sigma-Aldrich, cat#P1379), except for vinculin (1:2000) that was diluted in casein blocker (Sigma-Aldrich, cat#1610782). This was followed by incubation with the secondary HRP-conjugated antibodies (anti-mouse HRP, Bio-rad, Watford, UK, cat#1705047; anti-rabbit HRP, Sigma-Aldrich, cat#NA934) in casein for 1 h at RT. ECL HRP substrate (SuperSignal™ West Pico, Fisher Scientific, cat#34578) was then added to detect protein targets by chemiluminescence and images acquired using the ChemiDoc™ system (Bio-Rad). Quantitative densitometry analysis of protein bands was performed using Image J v1.53 software.

## CETSA assay in cells

Cells ($1.5 \times 10^6$ cells) were plated into 75 cm² flasks and allowed to attached overnight. After treatment with compound **4** or **5** for 6 h, they were harvested by trypsinisation and washed x2 in ice cold PBS. Cells were resuspended in 1 ml of cold PBS supplemented with protease and phosphatase inhibitors (Fisher Scientific, cat#1861281). Resuspended cells were divided into 100 µl aliquots and centrifuged at 300 x g for 3 min at 4 °C. Following centrifugation, 75µl of supernatant was removed and cells resuspended in the remaining volume. Cell suspensions were heated at 40.9, 43.7, 48.1, 52.9, 57.6, 62.4, 67.2, 71.9, 76.3 and 79.1 °C for 3 mins followed by cooling at 20 °C for 3 min[25]. Suspensions were freeze-thawed twice using liquid nitrogen. Soluble fractions were separated from cell debris by centrifuging at 20,000 x g for 30 min at 4 °C. Samples were then analysed by Western Blot.

## Proteome Profiling

Proteomes were profiled by LC-MS/MS using label-free data-independent acquisition (DIA; BGI address) following 72 hr exposure of the four H1299 eIF4E-dTAG clones (C2-2, C3-1, N2-1 and N3-2) to 0.5 µM dTAG^V-1 or H1299 parent cells transfected with 1µg eIF4E siRNA (siTOOLS, Planegg, Germany; cat# si-G100-1977) using Lipofectamine 3000 transfection reagent (Fisher Scientific, cat#L3000008). Cells ($1 \times 10^5$ cells/ml) were lysed using SDS-free lysis and the lysate made up to 1 mL. Disulphide bounds were disrupted by reduction and alkylation. DTT was added to sample with a final concentration of 10 mM and incubated at 37 °C for 30 min. Iodoacetamide was then added to sample with a final concentration of 55 mM and incubated in the dark at room temperature for 30 min. The protein mixture was then enriched through a solid phase (SPE) C18 column. The C18 column was activated with 1 ml methanol at a rate of 3 drops/second, followed by 1 ml 0.1% formic acid to condition the column. The diluted protein mixture was loaded onto the column at 1 drop/sec, the column was washed with 3 ml 0.1% formic acid and enriched proteins were eluted with 800 µl 75% acetonitrile at a rate of 0.5 drops per second and the eluate was freeze-dried. The freeze-dried proteins were redissolved in 20 µl 50 mM ammonium bicarbonate and quantified according to the manufacturer's instructions (Pierce Quantitative Fluorometric Peptide Assay cat# 23290). A small aliquot was removed for PAGE QC analysis on a 12% polyacrylamide gel. The remainder of the protein samples were digested with trypsin at a ratio of 1:20 for enzyme and proteins (w/w). The mixture was incubated for 14–16 h at 37 °C.

For DDA (Data Dependent Acquisition) library construction (Nano-LC-MS/MS) the dried peptide samples were reconstituted with mobile phase A (2% ACN, 0.1% FA), centrifuged at 20,000 g for 10 minutes, and the supernatant was taken for injection. Separation was carried out by a Thermo UltiMate 3000 UHPLC liquid chromatograph. The sample was first enriched in the trap column and desalted, and then entered a tandem self-packed C18 column (150 µm internal diameter, 1.8 µm column size, 35 cm column length), and separated at a flow rate of 500nL/min by the following effective gradient: 0 ~ 5 min, 5% mobile phase B (98% ACN, 0.1% FA); 5 ~ 120 minutes, mobile phase B linearly increased from 5% to 25%; 120 ~ 160 min, mobile phase B rose from 25% to 35%; 160 ~ 170 minutes, mobile phase B rose from 35% to 80%; 170 ~ 175 minutes, 80% mobile phase B; 175.5 ~ 180 min, 5% mobile phase B. The peptides separated by liquid phase chromatography were ionized by a nanoESI source and then passed to a tandem mass spectrometer Oritrap Exploris 480 (Thermo Fisher Scientific, San Jose, CA) for DDA mode detection. The main parameters were set: ion source voltage was set to 1.9 kV, MS1 mass spectrometer scanning range was 350 ~ 1650 m/z; resolution was set to 120,000; maximal injection time (MIT) 90 ms; MS/MS collision type HCD, collision energy NCE 30; MS/MS resolution 30,000, MIT was auto mode, dynamic exclusion duration 120 seconds. The start m/z for MS/MS was fixed to auto mode. Precursor for MS/MS scan satisfied: charge range 2+ to 6 + , top 30 precursors with intensity over 2E4. AGC was: MS 300%, MS/MS 100%.

For DIA quantification (Nano-LC-MS/MS) dried peptide samples were treated as described in the previous paragraph but separated by the C18 column at a flow rate of 500nL/min by the following effective gradient: 0 ~ 5 min, 5% mobile phase B (98% ACN, 0.1% FA); 5 ~ 90 min, mobile phase B linearly increased from 5% to 25%; 90 ~ 100 min, mobile phase B rose from 25% to 35%; 100 ~ 108 min, mobile phase B rose from 35% to 80%; 108 ~ 113 min, 80% mobile phase B; 113.5 ~ 120 min 5% mobile phase.

Peptides separated by liquid phase chromatography were ionized by a nanoESI source and then passed to an Orbitrap Exploris480 (Thermo Fisher Scientific, San Jose,CA) to acquire mass spectrometry data in DIA mode. The main parameters were set: ion source voltage was set to 1.9 kV, MS1 mass spectrometer scanning range was 400 ~ 1250 m/z; resolution was set to 120,000; maximal injection time (MIT) 90 ms; 400 ~ 1250 m/z was equally divided to 50 continuous windows MS/MS scan. MS/MS collision type HCD, collision energy NCE 30; MIT was auto mode. Fragment ions were scanned in Orbitrap, MS/MS resolution 30,000. AGC was: MS 300%, MS/MS 1000%.

DIA data was analysed using the iRT peptides for retention time calibration. Then, based on the target-decoy model applicable to SWATH-MS, false positive control was performed with FDR 1% to obtain significant quantitative results. MSstats was used for the statistical evaluation of significant differences in proteins or peptides from different samples. The core algorithm is linear mixed effect model that pre-processes the data according to the predefined comparison group, and then performed the significance test based on the model. Thereafter, differential protein screening was performed based on the fold change >2 and p-value < 0.05 as the criterion for the significant difference.

## Statistical analysis

Significances were analysed using GraphPad Prism 10 software. All histograms represent quantitative data expressed as mean ± standard deviation or error on $n \geq 3$ biological replicates unless otherwise indicated. Unpaired t-test was used for the comparisons between two different conditions. When comparing one sample to a hypothetical value as control, a one-sample t-test was used. Experiments comparing three or more groups, were analysed using the ordinary one-way ANOVA with Tukey's test for multiple comparisons[79].

## Reporting summary

Further information on research design is available in the Nature Portfolio Reporting Summary linked to this article.

## Data availability

Source data are provided as a Source Data file. Atomic coordinates and structure factors of the protein-ligand complexes generated in this study have been deposited in the Protein Data Bank (PDB) under accession codes 8QM4, 8QM5, 8QM6, 8QM7, 8QM8, 8QM9. The mass spectrometry proteomics data have been deposited to the ProteomeXchange Consortium via the PRIDE partner repository with the dataset identifier PXD057122. PDB codes of previously published structures used in this study are 5T46, 4UED and 3U7X. Source data are provided with this paper.

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

## Acknowledgements

This work was supported by both Astex Pharmaceuticals, CR UK (grant number C309/A11566; PAC) and ICR core funding (PAC). MM was supported by the Astex Sustaining Innovation Post-Doctoral program.

## Author contributions

S.Y.S.: Investigation, Methodology, Writing – Original Draft, Review & Editing. S.D.A.: Investigation, Methodology, Writing – Revision Editing. M.M., C.I.M., E.C., B.D.C., M.G.C., J.C., C.E.E., S.D.H., C.M.F., P.N.M., N.P., P.P., J.S.D., K.S., M.V., H.W., Ge.W., G.l.W.: Investigation, Methodology. MVP: Visualization, Writing – Review & Editing. S.M.S.: Conceptualization, Investigation, Methodology. C.J.R., A.J.W.: Conceptualization, Investigation, Methodology, Supervision, Writing – Original Draft,

Review & Editing. P.A.C.: Conceptualization, Supervision, Writing – Original Draft, Review & Editing.

## Competing interests

S.Y.S., S.D.A., M.M., C.I.M., K.S., M.V.P., and P.A.C. are current or previous employees of The Institute of Cancer Research, which has a commercial interest in a range of drug targets and operates a Rewards to Discoverers scheme, through which employees may receive financial benefits following the commercial licensing of a project. A.J.W., C.J.R., M.G.C., B.D.C., C.E.E., E.C., J.C., S.D.H., C.M.F., P.N.M., N.P., P.P., S.M.S., J.S.D., M.V., Ge.W., H.W., and G.l.W. are current or previous employees of Astex Pharmaceuticals which has a commercial interest in a range of drug targets.
