## [Transparent Peer Review file · Nature Communications]

Integrating fragment-based screening with targeted protein degradation and genetic rescue to explore eIF4E function

Corresponding Author: Dr Paul Clarke

Version 0:

Reviewer comments:

Reviewer #1

(Remarks to the Author)

The manuscript by Sharp et al. describes the usage of fragment-based screening to identify a novel site on eIF4E that disrupts binding to eIF4G. eIF4E and eIF4G form a protein-protein interaction which stimulates cap-dependent translation, a process that becomes overactivated in many human cancers. eIF4E is a widely explored target, particularly since the discovery of the first eIF4E-eIF4G PPI inhibitor, 4EGI-1. This work reports a new scaffold that binds to a novel site outside of that of 4EGI-1 and the canonical eIF4E protein ligands, eIF4G and 4EBP1. In addition, the work also describes genetic characterization of this new binding site through dTAG-mediated genetic manipulation and rescue experiments. Unfortunately, while the binding site data translated to cellular models, compound activity was quite poor which the authors hypothesize is due to inadequate disruption of eIF4G binding. These finds, although not spectacular in the later regard, are of value to the larger field. I will provide some additional comments for improving the manuscript to enhance its rigor and novelty for publication in Nat. Commun.

1. With respect to the protein expression, eIF4E is not necessarily unstable as a monomer, just when unbound to m7G/capped mRNA; this should be updated in the text.
2. I am curious to know a bit more why the 4EBP1 fragment was chosen to fuse to eIF4E since there are many crystal structures of eIF4E absent the disordered N-terminus. Would this fusion bias the results of the screen?
3. m7GxP was not used to stabilize the cap-binding site in the protein prep. What was the rationale for this? Inclusion is standard in the field unless you are looking for cap-competitive inhibitors or studying cap analogues.
4. Full-length (non-phospho) 4E-BP1 and eIF4G should both compete for binding to the second site identified. How do the authors explain the specificity observed for the eIF4G PPI?
5. Even in lysate, the compound is much less active than the Kd would predict. The authors should perform CETSA or some other target engagement assay to demonstrate binding in this assay. Maybe it binds to many off-targets which decreases activity? Have the authors tried to express the proteins and perform biochemical studies? The authors should also consider a traditional cap pulldown assay as another standard assay to look at PPI integrity.
6. The data in Figure 4E for compound 4 shows no dose-dependence at the higher concentrations. Based on the data presented, I am not sure how confident that one can be that this activity is on-mechanism.
7. On a positive note, the dTAG studies nicely demonstrate the complexity of regulation of cap-dependent translation initiation. 4EGI-1 has not been shown to directly bind to eIF4E in cells, and the authors are correct, that many other liabilities exist with other reported compounds.

Reviewer #2

(Remarks to the Author)

The work presented in the manuscript by Sharp and colleagues explores drugability and function of eIF4E through alternative, underexamined routes addressing canonical and non-canonical binding sites of eIF4E. For this they integrate fragment-based screening, structural biology and functional assays utilising a novel designed eIF4E variant which allows to survey fragment/compound binding to previously inaccessible pockets of eIF4E due to protein stability issues. Through their approach, they identify a tool compound binding to a pocket of eIF4E involved in non-canonical binding to eIF4G and 4E-BP1. As a results of compound binding, the interaction between eIF4E-eIF4G in cell lysates is selectively disrupted leaving

the 4E-BP1 association to eIF4E intact. In vitro reporter translation assay reveal that the new compound specifically inhibits cap-dependent translation. Based on their structural data they identify key residues of eIF4E involved in eIF4G binding via the non-canonical binding, which they validate by testing eIF4G-binding to derived variants of eIF4E in cell lysates. To then investigate the role of the non-canonical eIF4G-binding site in cells they develop and establish a fast-response, targeted eIF4E degradation, knock-in cell system to overcome limitation of cell viability to due eIF4E loss by conventional strategies. They validate the eIF4E-degradation system by examining the rescue potential of various eIF4E variants that are overexpressed simultaneously to the induced degradation of endogenous eIF4E. With this they discover that some of the structure-guided eIF4E variants mutated in the non-canonical eIF4G binding site cause cell growth defect and MCL-1 repression, as a reference for eIF4E-dependent translation. Finally, they investigate the effect of their tool compound in the engineered cells. While the compound disrupts the eIF4G-eIF4E interaction in lysates of these cells, the compound lacks potency when directly administered to the cells. As a result the compound itself does not impact cell growth. However the compound shows some synergy with a variant of eIF4E that is partially disrupted in its canonical eIF4G binding site. The authors conclude their works that full potential of disrupting the non-canonical eIF4G binding site is only accessed in combination with interference at the canonical eIF4G binding site. They further suggest that their tool compounds could be used for development of dual-targeting eIF4E for degradation.

The presented work explores two major directions: defining the biological impact of the non-canonical eIF4G interaction site and determination of the drugability of that site. The authors present a series set of experiments using in most cases reasonable controls supporting their conclusions. The overall presentation is generally compelling, interesting and has good potential to provide new directions and tools in targeting eIF4E and uncover more about it's biology. However, some key observations and conclusions are based on only one type of experiment (the luminescence assay) and require more thorough investigation before it is suitable for publication. I have three major points that I believe need to be addressed: The biological role of site 2:

a) There is an apparent, broad functional disconnect between the effect of the generated eIF4E variants in site 2 on eIF4G binding after overexpression (Fig 56-c) and their impact on eIF4E function (Fig 7). Experiments from Fig 5 were suggesting different protein stabilities of the mutant 4Es. Is that somewhat compromising interpretation of the capability of these mutants to rescue or not rescue the induced 4E loss in Fig 7? They state in lines 364-367 that blots were suggesting similar protein levels but blots seem to stem from different membranes and second the blots wouldn't show turnover of the protein as the mutants might not be able to form their native complexes due to stability issues as pointed out in Fig 5. Particularly L134R. Furthermore they state L85R had a partial effect. How do they explain the disconnect and the reasoning for the double mutation, what's the stability of the double mutant?

b) In my opinion the generated dTAG-eIF4E cell line is insufficiently characterised in regards to its effect on eIF4E function. The authors stem their most important experiments investigating the biological relevance of site 2 and their drug on this cell line. However, they only control their system with the cell lysate assay that seems to be disconnected from the in cell behaviour, and assess eIF4E functionality through following expression of one protein MCL1. In addition it appears that tagged-4E expression levels might be different as compared to parental wildtype levels, caused by the different stabilities they observed in Fig 5. The functionality of the engineered eIF4E in this cell line should be addressed via

i) at least one global technique like RNA-seq or proteomics comparing parental and engineering lines. MCL1 has a short half life, what about long lived 4E targets?

ii) m7G pulldowns and controlling integrity of eIF4F complex formation

iii) providing a time course induction of eIF4E degradation and kick-off of cell effects in the presence and absence of overexpression of their variants. At the moment only very long time courses are given.

The compound activity: The authors have identified a tool compound with binding capacity to eIF4E site 2 and activity of this compound when added to cell lysate. However, the compound seems to lack activity when administered directly to cells. While this seems not uncommon for early stage in compound development, it should be more addressed experimentally why this disconnect establishes in order to support suitability of the compound for further development as suggested by the authors. I suggest the following points:

a) Does the compound bind to eIF4E in cells? What is the binding constant in cells? Potential off target or serum binding?

b) Binding constant of compound to the different engineered versions of eIF4E used in the cell lines?

c) Metabolic turnover of the compound?

d) They generated variants of eIF4E disrupted in site 2 but did not test if these are resistant to compound binding

e) They only show growth curves and the luminescence assay as a read out. Is MCL1 (or other eIF4E targets) affected in their expression in Fig 7d and 8d?

Insufficient labels of figures and description of methods: Throughout the manuscript there are many instances in which figures legends and methods are insufficiently described or completely in the wrong place, which should be address to allow better understanding of how experiments were performed

a) Fig 2 and Supplementary Fig 1 and 2: Unclear if screening was done in presence or absence of m7G

b) Crystal structures shown with m7GTP in the structure but its unclear if co crystallised/soaked with compound and m7 ? not stated in methods

c) Is it possible to describe more details of what the Astex Pyramid platform is and some basics of the screening.

d) Cross species sequence conservation analysis: How was it done and which species does it include? No information given. What are the data points in the graphs of supplementary figure 2? Which structures have been used in d e f? If it's the ones from the main text, this is all human structures. What is the general background conservation? Seems generally very high so that eIF4E and site 2 is generally just highly conserved in general?

e) Supplementary Fig 4 legend for b c is incorrect

f) Fig 4c it is unclear from methods and description if data for flag-4E or endogenous 4E interactions is shown.

g) Various figures lack units on the y-axis or explanation to what the data was normalised.

h) Suppl8: What means "WT Cterm" or "WT Nterm"? I find the description confusing

Minor:

- a) No statistical methods are used to support statements made in the manuscript (except Fig 5b). It could help to strengthen arguments
- b) Line 99: More information on “drug like” could be helpful: what features are you referring to would good to improve
- c) Do they have kinetics of binding of compound 4, slow fast binder?
- d) Compound 4 seems to target only 4G-4E but not 4E-BP1-4E, but also does not facilitate binding of 4EBP1 to freed up binding site. Have they done binding of compound 4 to eIF4E in presence of 4G/4E-BP1?
- e) Fig4a: It is stated compound 5 doesn't have an effect but it is not quantified. It appears that 5 has an effect. I suggest quantification or adjustment of the strength of the statement.
- f) Most proliferative cells will show dependency on eIF4F as they are dependent on cap-translation (Fig5). Do you have perhaps any comparison? Otherwise I find this piece of data relatively weak to make a case for a specific eIF4E-dependency of the chosen cell line.
- g) Line 308: While I agree that its plausible that 4E knockout is lethal, I disagree that supplementary figure 5 is necessarily proving this point (see above). Another piece of data would be great to support this argument or references.
- h) Fig 5b does it take the lower expression into account? In my opinion this should be normalised to 4E variant expression levels to reveal the fraction 4G bound? While I agree with L85R, L134R might not actually a strong candidate. Also in regards to the disconnect between the assay this could help.
- i) Supplementary Fig 8: How do they explain that the apparent readout is a lot lower than for example 5b. Do they have the expression data alongside?
- j) What's the IC50 or at what IC is the dTAG-V-1 in the experiments?
- k) It would be great to show 4EBP1 levels in blots of Fig6-8

Reviewer #3

(Remarks to the Author)

Summary of results described:

Sharp et al, present a new approach for discovery of small molecule inhibitors of the eIF4E/eIF4G interaction in order to find new compounds as anti-cancer agents. This interaction is naturally regulated by the 4E-BPs and their phosphorylation. Targeting the eIF4E/eIF4G This interaction has been of interest for many years, and a small molecule inhibitor, 4EGI-1 was described already in 2007. Despite many efforts, the exact mechanism of inhibition has not yet well understood, and there were several potential binding sites discussed.

Here the authors use a new approach by replacing the first 35 residues of eIF4E with the canonical binding sequence of 4E-BP1 attached with a flexible linker to L36. This resulted in a stable construct that could be overexpressed, a crystal structure could be solved with the linked 4E-BP1 fragment bound. The construct was also well suited for fragment-based screening, both searching for ligands in the cap site and the hydrophobic dorsal face.

Using this achievement, the authors applied a structure based search for inhibitors, using the Astex Pyramid compound platform, NMR and crystallography. They succeeded to get tight compound, the best was compound 4 with a Kd of 15 nM. This is great progress, and the authors should be complemented. However, the effect the improved inhibitor has on cancer cells is not so clear.

Subsequently, the authors applied a quantitative electro-chemiluminescent co-immunoprecipitation binding assay. They show that compound 4 displaces eIF4E:eIF4G but not eIF4E:4EBP1.

Mutants relevant to compound 4 binding yielded consistent results in binding and functional assays.

Concerns, suggestions and comments:

1. The authors mention numerous residues and mutations. However, the locations in the protein structure are difficult to follow. This would be much easier to follow if they included in the text or the supplement a figure with a sequence alignment of their construct with the wild-type. This should also include an alignment with the 4E-BP1 segment they used and those in the Sekiyama et al. and one of the Gruner papers. The data are given in Supplementary tables 1 and 2; however a figure would help and should be included.

It appears that the authors were not aware of the earlier structure for the eIF4E/4E-BP complex with a 40 residue 4E-BP1 segment (residues 44-84) published in 2015 by Sekiyama et al. (E4036–E4045 | PNAS | Published online July 13, 2015). That paper also showed that the C-terminal loop of the 4E-BP1 peptide is needed for activity in a dual luciferase assay. The Sekiyama structure of the complex is significantly different for the binding interface of the fused construct shown in Fig. 2a. The authors need to discuss this.

Fig. 1 shows an overlay of the crystal structures of eIF4E with the eIF4G segment bound in the Gruner structure from 2016. It would be helpful to also overlay the structure of the eIF4E complex with a 40 residue 4E-BP1 segment (residues 44-84) published in 2015 by Sekiyama et al. That paper also showed that the C-terminal loop of the 4E-BP1 peptide is needed for activity in a dual luciferase assay.

Lines: 454-458:

This section is not clear and must be changed to: Previously, Fischer and colleagues 64 described a biphenyl analogue of the original 4EGI-1 compound converted in a PROTAC tool. The biphenyl end binds to the same cavity as compound 4, which is adjacent to the original 4EGI-1 binding site 28, 51,64.

Lines 230-235:

The Authors suggest that the conformational change might explain the impact on 4G binding and cite Figure 3c. It was difficult for me to see the conformational change in 3c, I assume it is the yellow helix. It might be easier to show an overlay of the original and the changed conformation. It seems that the c-term residue in their 4G peptide is causing the clash. This clash probably does not happen in the full-length 4G protein.

Recommendation:

This is an important manuscript describing the discovery of highly efficient inhibitor of translation initiation following up of previous earlier attempts. The paper is suitable for publication in Nature Communications after several modifications, in particular those listed below.

1. The binding of a bi-phenyl compound to the dorsal hydrophobic site of eIF4E has already been described recently in an attempt to use it for defining a degrader (Fischer et al., Eur.J. Medicinal Chemistry, 2019 (2021) 113435. The binding cavity is essentially the same as the one described here.
2. The crystal structure of the complex of eIF4E with an ~80 residue fragment of 4E-BP1 was first presented at a Cold Spring Harbor Symposium in 2014 and subsequently by Sekiyama et al. (E4036–E4045 | PNAS | Published online July 13, 2015) and must be cited.
3. The electro-chemiluminescent co-immunoprecipitation binding assay is very poorly described and should be improved to make it better understandable.

Reviewer #4

(Remarks to the Author)

The authors present the development of an inhibitor of eIF4E using a fragment-based approach. 4E is the mRNA cap binding protein in the eIF4F complex, which regulates the rate-limiting step of mRNA translation initiation. Because of its critical role in translation, 4E is important in cancer cell transformation and is therefore a target for developing anti-cancer drugs. Fragment-based screening is less biased than HTS and this, combined with several rounds of structure-based modifications led to the identification of a nM binding compound that disrupted the 4E:4G interaction, but not the 4E:4E-BP interaction, and inhibited cap-dependent translation in cell lysates. Structural analysis showed that the compound bound to a hydrophobic pocket away from the cap-binding site and near the non-canonical binding region of 4G where it causes the extension of helix a1 and a loop conformational change, which is enough to disrupt the 4G peptide interaction site. The activity of the compound was tested in intact cells via a genetic rescue approach that showed the compound could weakly inhibit 4G binding (IC50 ~ 100 uM) but lacked the ability to inhibit protein synthesis. Mutational analysis of the compound binding site did not provide significant insight on whether it has an important functional role.

Overall, this is an interesting and well-written paper which uses an array of experimental approaches to arrive at a potential inhibitor of an otherwise difficult to target protein. Particularly impressive were the genetic rescue experiments and the expression of a soluble form of human 4E by blocking the exposed hydrophobic patches with its natural ligand. However, the final results were not compelling as the compound was not very effective in intact cells and the analysis on its binding site did not reveal an interesting functional role. The biggest issue with the paper is that it seems to be presented as if the compound 4 binding site is novel and not yet known, when another compound (similar at least with regards to the core bi-phenyl group), i4EG-BiP, was already found to bind to the identical site and its similar conformational consequences documented. Although the i4EG-BiP study was mentioned in passing in the discussion, it was published over two years ago and therefore should be part of the introduction for what is already known regarding 4E, namely, that the present study identified a binding pocket which was previously identified by another study. That being said, this paper does highlight the power of the fragment-based approach for identifying binding pockets that HTS may miss.

Other points to address:

1. Based on the structural analyses to date, 4G and 4E-BP use very similar binding modes and sites on 4E. There was no satisfactory explanation given as to why compound 4 affects 4G binding but not 4E-BP.
2. Given the significant difference in activity of compound 4 in cell lysates vs. intact cells, can the authors provide an explanation for this disparity? This is important as it could also explain why compound 4 had limited functional effects. How were permeability and efflux scores calculated?
3. There are at least two mutations (W73F and H85R) that have been shown to disrupt eIF4G binding to eIF4E, but not the functionality of the eIF4F complex. Why is this the case?
4. Where are the X-ray data collection statistics? It would be useful to mention the resolution of your structures in the figure

legends. Also, please show discovery maps of the compound fragment structures.

5. Only 3 hits (including the one in Supp. Fig. 1b) from the 1371 fragment library are shown, were there others at site 2?
6. Line 140-142: "The X-ray structure of this engineered eIF4E matched X-ray crystal structures of wild-type unmodified eIF4E" - Indicate the RMSD between the structures.
7. L217 show the ITC profile for compound 4 in Supplementary.
8. Fig. 4a, why is there a second band in the eIF4G row input control pulldown?
9. Lines 328-337: As the authors further proceed with clone C3-1, Fig. 6 can be moved to the supplementary figures, and the explanations here can be shortened.
10. It would be easier to follow the hit optimization if the 2D chemical structures were shown in Figure 2.
11. Fig 2b, the label for N127 spreads into a second line.
12. Supplementary Fig. 2: structure figures are very difficult to discern with black background and dark blue colors.
13. Supplementary Fig. 3a: The purple mesh should be brightened; what type of map is the mesh representing?
14. Supplementary Fig. 4, L55: delete the letter C.
15. L445 "We focused on a second site as multiple" change 'a' to 'the'
16. L246, use a better synonym for "engenders".
17. L576, "20 mM phosphate" is mentioned twice.

Version 1:

Reviewer comments:

Reviewer #1

(Remarks to the Author)

Upon review of the revised manuscript, it is evident that the authors considered the previous critiques adequately leading to a greatly improved manuscript. In my opinion, the manuscript is now ready for publication. This is poised to be a notable contribution to the eIF4E and fragment-based drug discovery literature.

Reviewer #2

(Remarks to the Author)

I acknowledge the effort and time Sharp and colleagues invested to address all my and the other reviewers comments. They have satisfyingly answered my questions, included new data and corrected/revised text accordingly, which has strongly improved understanding of the data and supports their conclusion well. I am happy to recommend the manuscript for publication.

Reviewer #4

(Remarks to the Author)

The manuscript is significantly improved and easier to follow now. All the points were appropriately addressed.

Final minor points:

1. Supplementary Fig. 3, indicate the sigma levels at which the difference maps have been contoured.
2. Supplementary Fig. 7c, there appears to be a faint purple mesh around some of the residues, either remove it or brighten it and explain what it is.
3. pg. 8, L199: "elaborate" doesn't seem like the correct word to use here, perhaps "develop", "improve", or "enhance" is better.

RESPONSE TO MANUSCRIPT REVIEW

REVIEWER COMMENTS

Reviewer #1 (Remarks to the Author):

The manuscript by Sharp et al. describes the usage of fragment-based screening to identify a novel site on eIF4E that disrupts binding to eIF4G. eIF4E and eIF4G form a protein-protein interaction which stimulates cap-dependent translation, a process that becomes overactivated in many human cancers. eIF4E is a widely explored target, particularly since the discovery of the first eIF4E-eIF4G PPI inhibitor, 4EGI-1. This work reports a new scaffold that binds to a novel site outside of that of 4EGI-1 and the canonical eIF4E protein ligands, eIF4G and 4EBP1. In addition, the work also describes genetic characterization of this new binding site through dTAG-mediated genetic manipulation and rescue experiments. Unfortunately, while the binding site data translated to cellular models, compound activity was quite poor which the authors hypothesize is due to inadequate disruption of eIF4G binding. These finds, although not spectacular in the later regard, are of value to the larger field. I will provide some additional comments for improving the manuscript to enhance its rigor and novelty for publication in Nat. Commun.

1. With respect to the protein expression, eIF4E is not necessarily unstable as a monomer, just when unbound to m7G/capped mRNA; this should be updated in the text.

We agree with the reviewer and have clarified this in the text.

Line 126: "The stability of recombinant eIF4E can be improved by the presence of m7-GTP, its analogues or m7-GTP capped mRNA, however, this would lead to occlusion of the cap-binding site which was undesirable for our screening purposes."

2. I am curious to know a bit more why the 4EBP1 fragment was chosen to fuse to eIF4E since there are many crystal structures of eIF4E absent the disordered N-terminus. Would this fusion bias the results of the screen?

Both X-ray crystallographic and NMR fragment-based screening require substantial amounts of highly soluble protein. The X-ray system required apo-form crystals, many 100s of crystals, diffraction to high resolution and be amenable to compound soaking (upwards of 100mg of protein can be required for enablement studies, screening and follow up). The disordered N-terminus was chosen to attach the 4E-BP1 peptide, partly due to its proximity to the canonical site and partly due to its disordered nature, it is not observed in crystal structures. So we felt that its modification was unlikely to impact on ligand binding events and would be a suitable attachment point to target and block the hydrophobic surface with a short peptide sequence. This would deliver a monomeric protein that would be more straightforward to work with than a protein-peptide complex comprised of separate components. At the time we initiated the Project, only a limited number of structures were available in the PDB (4DUM, 4DT6, 3U7X, 2W97, 3HXL, 3HXG, 2V8Y, 2V8X, 2V8W, 2JGC, 2JGB, 2GPQ, 1WKW, 1IPC, 1IPB, 1EJ4, 1EJH and 1EJ1), few of which included protein-protein interaction partners. While being suitable for individual X-ray crystallography studies none of the

X-ray crystallography methods we tested fulfilled all the criteria required for our fragment screening approach. Therefore, based on the experimental data we generated from multiple eIF4E constructs, we took a pragmatic decision to use the 4EBP1-eIF4E fusion protein, accepting that the flat strongly hydrophobic canonical site was occluded in the engineered protein and that we would not identify hits binding there. The modified eIF4E maintained the features of the published eIF4Es with a RMSD of 0.9 Å (**Supplementary Fig. 2a**) so we were hopeful that we would identify hits binding in the Cap site or elsewhere. We have modified the text to emphasise that our engineered eIF4E maintained the structural features of the unmodified eIF4E.

Line 144: *“Occlusion of the canonical eIF4G/4E-BP binding site could disadvantage screening using this engineered system, as potential hits would be missed. However, this is balanced by the flat, strongly hydrophobic nature of the occluded region rendering fragment binding at this site unlikely. Importantly, a global alignment of the engineered eIF4E structure with the X-ray crystal structure of wild-type unmodified eIF4E (from PDB structure 5T46), reveals similar structural features verified by the root-mean-square deviation (RMSD) of 0.9Å (Supplementary Fig. 2a), making it suitable for our fragment screen strategy.”*

3. m7GxP was not used to stabilize the cap-binding site in the protein prep. What was the rationale for this? Inclusion is standard in the field unless you are looking for cap-competitive inhibitors or studying cap analogues.

We agree with the reviewer that standard practice is to include m7GxP to stabilise eIF4E, however, this would have also led to occlusion of the Cap-binding site which was undesirable for our screening purposes. With the caveats of using the eIF4E construct with the occluded canonical binding site, described in response to comment 2, we wanted to run as unbiased a screen as possible using a different hit finding strategy distinct from the previously described compound screens. The text we have added in response to reviewer comment 1 (above) also addresses this query.

4. Full-length (non-phospho) 4E-BP1 and eIF4G should both compete for binding to the second site identified. How do the authors explain the specificity observed for the eIF4G PPI?

The differential effect observed for compound **4** on eIF4G and 4E-BP1 binding to eIF4E in the gel based co-immunoprecipitation and the electro-chemiluminescent capture assays and in two cell lines was surprising (Fig. 4), as both proteins are reported to occupy similar canonical, non-canonical and lateral regions of eIF4E (Gruner et al. 2016; Peter et al. 2015). We validated the assays using a peptide (RIIY; RIIYDRKFLMECRNSPV) derived from 4E-BP1, that competed the interaction of 4E-BP1 and eIF4G with eIF4E in HeLa and SW620 cells as expected (Fig. 4).

We speculate that compared to eIF4G, 4E-BP is more tightly bound to eIF4E as a result of interactions with the non-canonical binding region, including a more rigid proline containing region in 4E-BP1 linking the canonical and non-canonical motifs. This results in \approx 5 to 10-fold tighter binding affinity for 4E-BP1:eIF4E compared to that for eIF4G:eIF4E, although with the caveat that this analysis is derived from a comparison of different modified proteins and peptides from different studies (Tomoo et al. 2006; Abiko et al. 2007 ; Umenga et al. 2011; Gruner et al. 2016). We believe our data reflects the binding affinity of compound **4** for eIF4E being sufficient to

displace eIF4G from eIF4E in lysates and also prevent further recruitment of eIF4G to eIF4E, but is insufficient to displace 4E-BP1. We speculate that our eIF4E pull-downs are detecting at least two separate pools of eIF4E, one bound to eIF4G that is affected by compound and another complexed with 4E-BP1 that is unaffected by compound exposure

Line 600: *“Displacement of eIF4G and recruitment of 4E-BP1 to eIF4E has been reported following incubation of lysates with 4EGI-1 and i4EG-BIP small molecules^{28, 51}. In contrast, despite validating the two binding assays with a 4E-BP-derived peptide, we consistently found no evidence for displacement or recruitment of 4E-BP1 following addition of compound 4 to cell lysates. This was unexpected as structural data suggests the conformational change induced by compound 4 binding to eIF4E should disrupt the non-canonical binding interface of both eIF4G and 4E-BPs (Supplementary Fig. 2c). We speculate we are measuring different pools of eIF4G:eIF4E and 4E-BP1:eIF4E and the differential displacement of eIF4G, but not 4E-BP1, by 4 in cell lysates may result from disruption of the pool of weaker eIF4G:eIF4E interactions alone^{51, 69-71}. These observations suggest further in-depth studies to understand the complexities of 4E-BP1 or eIF4G binding to eIF4E and the impact of small molecules in lysates or intact cells.”*

5. Even in lysate, the compound is much less active than the K_d would predict. The authors should perform CETSA or some other target engagement assay to demonstrate binding in this assay. Maybe it binds to many off-targets which decreases activity? Have the authors tried to express the proteins and perform biochemical studies? The authors should also consider a traditional cap pulldown assay as another standard assay to look at PPI integrity.

We now have new data on eIF4E binding using isothermal titration calorimetry and target engagement in cells using CETSA-based assay in intact cells.

1. We have characterised binding of compound 4 to purified eIF4E using isothermal titration calorimetry (ITC) in a standard and reverse format (**Table 1** and additional data now shown in **Supplementary Fig. 8**) and have modified the text accordingly. The K_d for the N127 minor variant of eIF4E was 17nM, compared to 90nM for the D127 major variant.

Line 239: *“Thermodynamically, binding of compound 4 to both eIF4E variants was driven by a large favourable enthalpic contribution with a small entropic penalty (Supplementary Fig. 8). In all cases the stoichiometry estimates indicated a 1:1 interaction between compound 4 and eIF4E.”*

We also employed a CETSA approach to explore engagement of 4 with eIF4E in intact cells (**Fig. 5a, b**). Here we clearly show that in cells, compound 4 protects eIF4E from heat denaturation in a dose dependent manner with a 2 μM EC₅₀. Importantly, the less active control compound 5 shows no evidence for target engagement at 50 μM (**Fig. 5a**) in intact cells. The CETSA binding activity in cells reflected that seen in the eIF4G:eIF4E electro-chemiluminescent binding assay for three different cell line lysates (EC₅₀s 1.4 - 2.6 μM; **Figs. 4d, e, g**). Given the positive CETSA results showing target engagement in intact cells we believe that additional cap-binding experiments are not necessary.

A 10-fold difference between binding in a recombinant protein assay and a more native environment is not uncommon. Thus the differential between the eIF4E ITC and the lysate or cellular binding data may reflect a comparison of **4** binding to purified recombinant eIF4E protein in a biophysical assay or to a complexed eIF4E making multiple protein-protein interactions in lysates or intact cells.

Finally compound **4** is also bound by plasma proteins (98.9%; **Supplementary notes**) that in cell culture experiments, supplemented with 10% serum, may contribute to some of the \approx 10-fold drop-off between ITC for **4** binding to eIF4E and eIF4E-binding in intact cells determined by CETSA.

Line 317: “A cellular thermal shift assay performed in intact cells demonstrated target engagement, with compound **4** binding and stabilising eIF4E⁵⁷. We initially showed that treatment of H1299 cells with 50 μ M of **4**, but not **5**, protected eIF4E from thermal denaturation (Fig. 5a). We subsequently showed clear dose-dependent thermal protection of the eIF4E ($EC_{50} = 2 \mu$ M; Fig. 5b) in intact cells which was similar to the EC_{50} s for disrupting eIF4G:eIF4E binding in lysates (Fig. 4).”

6. The data in figure 4E for compound 4 shows no dose-dependence at the higher concentrations. Based on the data presented, I am not sure how confident that one can be that this activity is on-mechanism.

The 50% inhibition of the cap-dependent translation assay required for cap dependent translation for compound **4** falls between 1-10 μ M (**data is now Fig. 4h**) and is similar to the single digit μ M IC_{50} s for disruption of binding of eIF4G to eIF4E in lysates from multiple cell lines (**Fig. 4c-e,g**). Importantly there is no activity in the control cap-independent translation reporter (**Fig. 4h**).

To give us more confidence in selectivity/on-target activity of compound **4** we also employed a negative control probe, compound **5**. Comparison of the physical and topological properties of chemical probes and their matched negative controls suggests that chemical similarity is a critical factor that dictates the functional value of negative controls (Lee and Schapira 2021; <https://openlabnotebooks.org/are-enantiomer-of-chemical-probes-good-negative-controls/>). Compound **5**, is a diastereoisomer of compound **4** that has the same chemical composition as compound **4** but with inverted stereochemistry at the benzylic position. Compound **5** binding is significantly less potent (30-fold reduced binding affinity by ITC; **Table 1**) and the X-ray crystal structure of compound **5** with eIF4E shows it is unable to form the same network of hydrogen bonding interactions as the active compound **4** (**Supplementary Fig. 7c, d**). For all our experiments the diastereomeric-purity of compounds **4** and **5** were confirmed by quantitative ¹HNMR and high-performance liquid chromatography (**Supplementary information**). In lysate binding assays from three different cell backgrounds (HeLa, SW620 and H1299; **Figs. 4c-e and g**) compound **4** significantly inhibits eIF4G binding to eIF4E, but compound **5** does not (**Fig. 4e, g**). Importantly, in the *in vitro* translation assay questioned here in **Fig. 4h** compound **5** has little or no activity at any of the concentrations tested. CETSA showed no evidence that compound **5** protected eIF4E at concentrations as high as 50 μ M (**Fig. 5a**) in intact cells. Taken together we firmly believe that compound **4** is binding to eIF4E in lysates or cells and where activity is detected is acting through an on-target mechanism.

Line 244: *“During the development of tool compounds or drugs a recommended practice is to use a negative control compound to give confidence that the observed cellular phenotype may be driven by inhibition of the targeted protein. Comparison of the properties of chemical probes and their matched negative controls suggests that chemical similarity is a critical factor that dictates the value of a negative control⁵⁵. The diastereoisomer of **4**, compound **5**, with inverted stereochemistry at the benzylic position fulfilled the criteria for a negative control as it had the same chemical composition as **4** but was unable to form the same network of hydrogen bonding interactions as **4** leading to significantly reduced potency (Supplementary Fig. 7c, d and Table 1). To provide additional confidence in the selectivity/on-target activity of **4** we used **5** as a negative control in subsequent biological validation work⁵⁵.”*

7. On a positive note, the dTAG studies nicely demonstrate the complexity of regulation of cap-dependent translation initiation. 4EGI-1 has not been shown to directly bind to eIF4E in cells, and the authors are correct, that many other liabilities exist with other reported compounds.

We thank the reviewer for these positive comments.

Reviewer #2 (Remarks to the Author):

The work presented in the manuscript by Sharp and colleagues explores drugability and function of eIF4E through alternative, underexamined routes addressing canonical and non-canonical binding sites of eIF4E. For this they integrate fragment-based screening, structural biology and functional assays utilising a novel designed eIF4E variant which allows to survey fragment/compound binding to previously inaccessible pockets of eIF4E due to protein stability issues. Through their approach, they identify a tool compound binding to a pocket of eIF4E involved in non-canonical binding to eIF4G and 4E-BP1. As a results of compound binding, the interaction between eIF4E-eIF4G in cell lysates is selectively disrupted leaving the 4E-BP1 association to eIF4E intact. In vitro reporter translation assay reveal that the new compound specifically inhibits cap-dependent translation. Based on their structural data they identify key residues of eIF4E involved in eIF4G binding via the non-canonical binding, which they validate by testing eIF4G-binding to derived variants of eIF4E in cell lysates. To then investigate the role of the non-canonical eIF4G-binding site in cells they develop and establish a fast-response, targeted eIF4E degradation, knock-in cell system to overcome limitation of cell viability to due eIF4E loss by conventional strategies. They validate the eIF4E-degradation system by examining the rescue potential of various eIF4E variants that are overexpressed simultaneously to the induced degradation of endogenous eIF4E. With this they discover that some of the structure-guided eIF4E variants mutated in the non-canonical eIF4G binding site cause cell growth defect and MCL1 repression, as a reference for eIF4E-dependent translation. Finally, they investigate the effect of their tool compound in the engineered cells. While the compound disrupts the eIF4G-eIF4E interaction in lysates of these cells, the compound lacks potency when directly administered to the cells. As a result the compound itself does not impact cell growth. However the compound shows some synergy with a variant of eIF4E that is partially disrupted in its canonical eIF4G binding site. The authors conclude their works that full potential of disrupting the non-canonical eIF4G binding site is only accessed in combination with interference at the canonical eIF4G binding site. They further suggest that their tool compounds could be used for development of dual-targeting eIF4E for degradation.

The presented work explores two major directions: defining the biological impact of the non-canonical eIF4G interaction site and determination of the drugability of that site. The authors present a series set of experiments using in most cases reasonable controls supporting their conclusions. The overall presentation is generally compelling, interesting and has good potential to provide new directions and tools in targeting eIF4E and uncover more about it's biology. However, some key observations and conclusions are based on only one type of experiment (the luminescence assay) and require more thorough investigation before it is suitable for publication. I have three major points that I believe need to be addressed:

We thank the reviewer for their positive and constructive comments and also for their detailed review, we have addressed their three major review points below.

The biological role of site 2:

a) There is an apparent, broad functional disconnect between the effect of the generated eIF4E variants in site 2 on eIF4G binding after overexpression (Fig 56-c) and their impact on eIF4E function (Fig 7). Experiments from Fig 5 were suggesting different protein stabilities of the mutant 4Es. Is that somewhat compromising interpretation of the capability of these mutants to rescue or not rescue the induced 4E loss in Fig 7? They state in lines 364-367 that blots were suggesting similar protein levels but blots seem to stem from different membranes and second the blots wouldn't show turnover of the protein as the mutants might not be able to form their native complexes due to stability issues as pointed out in Fig 5. Particularly L134R. Furthermore they state L85R had a partial effect. How do they explain the disconnect and the reasoning for the double mutation, what is the stability of the double mutant?

Previous reports have demonstrated poor or no expression of some eIF4E mutants due to increased proteasomal degradation of the mutant predicted to result from disruption of binding partner interactions that normally would protect eIF4E from turnover (Murata and Shimotohno, 2006; Wendel et al, 2007). With this in mind, we decided to first determine mutant expression and found that 7 of the 9 site 2 eIF4E mutants repeatedly showed < 5% of the wild-type control following transfection, possibly due to eIF4E destabilisation (**Supplementary Fig. 12**). Only mutations that showed expression level of > 25% of the wild-type control were used in subsequent experiments (W56A, W73F, L85R, L134R and S209A). For the electrochemiluminescent binding assay we not only measured captured eIF4G, but also total FLAG-eIF4E to allow normalisation. We have added an additional plot showing normalisation of eIF4G binding relative to expression of the expressed eIF4E (**Fig. 6d**). This data clearly shows that W73F, L85R and L134R all have significantly ($p < 0.01$) impaired binding to eIF4G, irrespective of the levels of their expression.

Supplementary Fig. 17b now shows the protein expression of the rescue constructs. we cropped the blots into blocks representing each rescue condition to aid interpretation. The original immunoblot raw data images for the whole paper are now shown as in the **Supplementary Information** file and we have amended the figure legend accordingly.

Supplementary Fig. legend 17b: *“The immunoblots were cropped to aid interpretation of the data, original immunoblot images for all repeats are available in the Supplementary information file”*

We could not accommodate all the samples on a single gel but did include vinculin, a protein not affected by eIF4E loss, as a loading control for normalisation. To quantify the degree of variability in expression of eIF4E from the different rescue constructs we have now normalised the eIF4E WT or mutant expression shown in the immunoblots in **Supplementary Fig. 17b, c** and plotted the data relative to the wild-type control construct (**Supplementary Fig. 17e, f**). The data shows that the W73F, L85R, L134R and W73F/L85R constructs have similar levels of expression ($p = ns$), that are significantly ($p < 0.05$) lower than the expression of the wild-type construct. However, despite having similar levels of reduced expression W73F, L85R, L134R and W73F/L85R constructs showed different degrees of rescue with L134R and double W73F/L85R mutants showing impaired rescue of eIF4E, while W73F rescued proliferation and MCL1 biomarker expression, and L85R rescued proliferation but only partially rescued the MCL1 biomarker. This suggests that in the rescue experiments

described here the variation in expression of mutant eIF4Es we have used is unlikely to be a driving influence on the degree of rescue in these experiments.

Line 480: *“Importantly, in these experiments the observed levels of variation in rescue construct expression were unlikely to account for differences in their ability or inability to rescue loss of the eIF4E-dTAG as for example, the L85R and L134R mutants were equally expressed in the dTAG-eIF4E model (Supplementary Fig. 17e-f), but showed opposing effects in the rescue experiments (Fig. 8a-c; Supplementary Fig. 17b-d).”*

The double mutant was constructed based on our observations that both W73F and L85R mutants retained function and were able to rescue proliferation and completely or partially rescuing the MCL1 biomarker respectively. We have also clarified the rationale for creating the W73F/L85R double mutants.

Line 487: *“The full retention of activity of the W73F mutant in the proliferation and MCL1 biomarker rescue assays and the full rescue of proliferation, but partial rescue of the MCL1 biomarker by the L85R mutant suggested that in intact cells the impact of these mutants was not sufficient to fully disrupt the eIF4E function.”*

While setting up the CETSA binding assays we also determined the thermal denaturation profile for the W73F, L85R, L134R and W73F/L85R mutants (**Supplementary Fig. 18a,b**). The W73F and L85R mutants were more susceptible to denaturation than the WT protein, with the inactive L134R and double mutant showing even greater susceptibility to thermal denaturation.

Line 513: *“Binding studies with the mutant eIF4E expressing cells showed that the active W73F and L85R mutants were denatured at a lower temperature than the WT, an effect that was even more pronounced in the inactive L134R and W73F/L85R mutants (Supplementary Fig. 18a,b)”*

We also noted that only the single W73F mutant showed evidence for increased expression following dTAG^V-1 treatment (**Supplementary Fig. 17b**, new data in **Fig. 8d** and **Supplementary Fig. 19c**). Treatment with the active compound **4** decreased W73F expression and also prevented the increased expression associated with loss of the dTAG-eIF4E (**Fig. 8d** and **Supplementary Fig. 19c**). We selected W73F for our study as it has been previously reported that this mutant disrupts the canonical interaction of eIF4E with eIF4G and 4E-BP (Wendel *et al.*, 2007). A previous reports has suggested that poor expression of a different eIF4E-W73 mutant (W73A) was associated with loss of ligand binding leading to elevated proteasomal degradation compared to the wild-type eIF4E (Murata and Shimotohno, 2006).

We speculate that the W73F mutation weakens the affinity for eIF4G sufficiently that the W73F mutant cannot compete wild-type endogenous eIF4E:eIF4G binding in cells. This would explain the absence of binding to eIF4G in **Fig. 6** as the endogenous eIF4E is expressed in this binding assay. We confirmed the eIF4E-dTAG fusion protein binds eIF4E (**Supplementary Fig. 13a**) and therefore could also compete with W73F for eIF4G binding. We speculate that in the absence of the competitor eIF4E-dTAG protein following dTAG^V-1 treatment the expressed W73F mutant retains sufficient affinity to bind eIF4G and rescue function. This would also explain the increased levels of W73F protein following loss of dTAG-eIF4E as W73F bound to ligands such as

eIF4G will be protected from degradation. In this model, addition of the L85R to create the W73F/L85R double mutant would act cooperatively to further weaken the binding affinity for eIF4G, to a degree where the double mutant can no longer bind eIF4G or rescue function even in the absence of wild-type eIF4E. Similarly treatment of the W73F mutant with compound **4** would weaken its affinity for eIF4G leading to our observation of reduced rescue and decreased W73F protein expression compared to dTAG^V-1 alone condition.

Line 618: *“In addition to disrupting canonical binding of eIF4G and 4E-BP, W73 mutants are susceptible to proteasomal degradation thought to result from the loss of protective interactions with binding partners^{60, 62}. CETSA analysis showed that W73F was more susceptible to thermal denaturation but retained a site 2 capable of binding compound **4**. We speculate that in the initial co-immunoprecipitation binding experiments run in the presence of the endogenous eIF4E, the weaker binding exogenous W73F mutant cannot compete with the endogenous eIF4E for eIF4G-binding, resulting in the assay reporting an absence of binding for W73F. The rescue experimental format is run in the absence of any competitive eIF4E following dTAGV-1 treatment and leaves the weaker binding W73F mutant free to bind eIF4G and rescue function. This would also explain the increased levels of W73F protein following loss of eIF4E-dTAG that would result from W73F being protected from degradation through eIF4G binding^{60, 62}”*

b) In my opinion the generated dTAG-eIF4E cell line is insufficiently characterised in regards to its effect on eIF4E function.

Both our and DEPMap data (**Supplementary Fig. 11a, b**) show that genetic loss of eIF4E results in reduced H1299 cell proliferation/survival. The main purpose of the eIF4E-dTAG model is to allow the expression of mutants of eIF4E that may disrupt function and be lethal to cells. The establishment stable clonal lines lacking endogenous eIF4E, but expressing the eIF4E-dTAG-fusion protein tagged at its N or C-terminus is consistent with the dTAG fusion protein maintaining the essential cell survival functions of eIF4E. The cells also express similar levels of MCL1 as their parent counterpart suggesting the tag has no effect on short-lived proteins (**e.g. Fig. 7b and Supplementary Fig. 13b**). The cells expressing the fusion protein clearly remain dependent on eIF4E as degradation of the eIF4E-dTAG protein reduces cell proliferation and MCL1 biomarker expression that is rescued by WT eIF4E expression, but not the frequently used defective cap-binding W56A mutant that cannot bind mRNAs and initiate translation (**Fig. 8c**).

In addition, we confirmed that the N- or C- tagged-eIF4E fusion protein retained eIF4E function. We showed the addition of the FKBP12-tag did not significantly affect binding to eIF4G1 when compared to eIF4E lacking the FKBP12^{F36V}-tag in co-immunoprecipitation experiments (**Supplementary Fig. 13a**). We also showed expression of both N- or C- dTAG-eIF4E could rescue MCL1 expression, a commonly used molecular readout for eIF4E function (Wendel et al, 2007), following eIF4E siRNA knockdown. We have also now included proteome profiles following 72 hour culture and found few if any consistent changes in proteome profile between the parent line and the two C- or two N-terminal tagged-eIF4E clones (**Fig. 7e and Supplementary Fig. 16a**). Finally we also included new data showing that the C3-1 eIF4E-dTAG clone had a similar CETSA binding profile to the endogenous eIF4E for compound **4** in intact

cells (**Supplementary Fig. 13c**). Overall, these data are consistent with the tagged form of eIF4E retaining the function of the endogenous eIF4E.

Line 406: *“We confirmed that both the C- and N-terminally tagged eIF4E-dTAG protein retained the ability to bind to eIF4G and were able to rescue MCL1 expression, a commonly used cellular biomarker of eIF4E activity⁶⁰ following siRNA knockdown of endogenous eIF4E (Supplementary Fig. 13a,b).”*

Line 426: *“In addition, the eIF4E-dTAG protein in the C3-1 clone retained a similar target engagement profile as endogenous eIF4E in cells treated with compound 4 (Supplementary Fig. 13c; Fig. 5a,b).”*

Line 432: *“We confirmed that expression of the eIF4E-dTAG was similar to the expression of endogenous eIF4E in the parent line (eIF4E-dTAG/parent eIF4E protein expression ratio of 0.8 to 1.7 – fold; Fig. 7d). There were also no major significant differences between the global proteome profiles of the control parent and untreated control dTAG models (Fig. 7e Supplementary Fig. 16a).”*

Line 445: *“Overall, eIF4G binding, rescue from siRNA knockdown or CRISPR knockout and proteome profiling data confirmed that the eIF4E-dTAG could functionally replace the endogenous eIF4E and clearly showed the robustness of the dTAG-system to efficiently and rapidly remove the target proteins.”*

The authors stem their most important experiments investigating the biological relevance of site 2 and their drug on this cell line. However, they only control their system with the cell lysate assay that seems to be disconnected from the in cell behaviour, and assess eIF4E functionality through following expression of one protein MCL1.

We have addressed these comments as itemised responses below.

In addition it appears that tagged-4E expression levels might be different as compared to parental wildtype levels, caused by the different stabilities they observed in Fig5.

From proteome profiling experiments the ratio of expression of dTAG-eIF4E to endogenous-eIF4E expression in the parent line control ranged from 0.8 to 1.7-fold for the 4 different clones.

Line 430: *“The four eIF4E-dTAG CRISPR clones were also characterised by global proteome profiling comparing eIF4E-dTAG degradation with eIF4E siRNA knockdown in the parent H1299 cells. We confirmed that expression of the eIF4E-dTAG was similar to the expression of endogenous eIF4E in the parent line (eIF4E-dTAG/parent eIF4E protein expression ratio of 0.8 to 1.7 – fold; Fig. 7d).”*

The functionality of the engineered eIF4E in this cell line should be addressed via

i) at least one global technique like RNA-seq or proteomics comparing parental and engineering lines. MCL1 has a short half life, what about long lived 4E targets?

We have included proteome profiling data comparing dTAG depletion in 4 different clones (2 N- and 2 C-terminal tagged) to the parent line and to eIF4E siRNA treatment of the parent line at 72 hr (**Fig. 7d,e and Supplementary Fig. 16**). We selected 72 hr to allow for the siRNA knock-down to take effect and also to measure effects on long-lived proteins. The data shows there are very few consistent differences, none of which are significant, between the global proteome profiles of the control parent and the 4 untreated eIF4E-dTAG clones (**Fig. 7d,e and Supplementary Fig. 16**). dTAG treatment of the eIF4E-dTAG clones resulted in loss of eIF4e signal to background signal levels, while siRNA treatment resulted in reduced eIF4E expression but not to the same degree as observed in the dTAG models. This clearly shows the power of the dTAG to efficiently remove the target protein (**Fig. 7d**). Not surprisingly given the differences in eIF4E protein expression, the dTAG treatment resulted in a greater number of significant protein changes ($P_{adj} < 0.05$) than the eIF4E siRNA. However, there was some overlap between siRNA and dTAG and exemplars are shown (**Supplementary Fig. 16**). Interestingly, significant MCL1 depletion was only detected in the dTAG model, again perhaps reflecting the different efficiencies of eIF4E protein depletion. The overall number of altered proteins significantly regulated by eIF4E depletion was surprisingly low with either method of disrupting eIF4E expression, however, despite the low level of significantly altered expression, both knockdown and dTAG-mediated loss of eIF4E affected cell proliferation.

Line 430: *"The four eIF4E-dTAG CRISPR clones were also characterised by global proteome profiling comparing eIF4E-dTAG degradation with eIF4E siRNA knockdown in the parent H1299 cells. We confirmed that expression of the eIF4E-dTAG was similar to the expression of endogenous eIF4E in the parent line (eIF4E-dTAG/parent eIF4E protein expression ratio of 0.8 to 1.7 – fold; Fig. 7d). There were also no major significant differences between the global proteome profiles of the control parent and untreated control dTAG models (Fig. 7e Supplementary Fig. 16a). Compared to their controls, knockdown of eIF4E by siRNA treatment resulted in reduced eIF4E expression but not to the same extent as the eIF4E-dTAG models (Fig. 7d). dTAG^{V-1} treatment resulted in a greater number of significantly altered protein expression protein changes ($p_{adj} < 0.05$) compared to the eIF4E siRNA knockdown (Supplementary Fig. 16b, c). Significant depletion of MCL1 ($p_{adj} < 0.05$) was only detected in the dTAG models which may reflect the more robust depletion of eIF4E protein compared to the siRNA conditions (Fig. 7d).*

Overall, eIF4G binding, rescue from siRNA knockdown or CRISPR knockout and proteome profiling data confirmed that the eIF4E-dTAG could functionally replace the endogenous eIF4E and clearly showed the robustness of the dTAG-system to efficiently and rapidly remove the target proteins."

ii) m7G pulldowns and controlling integrity of eIF4F complex formation

We have not included m7G pulldowns as we have data showing co-immunoprecipitation of the eIF4E-dTAG with eIF4EG and with compound **4** using CETSA-binding. We also show the Cap-binding mutant W56A is unable to rescue eIF4E-dTAG degradation indicating that the eIF4E-dTAG requires its Cap-binding function to preserve cell survival.

iii) providing a time course induction of eIF4E degradation and kick-off of cell effects in the presence and absence of overexpression of their variants. At the moment only very long time courses are given.

We are uncertain what this comment refers to. Based on preliminary characterisation we detected degradation \approx 4 hrs post-dTAG^V-1 treatment (data not shown). Therefore, for the 4 clones and parent H1299 cell line we initially characterised MCL1 expression, as an exemplar of a published eIF4/cap-dependent molecular biomarker, at 6, 8 and 16 hrs (**Fig. 7b**) at \approx 2.5 and 5 x the dTAG^V-1 GI₅₀ (dTAG^V-1 IC₅₀ = 205 nM determined following 5 days continuous exposure; **Supplementary Fig. 14a**). We also determined effects on growth with measurements taken in real time every 4 hrs (from 0 to 144 hrs) comparing treatments with dTAG^V-1 with an inactive matched control dTAG^V-1-NEG (**Fig. 7c and Supplementary Fig. 14c**).

We selected the C3-1 clone for the rescue experiments and we show a 6-, 24-, 48- and 72-hour time-course of eIF4E-dTAG protein and MCL1 biomarker expression (**Supplementary Fig. 14b**) treated with 500 nM of the active dTAG^V-1 (\approx 2.5 x the dTAG^V-1 GI₅₀) or negative control dTAG^V-1-NEG. All the rescue experiments with the variants feature quantitation of the dTAG-eIF4E and MCL1 biomarker at 6 and 16 hrs (**Fig. 8a; Supplementary Fig. 17c-d**). In all the experiments with variants we also show real time growth assessed from 0 to 144 hrs with assay points every 4 hrs (**Fig. 8c**).

Collectively these data show molecular and cellular data ranging from 6 hr to 7 day exposures.

The compound activity: The authors have identified a tool compound with binding capacity to eIF4E site 2 and activity of this compound when added to cell lysate. However, the compound seems to lack activity when administered directly to cells. While this seems not uncommon for early stage in compound development, it should be more addressed experimentally why this disconnect establishes in order to support suitability of the compound for further development as suggested by the authors. I suggest the following points:

a) Does the compound bind to eIF4E in cells? What is the binding constant in cells? Potential off target or serum binding?

We have included new CETSA data showing compound engagement of eIF4E in intact cells. We clearly show that in intact cells compound **4** protects eIF4E from heat denaturation in a dose dependent manner with an EC₅₀ of 2 μ M, similar to the range of EC₅₀ values (1.4-2.6 μ M) determined for binding in cell lysates.

To help address off-target activity and give us more confidence in the selectivity/on-target activity of compound **4** we also employed a negative control probe. As described in the response to **reviewer 1 comment 6**, comparison of the physical and topological properties of chemical probes and their matched negative controls suggests that chemical similarity is a critical factor that dictates the functional value of negative controls (Lee and Schapira 2021; <https://openlabnotebooks.org/are-enantiomer-of-chemical-probes-good-negative-controls/>). Our negative control, compound **5**, is a diastereoisomer of compound **4** that has the same chemical composition as compound **4** but with inverted stereochemistry at the benzylic position. Compound **5** binding is significantly less potent (30-fold reduced binding affinity by

ITC; **Table 1**) and the X-ray crystal structure of compound **5** with eIF4E shows it is unable to form the same network of hydrogen bonding interactions as the active compound **4** (**Supplementary Fig. 7c,d**). For all our experiments the diastereomeric-purity of compounds **4** and **5** were confirmed by quantitative ¹HNMR and high-performance liquid chromatography (**Supplementary information**). In lysate binding assays from three different cell backgrounds (HeLa, SW620 and H1299; **Figs. 4c-e and g**) compound **4** significantly inhibits eIF4G1 binding to eIF4E, but compound **5** does not (**Fig. 4e, g**). In the *in vitro* translation assay in **Fig. 4h** compound **5** has little or no activity at any of the concentrations tested. Importantly, the less active control compound **5** also shows no evidence for cell target engagement at 50 μM in intact cells using CETSA analysis (**Fig. 5a**).

As detailed in response to **reviewer 1 comment 5**, a 10-fold difference between binding in a recombinant protein assay and a more native lysate or cellular environment is not uncommon. Thus the differential between the eIF4E ITC and the lysate or cellular binding data may reflect a comparison of **4** binding to purified recombinant eIF4E protein in a biophysical assay or to a complexed eIF4E making multiple protein-protein interactions in lysates or intact cells. Finally compound **4** is also bound by plasma proteins (98.9%; **Supplementary information**) that in cell culture experiments, supplemented with 10% serum, may contribute to some of the ≈ 10-fold drop-off between ITC for **4** binding to eIF4E and eIF4E-binding in intact cells determined by CETSA.

Taken together we firmly believe that our compound **4** is binding to eIF4E in lysates or cells.

Line 244: *“During the development of tool compounds or drugs a recommended practice is to use a negative control compound to give confidence that the observed cellular phenotype may be driven by inhibition of the targeted protein. Comparison of the properties of chemical probes and their matched negative controls suggests that chemical similarity is a critical factor that dictates the value of a negative control⁵⁵. The diastereoisomer of **4**, compound **5**, with inverted stereochemistry at the benzylic position fulfilled the criteria for a negative control as it had the same chemical composition as **4** but was unable to form the same network of hydrogen bonding interactions as **4** leading to significantly reduced potency (Supplementary Fig. 7c, d and Table 1). To provide additional confidence in the selectivity/on-target activity of **4** we used **5** as a negative control in subsequent biological validation work⁵⁵.”*

Line 317: *“A cellular thermal shift assay performed in intact cells demonstrated target engagement, with compound **4** binding and stabilising eIF4E⁵⁷. We initially showed that treatment of H1299 cells with 50 μM of **4**, but not **5**, protected eIF4E from thermal denaturation (Fig. 5a). We subsequently showed clear dose-dependent thermal protection of the eIF4E (EC₅₀ = 2 μM; Fig. 5b) in intact cells which was similar to the EC₅₀s for disrupting eIF4G:eIF4E binding in lysates (Fig. 4).”*

b) Binding constant of compound to the different engineered versions of eIF4E used in the cell lines

We have not determined compound binding constants for the different engineered variants of eIF4E, but have determined the binding of the variants to compound in intact cells using CETSA (see response to **reviewer 2 comment d** below).

c) Metabolic turnover of the compound?

Human liver microsome data suggests that **4** may have moderate clearance *in vivo* (**Supplementary information**), however, the enzymes present in microsomes are unlikely to be present in SW620 or HeLa cells at appreciable levels. Caco2 permeability data indicated good permeability and moderate to low efflux (Papp (A-B) 9.0×10^{-6} cm/s, efflux ratio 1.5) suggesting that poor cellular uptake or enhanced efflux was unlikely to be the cause of the drop-off for activity in intact cells.

Line 309: *“Characterisation of compound **4** in human liver microsomes suggested that **4** had moderate clearance in vitro (Supplementary information), however, the enzymes present in microsomes are unlikely to be present in cell line models at appreciable levels. Compound **4** also exhibited good permeability as measured in a Caco2 assay (Papp (A-B) 9.0×10^{-6} cm/s, efflux ratio 1.5; Supplementary information) indicating that poor cellular uptake or efflux was unlikely to influence compound **4** activity in intact cells (Supplementary information).”*

d) They generated variants of eIF4E disrupted in site 2 but did not test if these are resistant to compound binding

We used the CETSA assay to determine compound binding to the eIF4E mutants in intact cells. We noted that in the absence of compound the mutant were denatured at lower temperatures WT > W73, L85R > L134R, W73F/L85R (**Supplementary Fig. 18b**). The addition of compound **4** reduced W73F levels rather than stabilising the protein. Reduced W73F protein could result from engagement with compound **4** as ligand binding in CETSA can also result in protein destabilisation, however, this interpretation is complicated by our observation that W73F protein expression is reduced by the addition of compound **4** to dTAG-treated cells (**Fig. 8d**). There was evidence of stabilisation and binding to the L85R mutant while the two mutants that could not rescue eIF4E function (L134R and W73/L85R) showed no evidence for compound **4** binding. The negative control compound **5** showed no evidence for binding to any of the mutant proteins tested.

Line 513: *“Binding studies with the mutant eIF4E expressing cells showed that the active W73F and L85R mutants were denatured at a lower temperature than the WT, an effect that was even more pronounced in the inactive L134R and W73F/L85R mutants (Supplementary Fig. 18a,b). The W73F appeared to be destabilised, rather than stabilised, by compound **4** that may in part be due to the effects on its expression following treatment (Fig. 8d). Indeed, the destabilisation effect was specific to the active compound, but not the inactive compound **5** (Supplementary Fig. 18c). The L85R mutant showed evidence for compound **4** binding (Supplementary Fig. 18c). This was anticipated as switching the hydrophobic L85 to a larger, polar arginine side chain was predicted to cause steric and electrostatic clashes with F47, C89, Tyr91 and V156 and favour the loop out conformation required for compound **4** binding (Fig. 2 and Supplementary Fig. 7a). Neither the inactive L134R that is predicted to disrupt site*

2 nor the inactive W73F/L85R mutant that is predicted to disrupt both the canonical and non-canonical site showed evidence for compound 4 binding (Supplementary Fig. 18c).

e) They only show growth curves and the luminescence assay as a read out. Is MCL1 (or other eIF4E targets) affected in their expression in Fig 7d and 8d?

For Fig. 7d (now Fig. 8c) we show quantified MCL1 expression at 6 and 16 hours post-dTAG^V-1 treatment from three independent repeats (Fig. 8a and Supplementary Fig. 17b-d).

For Fig. 8d we have now included new MCL1 data (Fig. 8d and Supplementary Fig. 19c). dTAG^V-1 treatment of the C3-1 clone resulted in reduced MCL1 expression that was rescued by WT or W73F expression, and partially rescued by L85R. Treatment with compound 4 in the absence of dTAG^V-1 treatment had no effect on MCL1, consistent with its poor effects on protein synthesis assays in intact cells (Fig. 5d and Supplementary Fig. 10a, b). Co-treatment of 4 and dTAG^V-1 in presence of L85R gave no additional loss of MCL1. Co-treatment in the presence of W73F appeared to reduce MCL1 on the immunoblot images, however, on quantification this loss was not statistically significant (Supplementary Fig. 19c). This limited effect on MCL1 biomarker expression may reflect the partial (~50%) sensitisation detected in W73F rescue cells following the dTAG^V-1 and compound 4 co-treatment.

Line 529: “We treated the C3-1 clone, W73F and L85R dTAG-rescue models with 25 μM compound 4 in combination with dTAG^V-1. Treatment with compound 4 in the absence of dTAG^V-1 treatment had no effect on the MCL1 biomarker or proliferation in any of the models tested, consistent with previously detected poor effects on protein synthesis in intact cells (Fig. 8d, Fig. 5e, Supplementary Fig. 10). Compound 4 and dTAG^V-1 co-treatment of C3-1 or the L85R rescue line gave no additional loss of proliferation or MCL1 expression (Fig. 8d,e and Supplementary Fig. 19a). The increased expression of W73F protein expression we previously detected following dTAG^V-1 treatment was reduced by co-treatment with 4 (Fig. 8d, Supplementary Figs. 17b,e and 19c). In addition, the combined treatment of compound 4 and dTAG^V-1 in the W73F mutant model resulted in a significant combinatorial effect on proliferation that was similar to that seen with the L85R/W73F double mutant (Fig. 7c and 8e). MCL1 protein expression was decreased on the immunoblot images, however, this decrease was not significant upon quantification (Fig. 8d and Supplementary Fig. 19c), which we concluded to be a reflection of the partial (~50%) sensitisation following the combined dTAG^V-1 and compound 4 co-treatment. Importantly no significant effects were seen with the neg-dTAG^V-1 and compound 4 (Supplementary Fig. 19a) or the less active compound 5 and dTAG^V-1 (Supplementary Fig. 19b).”

Insufficient labels of Figures and description of methods: Throughout the manuscript there are many instances in which Figure legends and methods are insufficiently described or completely in the wrong place, which should be address to allow better understanding of how experiments were performed

We have addressed the individual points listed below.

a) Fig 2 and Supplementary Fig 1 and 2: Unclear if screening was done in presence or absence of m7G

As described above in the response to **reviewer 1 comment 1**, we wished to run as unbiased a screen as possible using a different hit finding strategy – a fragment-based approach – distinct from the previously described compound screens. To accomplish this we avoided the use of m7-GTP and its analogues as this would have led to occlusion of the Cap-binding site which was undesirable for screening purposes. Our approach was exemplified by the discovery of multiple fragments binding to the Cap-binding site or the second binding site (Supplementary fig. 2a,b). For clarification we have edited the text to:

Line 126: “The stability of recombinant eIF4E can be improved by the presence of m7-GTP, its analogues or m7-GTP capped mRNA, however, this would lead to occlusion of the cap-binding site which was undesirable for our screening purposes.”

b) Crystal structures shown with m7GTP in the structure but its unclear if co-crystallised/soaked with compound and m7 ? not stated in methods

m7-GTP was not included in X-ray crystallography, in the original figure we added a ligand to the image for illustrative purposes only. We have amended the legend for Fig. 2 to make this clearer. The only exception was when cooperative binding with m7-GDP was observed during the NMR fragment screen. A co-soaking experiment was then carried out for those limited examples (see **Supplementary Fig. 2b** for an example of a fragment hit bound cooperatively with m7-GDP).

Line 1270: “(the Cap-site ligand (m7-GTP) is shown for illustration purposes only and was not included during protein purification or the screening process) “

c) Is it possible to describe more details of what the Astex Pyramid platform is and some basics of the screening.

Fragment-based methodology and Astex’s Pyramid platform has been extensively published on over the past 20 years. The Astex Pyramid platform integrates a range of high-throughput biophysical techniques for screening. These include X-ray crystallography, nuclear magnetic resonance spectroscopy, thermal shift, surface plasmon resonance, mass spectroscopy and calorimetry, combined with fragment library design and computational methodologies. It is used to experimentally characterize in detail the interactions of hits from fragment-based screening (very low molecular weight compounds) with their target proteins. Details of the ligand observed NMR screen, crystallography and calorimetry are presented in the materials and methods section. We have added a brief sentence and additional references for the platform in the results section.

Line 154: “A library of 1371 fragments was screened against the apo, engineered protein using Astex’s Pyramid™ platform with a combination of ligand observed NMR and X-ray crystallography^{35, 45-47}.”

d) Cross species sequence conservation analysis: How was it done and which species does it include? No information given. What are the data points in the

graphs of supplementary Fig. 2? Which structures have been used in d e f? If it's the ones from the main text, this is all human structures. What is the general background conservation? Seems generally very high so that eIF4E and site 2 is generally just highly conserved in general?

We have included a Methods section detailing the approach we used and added **Supplementary data (sheets 3, 4)** and **Supplementary Fig. 4** to show a comparison of human and representative ortholog sequences of eIF4E used for the analysis and the sequence conservation for the entire protein compared to the three sites of interest. As can be seen from **Supplementary data sheets 3 and 4** the global sequence conservation across 115 orthologues is reasonably high at > 75%, however for each of the 3 binding sites the trend is for higher conservation than the global. For site 2, 95% of orthologues have the same or greater sequence identity compared to the global sequence.

Line 743: “Protocol for evaluating the evolutionary conservation of binding sites eIF4E orthologs were identified by BLASTP searches (E value < 0.01) against SwissProt / TrEMBL protein sequences from the mammalian, vertebrate, and rodent databases⁷⁴. A multiple-sequence alignment (MSA) of the top BLASTP hits from each species was carried out using MAFFT75. The global sequence identity of each ortholog was plotted against the sequence identity of each identified binding site (Supplementary Fig. 5) (cap site (site 1), site 2 and the canonical PPI site). To define the site, we selected the 20 protein residues closest to a representative ligand, for site 1 – m7-GTP, for site 2 – compound 4 and for the canonical PPI site – a 4E-BP1 derived peptide (from PDB structure 3U7X). Overall sequence and specific site identity for representative examples is included in tabulated form for comparison (Supplementary Fig. 4 and Supplementary data 3 and 4).”

e) Supplementary Fig 4 legend for b c is incorrect

The data from Supplementary Fig. 4d has now been incorporated into **Supplementary Fig. 9b** and the legend corrected.

f) Fig 4c it is unclear from methods and description if data for flag-4E or endogenous 4E interactions is shown.

Endogenous eIF4E was used for the interaction data (Fig. 4c now **Fig. 4b,c**). We have edited the Fig. legend accordingly (**Line 1298**).

g) Various figures lack units on the y-axis or explanation to what the data was normalised.

We have been through the figures and their legends and addressed units and data normalisation where necessary.

h) Suppl8: What means “WT Cterm” or “WT Nterm”? I find the description confusing

We have edited the figure legend and figure to clarify that the terms refer to FLAG or dTAG protein tags on the N- or C-terminus of eIF4E (now **Supplementary Fig. 13**).

Minor:

a) No statistical methods are used to support statements made in the manuscript (except Fig 5b). It could help to strengthen arguments

We have added statistical methods to all experiments where possible and have also added a Statistics section to the Material and Methods.

Line 1026: “Statistical analysis

Significances were analysed using GraphPad Prism 10 software. All histograms represent quantitative data expressed as mean \pm standard deviation or error on $n \geq 3$ replicates unless otherwise indicated. Unpaired t-test was used for the comparisons between two different conditions. When comparing one sample to a hypothetical value as control, a one-sample t-test was used. Experiments comparing three or more groups, were analysed using the ordinary one-way ANOVA with Tukey’s test for multiple comparisons⁷⁶.”

b) Line 99: More information on “drug like” could be helpful: what features are you referring to would good to improve

We have added a reference article that summarises the promises and pitfalls of developing chemical tools and drugs (Arrowsmith, C.H. et al. The promise and peril of chemical probes. Nat Chem Biol 11, 536-541 (2015). We have also provided some additional detail in the main text.

Line 98: “However, these published tool compounds lack features of high-quality chemical probes that in addition to physicochemical properties such as solubility can include confirmed selectivity, cellular potency, and the availability of structurally related inactive controls³¹. To date there are no reports of these initial tool compounds progressing to chemical series with more drug-like physicochemical and pharmaceutical characteristics^{31, 32}.”

c) Do they have kinetics of binding of compound 4, slow fast binder?

We have not carried out a kinetic assessment of compound **4**, however earlier compounds demonstrated fast on / fast off kinetics. Allosteric changes that cause substantial rearrangements to domains or protein structure often impact on the dynamics of binding and the regular fast on/off binding kinetics of a ligand. Our structural data indicated that the structural impact resulting from ligand binding were comparatively small and predicted not to cause deviation from standard fast on-off rates. However, we have included some additional ITC characterisation of compound **4** binding to eIF4E. We have added ITC data in a standard format (titrating eIF4E with compound **4**) and in a reverse format (titrating compound **4** with eIF4E). The data estimates a 1:1 stoichiometry with a thermodynamic profile showing a large favourable enthalpy and a small entropic penalty consistent with strong and direct interactions between the protein and ligand. We have edited the text and **supplementary figure 8** legend accordingly.

Line 239: “Thermodynamically, binding of compound 4 to both eIF4E variants was driven by a large favourable enthalpic contribution with a small entropic penalty

(Supplementary Fig. 8). In all cases the stoichiometry estimates indicated a 1:1 interaction between compound 4 and eIF4E.”

“Supplementary Fig. 8 Standard and reverse ITC titrations of compound 4 binding to eIF4E D127N and eIF4E D127. a Standard ITC titration of eIF4E D127N with compound 4. **b** Reverse ITC titration of compound 4 with eIF4E D127N. **c** Standard ITC titration of eIF4E D127 with compound 4. **d** Reverse ITC titration of compound 4 with eIF4E D127. Binding of compound 4 to eIF4E D127N was driven by a large favourable enthalpic contribution ($\Delta H = -12.0 \pm 0.5$ Kcal/mol) with a small entropic penalty ($-T\Delta S = 1.4 \pm 0.5$ Kcal/mol). Binding of compound 4 to eIF4E D127 showed a similar thermodynamic profile, with a large favourable enthalpy ($\Delta H = -10.2 \pm 0.4$ Kcal/mol) and a small entropic penalty ($-T\Delta S = 0.6 \pm 0.4$ Kcal/mol).”

d) Compound 4 seems to target only 4G-4E but not 4E-BP1-4E, but also does not facilitate binding of 4EBP1 to freed up binding site. Have they done binding of compound 4 to eIF4E in presence of 4G/4E-BP1?

We considered that binding studies with mixtures of purified full length eIF4E, eIF4G and 4E-BP1 would be too challenging and would be better assessed using cell lysates. We validated the lysate assays with the RIIY peptide derived from 4E-BP1 and negative control RIIG peptide, and showed the active peptide (**Fig. 4, a, b, c and f; Supplementary Fig. 9b**) disrupted both eIF4G1 and 4E-BP1 binding to eIF4E in HeLa and SW620 cells. In these lysates we also showed that the less active control compound 5 showed limited or no activity. We have included some additional new data using the electro-chemiluminescent binding assay to quantify a dose response inhibition of compound 4 against eIF4G1 binding to eIF4E, but no effect on disruption or recruitment, of 4E-BP1 binding to eIF4E in SW620 cells (**Fig. 4e**).

Line 285: “As before, compound 4 treatment of SW620 cells only affected eIF4G ($EC_{50} = 2.6 \mu M$) and not 4E-BP1 binding to eIF4E (Fig. 4b-d). Similarly, in HeLa cervical carcinoma cell lysates, treatment with compound 4 disrupted the interaction with eIF4G ($EC_{50} = 1.4 \mu M$) but not 4E-BP1 binding (Fig. 4b, c and e). Treatment with compound 5 had a limited effect on eIF4G binding compared to 4 (Fig. 4e).”

e) Fig4a: It is stated compound 5 doesn't have an effect but it is not quantified. It appears that 5 has an effect. I suggest quantification or adjustment of the strength of the statement.

We have amended the text as requested.

Line 272: “In the same assay, the weaker affinity compound 5, had a limited effect on eIF4G or 4E-BP1 binding at the top concentration tested (Fig. 4a).”

f) Most proliferative cells will show dependency on eIF4F as they are dependent on cap-translation (Fig5). Do you have perhaps any comparison? Otherwise I find this piece of data relatively weak to make a case for a specific eIF4E-dependency of the chosen cell line.

We agree most cells will show some dependency on the eIF4F complex. The rationale for showing the supplementary data here was to demonstrate that targeting eIF4E1 in

our selected H1299 NSCLC cell line, would lead to measurable cellular and molecular responses. We have now included public DEPMap data for CRISPR dropout screens of NSCLC lines in the respective panels. We clearly recapitulated the public data for H1299 from DEPMap showing significant loss of multiple guides targeting eIF4E1, indicating a reliance on eIF4E1 for survival. Additionally, targeting eIF4E2 and eIF4E3 had no impact suggesting there would be no compensatory redundancy from eIF4E isoforms following genetic targeting of eIF4E1.

Line 350: *“We selected the human H1299 NSCLC cell line for these studies as literature⁵⁹, CRISPR DEPMap public data (<https://depmap.org/portal/gene/EIF4E?tab=overview>) or in-house shRNA targeting eIF4F components, showed that eIF4A1, eIF4E1 and eIF4G1 were essential for H1299 cell survival (Supplementary Fig. 11).”*

g) Line 308: While I agree that its plausible that 4E knockout is lethal, I disagree that supplementary fig. 5 is necessarily proving this point (see above). Another piece of data would be great to support this argument or references.

We included the statement in **line: 308** primarily to emphasise the exploration of eIF4E by introducing non-functional mutations would be challenging or not possible without using a rescue approach. We have added a plot of DEPMap CRISPR gene score data for eIF4F components for non-small cell lung cancer cells and our selected line H1299 (**Supplemental Fig. 11**) that shows the essentiality of eIF4E1.

Line 393: *“A CRISPR knock-in approach to interrogate the function of eIF4E mutants would be challenging, as altered or loss of eIF4E function for the extended period of time required to establish a knockout or knock-in isogenic line is unlikely to be tolerated by H1299 cells (Supplementary Fig. 11).”*

h) Fig 5b does it take the lower expression into account? In my opinion this should be normalised to 4E variant expression levels to reveal the fraction 4G bound? While I agree with L85R, L134R might not actually a strong candidate. Also in regards to the disconnect between the assay this could help.

We have now also included a plot of eIF4G signal normalised to eIF4E variant expression (**Fig. 6d**). The data clearly shows that decreased expression of the variants alone cannot account for the significantly reduced eIF4G signal.

i) Supplementary Fig 8: How do they explain that the apparent readout is a lot lower than for example 5b. Do they have the expression data alongside?

The capture plates are made in batches and can show variation and also reduced signal with storage. Importantly all repeats and controls for the comparison of dTAG-ed and non-dTAG-ed eIF4E in Supplementary Fig. 8 (**now Supplementary Fig. 13a**) were run on the same plate and would be equally impacted by any batch effect.

j) What's the IC50 or at what IC is the dTAG^V-1 in the experiments?

The IC₅₀ for dTAG^V-1 in clone C3-1 is 206 nM and is used at ≈2-5x IC₅₀ in our experiments. We have added the data plot to Supplementary Fig. 14a and amended the text.

Line 415: *“Treatment with the heterobifunctional dTAG^V-1⁶⁴, that recruits VHL E3 ligase, for 6 hrs at ≈2-5x the IC₅₀ resulted in near complete loss of eIF4E-dTAG and a corresponding decrease in MCL1 protein for at least 72 hr following dTAG^V-1 treatment (Fig. 7b; Supplementary Fig. 14a, b).”*

k) It would be great to show 4EBP1 levels in blots of Figures 6-8

We have added a supplementary figure showing that loss of eIF4E in clone C3-1 does not significantly affect 4E-BP1 expression (**Supplementary Fig 15**) and have added a comment to the text. Given the absence of any effect of 4E-BP1 expression in the C3-1 clone following eIF4E degradation we did not think it was necessary to measure 4E-BP1 in the rescue experiments.

Line 418: *“We also measured 4E-BP1 expression and found no evidence for altered 4E-BP1 expression (Supplementary Fig. 15) following loss of eIF4E, so did not include 4E-BP1 in subsequent experiments.”*

Reviewer #3 (Remarks to the Author):

Summary of results described:

Sharp et al, present a new approach for discovery of small molecule inhibitors of the eIF4E/eIF4G interaction in order to find new compounds as anti-cancer agents. This interaction is naturally regulated by the 4E-BPs and their phosphorylation. Targeting the eIF4E/eIF4G This interaction has been of interest for many years, and a small molecule inhibitor, 4EGI-1 was described already in 2007. Despite many efforts, the exact mechanism of inhibition has not yet well understood, and there were several potential binding sites discussed.

Here the authors use a new approach by replacing the first 35 residues of eIF4E with the canonical binding sequence of 4E-BP1 attached with a flexible linker to L36. This resulted in a stable construct that could be overexpressed, a crystal structure could be solved with the linked 4E-BP1 fragment bound. The construct was also well suited for fragment-based screening, both searching for ligands in the cap site and the hydrophobic dorsal face.

Using this achievement, the authors applied a structure based search for inhibitors, using the Astex Pyramid compound platform, NMR and crystallography. They succeeded to get tight compound, the best was compound 4 with a K_d of 15 nM. This is great progress, and the authors should be complemented. However, the effect the improved inhibitor has on cancer cells is not so clear.

Subsequently, the authors applied a quantitative electro-chemiluminescent co-immunoprecipitation binding assay. They show that compound 4 displaces eIF4E:eIF4G but not eIF4E:4EBP1.

Mutants relevant to compound 4 binding yielded consistent results in binding and functional assays.

Concerns, suggestions and comments:

1. The authors mention numerous residues and mutations. However, the locations in the protein structure are difficult to follow. This would be much easier to follow if they included in the test or the supplement a figure with a sequence alignment of their construct with the wild-type. This should also include an alignment with the 4E-BP1 segment they used and those in the Sekiyama et al. and one of the Gruner papers. The data are given in Supplementary tables 1 and 2; however a figure would help and should be included.

We have included **Supplementary Fig. 1a and b** with sequence alignments of the canonical sequence with the constructs and key amino acid residues labelled. We have also included an alignment of the 4E-BP1 segment fused to eIF4E with the published 4E-BP1 and eIF4G peptide sequences.

It appears that the authors were not aware of the earlier structure for the eIF4E/4E-BP complex with a 40 residue 4E-BP1 segment (residues 44-84) published in 2015 by Sekiyama et al. (E4036–E4045 | PNAS | Published online

July 13, 2015). That paper also showed that the C-terminal loop of the 4E-BP1 peptide is needed for activity in a dual luciferase assay.

The Sekiyama structure of the complex is significantly different for the binding interface of the fused construct shown in Fig. 2a. The authors need to discuss this.

As outlined earlier exposure of the hydrophobic surface bound by eIF4G and the 4E-BPs affected the expression and purification of recombinant eIF4E, resulting in low yields and large amounts of aggregated material. The team felt that it would be more straightforward to block the hydrophobic surface with a short peptide sequence and to deliver a monomeric fusion protein rather than work with a multi-component protein-peptide complex. The disordered N-terminus of eIF4E was chosen to attach the 4E-BP1 peptide sequence partly due to its proximity to the canonical site and partly due to its disordered nature, it is not observed in crystal structures, so we felt that its modification was unlikely to impact on ligand binding events. Based on the experimental data we generated from multiple eIF4E constructs and identified a 4EBP1-eIF4E fusion protein that gave a suitable yield of soluble recombinant protein for screening. The modified eIF4E maintained the features of the published eIF4Es with a RMSD of 0.9 Å (**Supplementary Fig. 2a**). We have modified the text to emphasise that our engineered eIF4E maintained the structural features of the unmodified eIF4E. We only included residues 49-63 of 4E-BP that were sufficient to cover the hydrophobic patch so our structural data will not include the remaining c-terminal residues in the Sekiyama structure. However, we have added **Supplementary Fig 2c** comparing different 4E-BP structures to eIF4G.

Line 148: "Importantly, a global alignment of the engineered eIF4E structure with the X-ray crystal structure of wild-type unmodified eIF4E (from PDB structure 5T46), reveals similar structural features verified by the root-mean-square deviation (RMSD) of 0.9Å (Supplementary Fig. 2a), making it suitable for our fragment screen strategy."

Fig. 1 shows an overlay of the crystal structures of eIF4E with the eIF4G segment bound in the Gruner structure from 2016. It would be helpful to also overlay the structure of the eIF4E complex with a 40 residue 4E-BP1 segment (residues 44-84) published in 2015 by Sekiyama et al. That paper also showed that the C-terminal loop of the 4E-BP1 peptide is needed for activity in a dual luciferase assay.

We have added **Supplementary Fig. 2c** to show a sequence comparison of the peptide binding partners from Siddiqui 2011 – short 4E-BP1 peptide (3U7X), Peter 2014 – extended 4E-BP1 peptide (4UED), Sekiyama 2015 – extended 4E-BP1 peptide (5BXV) and Gruner 2016 – extended 4G peptide (5T46). The first only occupies the canonical binding site, the later 3, occupy the canonical site, traverse the lateral surface and engage the non-canonical site of eIF4E close to the flexible loop of site 2.

Lines: 454-458:

This section is not clear and must be changed to: Previously, Fischer and colleagues 64 described a biphenyl analogue of the original 4EGI-1 compound converted in a PROTAC tool. The biphenyl end binds to the same cavity as compound 4, which is adjacent to the original 4EGI-1 binding site 28, 51,64.

To make this section clearer we have edited this section.

Line 592: “Recently Fischer and colleagues⁴⁸ reported a biphenyl analogue of 4EGI-1, i4EG-BIP22 binding in the same cavity as compound **4**, rather than the original site reported for 4EGI-1^{22, 48, 68}. Although their study lacked a matched negative control compound and reported lower potencies (eIF4G peptide:eIF4E Fp assay IC₅₀ = 67 μM⁴⁸) than compound **4** (full-length eIF4G:eIF4E IC₅₀ ≈1-2 μM in lysates and intact cell engagement assays), its discovery nonetheless provides independent support for the functional relevance of site 2.

Lines 230-235:

The Authors suggest that the conformational change might explain the impact on 4G binding and cite figure 3c. It was difficult for me to see the conformational change in 3c, I assume it is the yellow helix. It might be easier to show an overlay of the original and the changed conformation. It seems that the c-term residue in their 4G peptide is causing the clash. This clash probably does not happen in the full-length 4G protein.

We have added an additional figure (Fig. 3c and moved Fig. 3c to 3d), updated the figure legend and edited the text.

Line 259: “Comparison of the compound 4-eIF4E structure with that of an eIF4G peptide⁵¹ bound to eIF4E suggested that this conformational change would potentially impact on binding of eIF4G due to steric clashes between residues H78-N84 from eIF4E with residues D638-L641 from the eIF4G peptide (Fig. 3c,d).”

Line 1284: “c Overlay of eIF4G peptide (green ribbon) with the protein conformation of compound 4 bound eIF4E (grey ribbon), the reorganised α1-helix and flexible loop region is displayed as a yellow ribbon. The original loop conformation from the 5T46 structure of eIF4E bound to an eIF4G peptide is displayed as a red ribbon. ”

Recommendation:

This is an important manuscript describing the discovery of highly efficient inhibitor of translation initiation following up of previous earlier attempts. The paper is suitable for publication in Nature Communications after several modifications, in particular those listed below.

We thank the reviewer for their positive comments and have addressed the points listed below

1. The binding of a bi-phenyl compound to the dorsal hydrophobic site of eIF4E has already been described recently in an attempt to use it for defining a degrader (Fischer et al., Eur.J. Medicinal Chemistry, 2019 (2021) 113435. The binding cavity is essentially the same as the one described here.

We have cited the Fischer (i4EG-BiP) reference in the results section where we describe the pockets found in our screen and acknowledge their discovery of the site.

Line 156: “Fifty fragment hits were identified to bind to eIF4E (3.6% hit rate), with a small number occupying the mRNA cap-binding site (site 1) (Supplementary Fig. 2b) and the majority binding at a second site of unknown functional relevance (site 2), which at the time was unreported, but was subsequently identified by Fischer and

colleagues for a biphenyl-derivative of 4EGI-1 (i4EG-BiP)⁴⁸ (Fig. 2a, b and Supplementary Fig. 2a).”

2. The crystal structure of the complex of eIF4E with an ~80 residue fragment of 4E-BP1 was first presented at a Cold Spring Harbor Symposium in 2014 and subsequently by Sekiyama et al. (E4036–E4045 | PNAS | Published online July 13, 2015) and must be cited.

We have added the Sekiyama and colleagues 2015 citation (**Line 183**)

3. The electro-chemiluminescent co-immunoprecipitation binding assay is very poorly described and should be improved to make it better understandable.

To improve understanding, we have modified the text and have also added more detail to the materials and methods section.

Line 276: “We also developed a quantitative electro-chemiluminescent co-immunoprecipitation binding assay to assess eIF4E-eIF4G and eIF4E-4E-BP1 interactions in cell lysates (Supplementary Fig. 9a). The assay uses microwell plates coated with an anti-eIF4E or an anti-Flag-epitope tag antibody to capture protein complexes containing endogenous eIF4E or exogenous Flag-tagged eIF4E from cell lysates. The captured eIF4E or binding partners (eIF4G or 4EBP1) are detected by specific secondary antibodies and quantified by an electro-chemiluminescent reaction.”

Reviewer #4 (Remarks to the Author):

The authors present the development of an inhibitor of eIF4E using a fragment-based approach. 4E is the mRNA cap binding protein in the eIF4F complex, which regulates the rate-limiting step of mRNA translation initiation. Because of its critical role in translation, 4E is important in cancer cell transformation and is therefore a target for developing anti-cancer drugs. Fragment-based screening is less biased than HTS and this, combined with several rounds of structure-based modifications led to the identification of a nM binding compound that disrupted the 4E:4G interaction, but not the 4E:4E-BP interaction, and inhibited cap-dependent translation in cell lysates. Structural analysis showed that the compound bound to a hydrophobic pocket away from the cap-binding site and near the non-canonical binding region of 4G where it causes the extension of helix $\alpha 1$ and a loop conformational change, which is enough to disrupt the 4G peptide interaction site. The activity of the compound was tested in intact cells via a genetic rescue approach that showed the compound could weakly inhibit 4G binding ($IC_{50} \sim 100 \mu M$) but lacked the ability to inhibit protein synthesis. Mutational analysis of the compound binding site did not provide significant insight on whether it has an important functional role.

Overall, this is an interesting and well-written paper which uses an array of experimental approaches to arrive at a potential inhibitor of an otherwise difficult to target protein. Particularly impressive were the genetic rescue experiments and the expression of a soluble form of human 4E by blocking the exposed hydrophobic patches with its natural ligand. However, the final results were not compelling as the compound was not very effective in intact cells and the analysis on its binding site did not reveal an interesting functional role. The biggest issue with the paper is that it seems to be presented as if the compound's binding site is novel and not yet known, when another compound (similar at least with regards to the core bi-phenyl group), i4EG-BiP, was already found to bind to the identical site and its similar conformational consequences documented. Although the i4EG-BiP study was mentioned in passing in the discussion, it was published over two years ago and therefore should be part of the introduction for what is already known regarding 4E, namely, that the present study identified a binding pocket which was previously identified by another study. That being said, this paper does highlight the power of the fragment-based approach for identifying binding pockets that HTS may miss.

We have cited the Fischer (i4EG-BiP) reference in the results section where we describe the pockets found in our screen and acknowledge their discovery of the site.

Line 156: *"Fifty fragment hits were identified to bind to eIF4E (3.6% hit rate), with a small number occupying the mRNA cap-binding site (site 1) (Supplementary Fig. 2b) and the majority binding at a second site of unknown functional relevance (site 2), which at the time was unreported, but was subsequently identified by Fischer and colleagues for a biphenyl-derivative of 4EGI-1 (i4EG-BiP)⁴⁸ (Fig. 2a, b and Supplementary Fig. 2a)."*

Other points to address:

1. Based on the structural analyses to date, 4G and 4E-BP use very similar binding modes and sites on 4E. There was no satisfactory explanation given as to why compound 4 affects 4G binding but not 4E-BP.

The differential effect observed for compound 4 on eIF4G and 4E-BP1 binding to eIF4E could be surprising as both proteins are observed to occupy similar canonical, non-canonical and lateral regions of eIF4E (Gruner *et al.* 2016; Peter *et al.* 2015). Although, several studies have also observed a differential binding phenomena with inhibition of eIF4G binding and recruitment of 4E-BP1 binding following incubation of lysates with 4EGI-1 and i4EG-BIP compounds (Fischer, P.D. *et al.* 2021; Moeke *et al.* 2007). Fischer and colleagues (2021) used modelling to propose that binding of the 4EGI-1 and i4EG-BIP small molecules impaired eIF4G binding through a steric clash with the extended eIF4E:4G interface, that would not occur with 4E-BP1, suggesting simultaneous binding of compound and 4E-BP1 to eIF4E that resulted from displacement of eIF4G and recruitment of 4E-BP1. However, overlays of 4EGI-1 with X-ray structural data for the eIF4E binding partners show a clash between 4EGI-1 and both eIF4G and 4E-BP1 peptides (4E-BP1 residues 78, 81 and 82 and eIF4G residues 636, 639 and 640 – illustrated below) that is inconsistent with the formation of three-way 4E-BP1:eIF4E:4EGI-1 complex. However, it should be remembered that a crystal structure provides a static snapshot view that contrasts with the potentially more complex and dynamic cellular environment.

Legend: eIF4E is represented as a green ribbon, 4EGI-1 is shown as grey space-filled Connolly surface, eIF4G peptide (PDB structure 5T46) is displayed in cyan and the 4E-BP1 peptide (PDB structure 4UED) is displayed in yellow.

Our data differed from that described for 4EGI-1 and i4EG-BIP small molecules, as we detected displacement of eIF4G in lysates by compound 4, but found no evidence for recruitment or displacement of 4E-BP1 in two different cell lines by co-immunoprecipitation or electro-chemiluminescent binding assay (**Fig. 4a, c, d and e**). In the same assays the RIIY competitive peptide derived from 4E-BP competed both 4E-BP1 and eIF4G1 binding (**Fig. 4a,b and d and Supplementary Fig. 9b**). We propose that compared to eIF4G, 4E-BP is more tightly bound to eIF4E as a result of interactions with the non-canonical binding region, including a more rigid proline

containing region linking the canonical and non-canonical motifs. This results in ≈ 5 to 10-fold tighter binding affinity for 4E-BP1:eIF4E compared to that for eIF4G:eIF4E, although this does come with the caveat that it is a comparison of different modified proteins and peptides from different studies (Tomoo *et al.* 2006; Abiko *et al.* 2007; Umenga *et al.* 2011; Gruner *et al.* 2016).

Overall, we believe our data reflects the binding affinity of compound **4** being sufficient to displace eIF4G from eIF4E in lysates and also prevent further recruitment of eIF4G to eIF4E, but is insufficient to displace 4E-BP1. The explanation for the lack of 4E-BP1 recruitment predicted to occur following displacement of eIF4G is less clear. We speculate that our eIF4E pull-downs could be detecting at least two separate pools of eIF4E, one bound to eIF4G that is affected by compound and another complexed with 4E-BP1 that is unaffected by compound exposure.

Experiments to address the multifactorial complexity of cellular binding and the apparent discrepancies between biophysical binding or X-ray structural data and cellular data will require multiple head-to-head competition experiments between truncated peptides and full-length proteins. Although these are highly interesting studies, we feel that these experiments fall outside the scope of the current manuscript. However, we have edited the discussion to reflect these comments.

Line 600: *“Displacement of eIF4G and recruitment of 4E-BP1 to eIF4E has been reported following incubation of lysates with 4EGI-1 and i4EG-BIP small molecules^{28, 51}. In contrast, despite validating the two binding assays with a 4E-BP-derived peptide, we consistently found no evidence for displacement or recruitment of 4E-BP1 following addition of compound **4** to cell lysates. This was unexpected as structural data suggests the conformational change induced by compound **4** binding to eIF4E should disrupt the non-canonical binding interface of both eIF4G and 4E-BPs (Supplementary Fig. 2c). We speculate we are measuring different pools of eIF4G:eIF4E and 4E-BP1:eIF4E and the differential displacement of eIF4G, but not 4E-BP1, by **4** in cell lysates may result from disruption of the pool of weaker eIF4G:eIF4E interactions alone^{51, 69-71}. These observations suggest further in-depth studies to understand the complexities of 4E-BP1 or eIF4G binding to eIF4E and the impact of small molecules in lysates or intact cells.”*

2. Given the significant difference in activity of compound **4 is cell lysates vs. intact cells, can the authors provide an explanation for this disparity? This is important as it could also explain why compound **4** had limited functional effects. How were permeability and efflux scores calculated?**

We have included methods and a table in the **Supplementary information** section detailing permeability, efflux, metabolism and physiochemical characteristics. This data indicates that permeability, efflux or metabolism are unlikely to account for the reduced activity in cells (**Supplementary information**).

We have now also employed a CETSA approach to explore engagement of **4** with eIF4E in intact cells (**Fig. 5a, b**). Here we clearly show that in cells **4** protects eIF4E from heat denaturation in a dose dependent manner with a 2 μ M EC₅₀. Importantly, the less active control compound **5** shows no evidence for target engagement at 50 μ M (**Fig. 5a**) in intact cells. The CETSA binding activity in cells reflected that seen in

the eIF4G1:eIF4E electro-chemiluminescent binding assay for three different cell line lysates (EC_{50} s 1.4 - 2.6 μ M; Figs. 4c-e and g).

The Caco2 permeability and particularly the CETSA data are important as they show that compound **4** is reaching its target in intact cells and that the poor or lack of functional activity in cell cannot be ascribed to a lack of target engagement in cells. Collectively these data coupled with the mutant and compound combination data suggest that at least in intact cells compound **4** binding to eIF4E is not sufficiently potent to disrupt both canonical and non-canonical interactions.

Line 317: “A cellular thermal shift assay performed in intact cells demonstrated target engagement, with compound **4** binding and stabilising eIF4E⁵⁷. We initially showed that treatment of H1299 cells with 50 μ M of **4**, but not **5**, protected eIF4E from thermal denaturation (Fig. 5a). We subsequently showed clear dose-dependent thermal protection of the eIF4E (EC_{50} = 2 μ M; Fig. 5b) in intact cells which was similar to the EC_{50} s for disrupting eIF4G:eIF4E binding in lysates (Fig. 4).”

Line 654: “Collectively, these data suggest that site 2 has a role in eIF4E function, but also that for site 2 binding by a small molecule to effectively inhibit eIF4E activity in cells the compound would need to have sufficient potency to disrupt both the canonical and non-canonical interaction with eIF4G.”

We also address the difference between lysates and cell rescue experiments in response to **comment 3** (below)

3. There are at least two mutations (W73F and H85R) that have been shown to disrupt eIF4G binding to eIF4E, but not the functionality of the eIF4F complex. Why is this the case?

In response to reviewer 2 we noted that unlike the other single or double mutants studied, only the single W73F mutant showed evidence for increased expression following dTAG treatment (**Supplementary Fig. 17b, new data in Fig. 8d and Supplementary Fig. 19c**). Treatment with the active compound **4** decreased W73F expression and also prevented the increased expression associated with loss of the dTAG-eIF4E (**Fig. 8d and Supplementary Fig. 19c**). We selected W73F for our study as it has been previously demonstrated that this mutant disrupts the canonical interaction of eIF4E with eIF4G and 4E-BP (Wendel *et al.*, 2007). Previous reports have also suggested that poor expression of a W73 mutant (W73A) was associated with loss of ligand binding leading to significantly elevated proteasomal degradation compared to the wild-type (Murata and Shimotohno, 2006).

The lysate binding assays with exogenous FLAG-tagged eIF4E also have endogenous eIF4E present in the cells. We speculate that WT or variant mutants that retain the ability to bind eIF4G (e.g. W56A) can compete with endogenous eIF4E binding to eIF4G that will be detected by the electrochemiluminescent FLAG-pull down assay (**Fig. 6**). In contrast, the W73F mutation weakens the affinity of eIF4E for eIF4G sufficiently that the W73F mutant cannot compete wild-type endogenous eIF4E binding to eIF4G binding in cells. This would explain the absence of binding to eIF4G in **Fig. 6** where endogenous eIF4E is expressed. In the rescue model the eIF4E-dTAG fusion protein would also compete with W73F for eIF4G binding, however, in the absence of the

competitor eIF4E-dTAG following dTAG^V-1 treatment the weaker binding W73F mutant is free to bind eIF4G with sufficient affinity to rescue function. This would also explain the increased levels of W73F protein following loss of dTAG-eIF4E as W73F will be protected from degradation through ligand binding. This would also explain the similar discrepancy observed with the L85R variant, although in this case the stability of the L85R mutation is less reliant on ligand binding.

The addition of the L85R to create the W73F/L85R double mutant would act cooperatively to weaken the binding affinity for eIF4G, to a degree where the double mutant could no longer bind eIF4G or rescue function even in the absence of wild-type eIF4E. Similarly treatment of the W73F mutant with compound **4** would weaken its affinity for eIF4G leading to reduced rescue and the observed decreased W73F protein expression compared to dTAG^V-1 alone condition.

Line 618: *“In addition to disrupting canonical binding of eIF4G and 4E-BP, W73 mutants are susceptible to proteasomal degradation thought to result from the loss of protective interactions with binding partners^{60, 62}. CETSA analysis showed that W73F was more susceptible to thermal denaturation but retained a site 2 capable of binding compound **4**. We speculate that in the co-immunoprecipitation binding experiments run in the presence of the endogenous eIF4E, the weaker binding exogenous W73F mutant cannot compete with the endogenous eIF4E for eIF4G-binding, resulting in the assay reporting an absence of binding for W73F. The rescue experimental format is run in the absence of any competitive eIF4E following dTAG^V-1 treatment and leaves the weaker binding W73F mutant free to bind eIF4G and rescue function. This would also explain the increased levels of W73F protein following loss of eIF4E-dTAG that would result from W73F being protected from degradation through eIF4G binding^{60, 62}.”*

4. Where are the X-ray data collection statistics? It would be useful to mention the resolution of your structures in the figure legends. Also, please show discovery maps of the compound fragment structures.

We have added **Supplementary table 1** containing data collection and refinement statistics and provided X-ray crystallography data statistics for the novel structures presented using template data table provided by the journal. We have also added difference density (Fo-Fc) maps for fragment hits, compounds **1** and **2**, identified during the screening phase (**Supplementary Fig. 3a and b**). Structure resolutions have been included in the legend for **Fig. 2**.

5. Only 3 hits (including the one in Supp. Fig. 1b) from the 1371 fragment library are shown, were there others at site 2?

There were 50 hits in total, the majority at site 2, we have added a sentence to the results section. To keep the narrative simple we have only included fragment hits that are directly related to the development of compound **4**.

Line 156: *“Fifty fragment hits were identified to bind to eIF4E (3.6% hit rate), with a small number occupying the mRNA cap-binding site (site 1) (Supplementary Fig. 2b) and the majority binding at a second site of unknown functional relevance (site 2), which at the time was unreported, but was subsequently identified by Fischer and*

colleagues for a biphenyl-derivative of 4EGI-1 (i4EG-BiP)⁴⁸ (Fig. 2a, b and Supplementary Fig. 2a).“

6. Line 140-142: “The X-ray structure of this engineered eIF4E matched X-ray crystal structures of wild-type unmodified eIF4E” - Indicate the RMSD between the structures.

We have added an RMSD value to the text.

Line 148: “Importantly, a global alignment of the engineered eIF4E structure with the X-ray crystal structure of wild-type unmodified eIF4E (from PDB structure 5T46), reveals similar structural features verified by the root-mean-square deviation (RMSD) of 0.9Å (Supplementary Fig. 2a), making it suitable for our fragment screen strategy.”

7. L217 show the ITC profile for compound 4 in Supplementary.

We have added ITC data in a standard format (titrating eIF4E with compound 4) and in a reverse format (titrating compound 4 with eIF4E) as **Supplementary figure 8**. The data estimates a 1:1 stoichiometry with a thermodynamic profile showing a large favourable enthalpy and a small entropic penalty consistent with strong and direct interactions between the protein and ligand. We have edited the text and figure legend accordingly.

Line 239: “Thermodynamically, binding of compound 4 to both eIF4E variants was driven by a large favourable enthalpic contribution with a small entropic penalty (Supplementary Fig. 8). In all cases the stoichiometry estimates indicated a 1:1 interaction between compound 4 and eIF4E.”

“Supplementary Fig. 8 Standard and reverse ITC titrations of compound 4 binding to eIF4E D127N and eIF4E D127. a Standard ITC titration of eIF4E D127N with compound 4. **b** Reverse ITC titration of compound 4 with eIF4E D127N. **c** Standard ITC titration of eIF4E D127 with compound 4. **d** Reverse ITC titration of compound 4 with eIF4E D127. Binding of compound 4 to eIF4E D127N was driven by a large favourable enthalpic contribution ($\Delta H = -12.0 \pm 0.5$ Kcal/mol) with a small entropic penalty ($-T\Delta S = 1.4 \pm 0.5$ Kcal/mol). Binding of compound 4 to eIF4E D127 showed a similar thermodynamic profile, with a large favourable enthalpy ($\Delta H = -10.2 \pm 0.4$ Kcal/mol) and a small entropic penalty ($-T\Delta S = 0.6 \pm 0.4$ Kcal/mol).”

8. Fig. 4a, why is there a second band in the eIF4G row input control pulldown?

The second band is either a non-specific band, a splice variant of eIF4G or a related isoform of eIF4G. Importantly, only the full-length eIF4G1 co-immunoprecipitates with eIF4E and the specificity of this binding was benchmarked using a competitive 4E-BP1 peptide and a paired negative control peptide.

9. Lines 328-337: As the authors further proceed with clone C3-1, Fig. 6 can be moved to the supplementary figures, and the explanations here can be shortened.

We have retained Fig. 6 (**now Fig. 7**) as in response to reviewer 2 we have now added some new proteome profiling data to that Figure (**Fig. 7d,e**)

10. It would be easier to follow the hit optimization if the 2D chemical structures were shown in Figure 2.

Including the 2D chemical representations with each of the figures during optimisation is a good idea, however we didn't feel there was enough space with each of the Figures. We will request that the editorial team includes table 1 in close proximity to Fig 2. We have also added **Supplementary Fig. 6** showing 2D Protein-ligand interaction diagrams depicting the major interactions between eIF4E and compound **1** or **4**

11. Fig 2b, the label for N127 spreads into a second line.

We have corrected the N127 figure label

12. Supplementary Fig. 2: structure figures are very difficult to discern with black background and dark blue colors.

We have replotted the figure with a lighter colour scheme

13. Supplementary Fig. 3a: The purple mesh should be brightened; what type of map is the mesh representing?

We have removed the mesh as it did not significantly contribute to the figure

14. Supplementary Fig. 4, L55: delete the letter C.

The data is now located in **Supplementary Fig. 9b** and is replotted as line graph.

15. L445 “We focused on a second site as multiple” ◇ change ‘a’ to ‘the’

Line 579: edited “a” to “the”.

16. L246, use a better synonym for “engenders”.

The sentence has been edited and moved to the discussion and no longer uses engenders.

17. L576, “20 mM phosphate” is mentioned twice.

Line 751: duplication removed.

Re: NCOMMS-23-44142: Integrating fragment-based screening with targeted protein...

Reviewer #1 (Remarks to the Author):

Upon review of the revised manuscript, it is evident that the authors considered the previous critiques adequately leading to a greatly improved manuscript. In my opinion, the manuscript is now ready for publication. This is poised to be a notable contribution to the eIF4E and fragment-based drug discovery literature.

We thank the reviewer for their positive comments.

Reviewer #2 (Remarks to the Author):

I acknowledge the effort and time Sharp and colleagues invested to address all my and the other reviewers comments. They have satisfyingly answered my questions, included new data and corrected/revised text accordingly, which has strongly improved understanding of the data and supports their conclusion well. I am happy to recommend the manuscript for publication.

We thank the reviewer for appreciating the effort and time that has gone into addressing their comments and ultimately improving the manuscript.

Reviewer #4 (Remarks to the Author):

The manuscript is significantly improved and easier to follow now. All the points were appropriately addressed,

Final minor points:

- 1. Supplementary Fig. 3, indicate the sigma levels at which the difference maps have been contoured.** We have added a new supplementary figure 3 that contains both the 2Fo-Fc and omit maps with sigma levels for compounds 1-5. Refinement statistics are included in supplementary table 1
- 2. Supplementary Fig. 7c, there appears to be a faint purple mesh around some of the residues, either remove it or brighten it and explain what it is.** We have removed the purple mesh.
- 3. pg. 8, L199: "elaborate" doesn't seem like the correct word to use here, perhaps "develop", "improve", or "enhance" is better.** We have edited "elaborate" to "progress". *"We used structure guided optimisation to progress the low affinity fragments into tight binding chemical leads to investigate the functional relevance of this site."*